# The small GTPase ARF3 controls invasion modality and metastasis by regulating N-cadherin levels

Emma Sandilands[1,2], Eva C. Freckmann[1,2], Erin M. Cumming[1,2], Alvaro Román-Fernández[1,2], Lynn McGarry[2], Jayanthi Anand[2], Laura Galbraith[2], Susan Mason[2], Rachana Patel[2], Colin Nixon[2], Jared Cartwright[3], Hing Y. Leung[1,2], Karen Blyth[1,2], and David M. Bryant[1,2]

ARF GTPases are central regulators of membrane trafficking that control local membrane identity and remodeling facilitating vesicle formation. Unraveling their function is complicated by the overlapping association of ARFs with guanine nucleotide exchange factors (GEFs), GTPase-activating proteins (GAPs), and numerous interactors. Through a functional genomic screen of three-dimensional (3D) prostate cancer cell behavior, we explore the contribution of ARF GTPases, GEFs, GAPs, and interactors to collective invasion. This revealed that ARF3 GTPase regulates the modality of invasion, acting as a switch between leader cell-led chains of invasion or collective sheet movement. Functionally, the ability of ARF3 to control invasion modality is dependent on association and subsequent control of turnover of N-cadherin. In vivo, ARF3 levels acted as a rheostat for metastasis from intraprostatic tumor transplants and ARF3/N-cadherin expression can be used to identify prostate cancer patients with metastatic, poor-outcome disease. Our analysis defines a unique function for the ARF3 GTPase in controlling how cells collectively organize during invasion and metastasis.

## Introduction

ARF GTPases are highly evolutionarily conserved regulators of membrane trafficking (Donaldson and Jackson, 2011; Sztul et al., 2019). ARF proteins co-ordinate membrane trafficking by regulating the local identity of the membrane to which they are recruited, such as through modulation of phospholipid composition via phosphatidylinositol kinases (Donaldson and Jackson, 2011; Nacke et al., 2021). This allows the recruitment of adaptor and coat proteins, facilitating membrane protein clustering and membrane deformation and ultimately leading to vesicle budding of encapsulated cargoes (Donaldson and Jackson, 2011). ARF GTPases are therefore central players in the localization of most membrane proteins and have emerged as key regulators of polarized cell behaviors underpinning cancer cell growth and metastasis (Casalou et al., 2020; Chen et al., 2022).

ARF GTPases cycle between GDP- or GTP-bound forms with the assistance of guanine nucleotide exchange factors (GEFs) and GTPase-activating proteins (GAPs; Adarska et al., 2021). Rather than consideration of ARF-GTP as active, and ARF-GDP as inactive, the full cycle of GTP loading of an ARF by a GEF to allow recruitment of effectors, followed by nucleotide hydrolysis by a GAP to return to GDP-ARF is required for ARF function (Sztul et al., 2019). Therein, an ARF GAP acts as both terminator and effector of the ARF GTPase cycle. In humans, five ARF GTPases

are divided into three classes based on homology: Class I (ARF1, ARF3), Class II (ARF4, ARF5), and Class III (ARF6).

A complication in unraveling ARF GTPase function is their high degree of similarity in sequence and consequently their overlapping ability to associate with GEFs, GAPs, and interactors (Sztul et al., 2019). For instance, Class I ARFs (ARF1, ARF3; ARF2 was lost in humans during evolution) differ by seven amino acids in their N- and C-terminal regions, while their core ARF domain regions are identical. Moreover, of the 17 GEF and 23 GAP proteins, many of these share the ability to modulate nucleotide association on most ARFs in vitro. ARF GTPases can also act in amplifying loops, with a GEF acting as an ARF-GTP effector to activate further ARFs (Li and Guo, 2022; Padovani et al., 2014). This complexity makes it difficult to predict how ARFs and their regulators contribute to cellular behavior from individual single interactions and sets the stage for a systems-level analysis to identify how these potentially overlapping components functionally contribute to morphogenesis.

Here, we present a system-level characterization of ARF GTPase function in collective cellular behaviors using large-scale timelapse imaging of the morphogenesis of prostate cancer cells in 3D culture, machine learning to identify distinct resulting phenotypes, and molecular characterization of behaviors. This

---

[1]School of Cancer Sciences, University of Glasgow, Glasgow, UK; [2]The CRUK Beatson Institute, Glasgow, UK; [3]Department of Biology, University of York, York, UK.

Correspondence to David M. Bryant: david.bryant@glasgow.ac.uk

work identifies a key role for the poorly studied ARF3 GTPase in controlling how cells collectively organize into distinct phenotypes. ARF3 controls the modality of invasion, between leader cell-led chains of invasion versus collective sheet movement, by associating with and controlling turnover of the adhesion protein N-cadherin. ARF3 therefore acts as a rheostat for the modality of invasion, which regulates metastasis in vivo and can be used to identify prostate cancer patients with metastatic, poor-outcome disease. Our approach therefore allows elucidation of distinct functions of ARF GTPases in collective morphogenesis.

## Results

### A 3D screen for ARF GTPase contribution to collective cancer cell behavior

We interrogated the functional contribution of ARF GTPases, their GEFs, GAPs, and known interactors and effectors, which we term the "ARFome," to cancer cell morphogenesis (Fig. 1 A). We engineered a lentiviral system that co-encodes an shRNA and membrane-targeted mVenus (mem:Venus) fluorescent protein to transduced cells (Fig. 1 B), and generated a highly validated library targeting all ARFs, GEFs, GAPs, and 72 known interactors (Fig. 1 C and Table S1). Examination of ARF GTPase expression across nine prostate cancer cell lines indicated that metastatic PC3 cancer cells showed high expression levels of all ARF GTPases compared to normal prostate cells (RWPE-1, PRECLH; Fig. S1, A–G), particularly in 3D compared to 2D culture (Fig. S1, H and I). PC3 cells also expressed almost all components of the ARFome (Fig. S1 J). When PC3 cells were plated on a thin coat of ECM as a suspension of single cells in low percentage ECM-containing medium, they formed heterogenous multicellular structures polarized around a central lumen, which we termed acini (Freckmann et al., 2022; Nacke et al., 2021). We used these PC3 acini to examine ARFome contribution to 3D morphogenesis as (i) they have high levels of all ARF GTPases, (ii) they, upon intraprostatic xenograft, provide a model for metastatic tumorigenesis, and (iii) we have shown that they can be used to identify ARF GTPase modules that regulate 3D invasion, in vivo metastasis, and predict patient survival (Nacke et al., 2021).

We developed a high-throughput, arrayed, live imaging-based screening approach to determine the 3D phenotype of ARFome component depletion on multi-day morphogenesis. Control (Scramble shRNA-expressing, Scr) 3D acini could be distinguished from non-shRNA-expressing acini by the presence (Fig. 1, D and E, white arrowheads) or absence (Fig. 1, D and E, red arrowheads) of mem:Venus fluorescence, respectively. 3D acini were imaged every hour for 96 h (Videos 1 and 2) and size, shape, and movement features were extracted for thousands of mem:Venus-positive acini per manipulation (Fig. 1, E and F; and Table S1). This live imaging approach revealed that multiple distinct 3D phenotypes occur in these cells in parallel, confirming our previous observations (Nacke et al., 2021). To detect these alternate phenotypes, we generated a machine learning classifier based on the Fast Gentle Boosting algorithm to define three morphogenesis classes with high fidelity to true user classification (91–97%): acini that are spherical ("Round"), acini

that are elongated ("Spindle"), and those that are locally invading ("Spread"), which were then applied to classify and quantify the phenotype of all acini (Fig. 1 G). Application of these classes to timelapse sequences indicated that distinct phenotypes arose from single cells, and that cells could stay in the same cell state throughout observation or cycle between states to give rise to alternate phenotypes (Fig. 1 H).

To identify the phenotypes of individual ARFome component depletion, we compared the relative fold-change in frequency of Round, Spindle, and Spread phenotypes within each shRNA-expressing condition to control cells (Scr shRNA) over 96 h of observation (Fig. 2 A and Fig. S1 K). The resulting relative change in each phenotype across time allowed division of shRNAs against ARFome components into seven distinct Phenotype Groups based on clustering, including highly round (Group 3), increased local spreading (Group 1), or increased spindle-type behaviors (Group 2; Fig. 2, A and B; and Fig. S1 K). Some targets (22%, 27 out of 116) had different shRNAs mapping to different Phenotype Groups, likely reflecting differing knockdown (KD) efficiencies. Application of these groupings to network analysis of ARFome interactions from literature and publicly available databases indicated phenotypic clusters centered around different ARF GTPases (Fig. 2, C and D), which could not be easily appreciated based on connections between nodes alone due to the highly interconnected nature of the ARFome. ARF4 and ARF5 associated with Phenotype Groups 4 and 6 that are characterized by minimal change relative to control cells (Fig. 2, C and D). While ARF6 was associated solely with Phenotype Group 1, both ARF1 and ARF3 had one shRNA in each of Phenotype Groups 5 and 1, which displayed a modest but robust increase in Spindle and Spread behaviors, respectively. We therefore subsequently focused on exploring how these two highly similar Class I ARFs contributed to 3D multicellular morphogenesis.

### Contextual regulation of shape and invasion by Class I ARF GTPases

We independently validated depletion of each Class I ARF GTPase using an orthogonal approach of lentiviral shRNA expression and stable antibiotic selection (Fig. S1, L and M). PC3 cells stably expressing shRNA (two per gene) to *ARF1* or *ARF3* were cultured in ECM and imaged every hour for 96 h, as described for the ARFome shRNA screen. Size, shape, and movement features were measured for each acinus and machine learning classifications applied to categorize and quantify Round, Spindle, and Spread phenotypes. Analysis of 3D phenotypes revealed that Area, as an indirect measure of growth, was unaffected by ARF1 or ARF3 depletion (Fig. 2, E–G). Either ARF1 or ARF3 depletion induced Spindle-type behaviors in acini at the expense of Round phenotype, but in the case of ARF3 the Spread phenotype was also induced (Fig. 2, E–G). This indicates that upon longer term selection for stable depletion these highly similar ARFs do not share identical phenotypes.

We examined Class I ARF contribution to a range of cellular behaviors. Depletion of neither ARF1 nor ARF3 affected cell proliferation in 2D or 3D culture (Fig. S2, A–E), corroborating a lack of effect on Area measurements in 3D culture (Fig. 2, F and G). The effect of Class I ARFs on individual cell shape in 2D

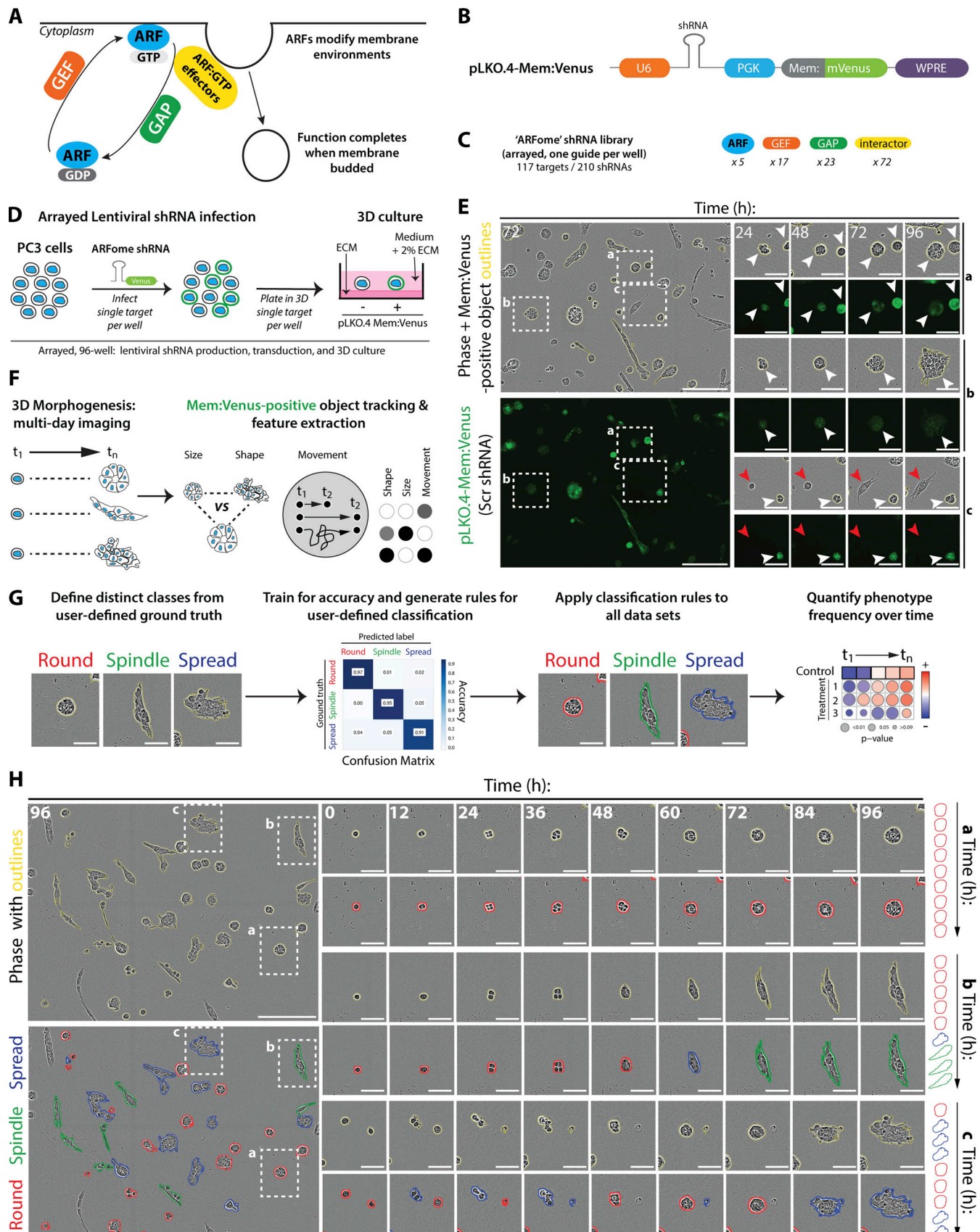

Figure 1. **Development of a 3D functional genomic screen to examine ARF GTPase contribution to collective cancer cell behavior. (A)** Cartoon, ARF GTPase cycle. **(B)** Cartoon, pLKO.4-mem:Venus shRNA lentiviral vector. PGK, phosphoglycerate kinase. WPRE, Woodchuck Hepatitis Virus post-transcriptional

regulatory element. **(C)** Schema, distribution of ARFome components in shRNA library. **(D)** Schema, 96-well based lentiviral infection of ARFome shRNA into PC3 cells that were then cultured as heterogeneous 3D acini in ECM. **(E)** Images of PC3 acini expressing pLKO.4-mem:Venus Scr shRNA. Yellow outlines, mem: Venus-positive acini. Scale bars, 300 µm. Magnified images of boxed regions (a–c) show acini at various times. Scale bars, 100 µm. White or red arrowheads, presence or absence of mem:Venus, respectively. **(F)** Schema, heterogeneous PC3 acini imaged over time vary in size, shape, and movement characteristics. Properties measured and information extracted for thousands of mem:Venus-positive acini. **(G)** Schema, phase images of three acini (yellow outlines) exhibiting morphological heterogeneity (Round, Spindle, Spread). Machine learning used to classify phenotypic states, train for accuracy, and generate user-defined rules. Rules then applied to all datasets; Round, Spindle, or Spread (red, green, blue outlines, respectively) and changes in global state frequency tracked over time. Scale bars, 100 µm. **(H)** PC3 acini (yellow outlines) and their user-defined classifications, Round, Spindle, or Spread (red, green, blue outlines, respectively). Scale bars, 300 µm. Magnified images of boxed regions (a–c) show classification of heterogeneous acini at various times. Scale bars, 100 µm. Cartoon, changes in user-defined classification over time.

culture was variably aligned with their respective collective 3D phenotypes upon ARF depletion (Fig. S2, E–H). ARF1 depletion increased the frequency of Spindle shape of single cells in 2D (Fig. S2, E–G), mirroring the collective Spindle phenotype induced in 3D upon ARF1 depletion (Fig. 2, E and F). In contrast, despite inducing both Spindle and Spread collective 3D behaviors, ARF3 depletion in 2D culture robustly induced Round single cell shape (Fig. S2, E, F, and H). Therefore, the effect of ARF3 on collective morphogenesis is specific to a 3D environment, not single cells. This emphasizes the requirement to examine ARF function in 3D systems that allow assessment of collective behaviors.

To determine the effect of Class I ARFs on collective behaviors, we examined the ability of wounded monolayers to invade, which can occur through wound repair via single-cell invasion, movement as a sheet, or as a leader cell-led chain of cells (Fig. 3 A; and Videos 3, 4, and 5). In the absence of exogenous ECM addition, this approach assays 2D migration. Plating of monolayers onto ECM and overlay of cells and wound with further ECM allows examination of collective invasion. Despite their differences in single cell shape effects, depletion of either ARF1 or ARF3 increased 2D migration ability, largely through the movement of single cells (Fig. 3, B and C). In 3D invasion contexts, either ARF1 or ARF3 depletion resulted in chain-type invasion mechanisms (Fig. 3, D and E; arrowheads), mirroring the induction of Spindle chains in 3D acinus culture in both conditions (Fig. 2, E–G). Co-depletion of ARF1 and ARF3 induced increased Spindle and Spread behaviors in both 3D and 2D culture and increased spindle-type invasion from wounded monolayers (Fig. 3, F–J). These data indicate individual and key roles for each of the Class I ARFs in suppressing invasion in cells and emphasize that the phenotype of ARF depletion is contextual on whether cells are assayed individually or collectively.

### ARF3 is a rheostat for the modality of collective invasion

Given our observations that depletion of ARF1 or ARF3 altered shape and movement in both 2D and 3D (Fig. 2, E–G and Fig. 3, B–E), we examined whether over-expression of Class I ARFs would also affect these processes. Overexpression of mNeon-Green (mNG)-tagged ARF1 (ARF1-mNG) or ARF3 (ARF3-mNG), both of which localized to intracellular puncta compared to cytoplasmic mNG alone, did not affect cell growth in either 2D or 3D contexts (Fig. S2, I–M), similar to depletion of these ARFs. The shape of 2D single cells was modulated by ARF1-mNG or ARF3-mNG in the converse fashion to depletion of each ARF: ARF1-mNG overexpression increased the Round single-cell

phenotype, while ARF3-mNG induced single cells to undergo spreading (compare Fig. S2, G and H to Fig. S2 N). This confirms distinct effects of ARF1 and ARF3 on 2D cell shape.

When examining the effects on cell movement, we observed that ARF1-mNG expression had no effect on 2D migration (Fig. S3, A and B) or 3D invasion (Fig. 4, A and B; white arrowheads demarcating chain-led invasion). In contrast, ARF3 drastically affected cell behaviors. ARF3-mNG overexpression increased both 2D wound closure and 3D invasion but did so by inducing sheet-like movement of the cell monolayer (Fig. 4, C and D; black arrowheads denoting sheet movement; Fig. S3, C and D). In 3D acinus culture, ARF1-mNG overexpression displayed largely no phenotypic alteration (Fig. 4 E and Fig. S2 I, bottom panel). In contrast, ARF3-mNG overexpression induced Spindle phenotypes at early time points that decreased over time, relative to control, while the Spread phenotype was robustly increased at all time points (Fig. 4 F and Fig. S2 I, bottom panel), mirroring the sheet like invasion of monolayers (Fig. 4 C; black arrowheads). This indicates a unique function of ARF3 as a rheostat that controls the modality of collective invasion; low ARF3 levels result in leader cell-led chain-type invasion, while elevated ARF3 levels switch cells to a collective sheet-movement invasive activity. It is important to note that these phenotypes manifest in 3D culture where collective activity is assayed for.

We mapped the unique ability of ARF3 to induce a collective sheet-type invasion phenotype by creating chimeras between ARF1 and ARF3, which only differ by 4 amino acids in their N-termini and 3 amino acids in their C-termini (Fig. 4 G). Examination of GTP-loading of ARF1, ARF3, and chimeras revealed that both ARF1-mNG and ARF3-mNG were GTP-loaded, with ARF1 displaying increased GTP levels compared to ARF3 (Fig. 4, G–I) and indicating that a lack of effect of ARF1 overexpression was not simply due to lack of GTP-loading of the tagged ARF1. ARF chimera with an ARF3 N-terminus and ARF1 C-terminus (3N/1C) displayed poor GTP loading despite robust expression, precluding full elucidation of the function of these alterations as this mutant may act as a dominant-negative GTPase. Conversely, ARF chimera with ARF1 N-terminus and ARF3 C-terminus (1N/ 3C) showed increased GTP-loading compared to ARF3 alone (Fig. 4, G–I).

We further characterized the functional effects on Class I ARF chimeras, examining their effect on localization, and 2D and 3D phenotypes (Fig. S3, E–K). Compared to the cytoplasmic and nuclear fluorescence of mNG alone, ARF3-mNG localized to intracellular puncta in 2D single cells and 3D acini (Fig. S3 E; white arrows; Fig. S3 H). The ARF 3N/1C-mNG chimera resulted in

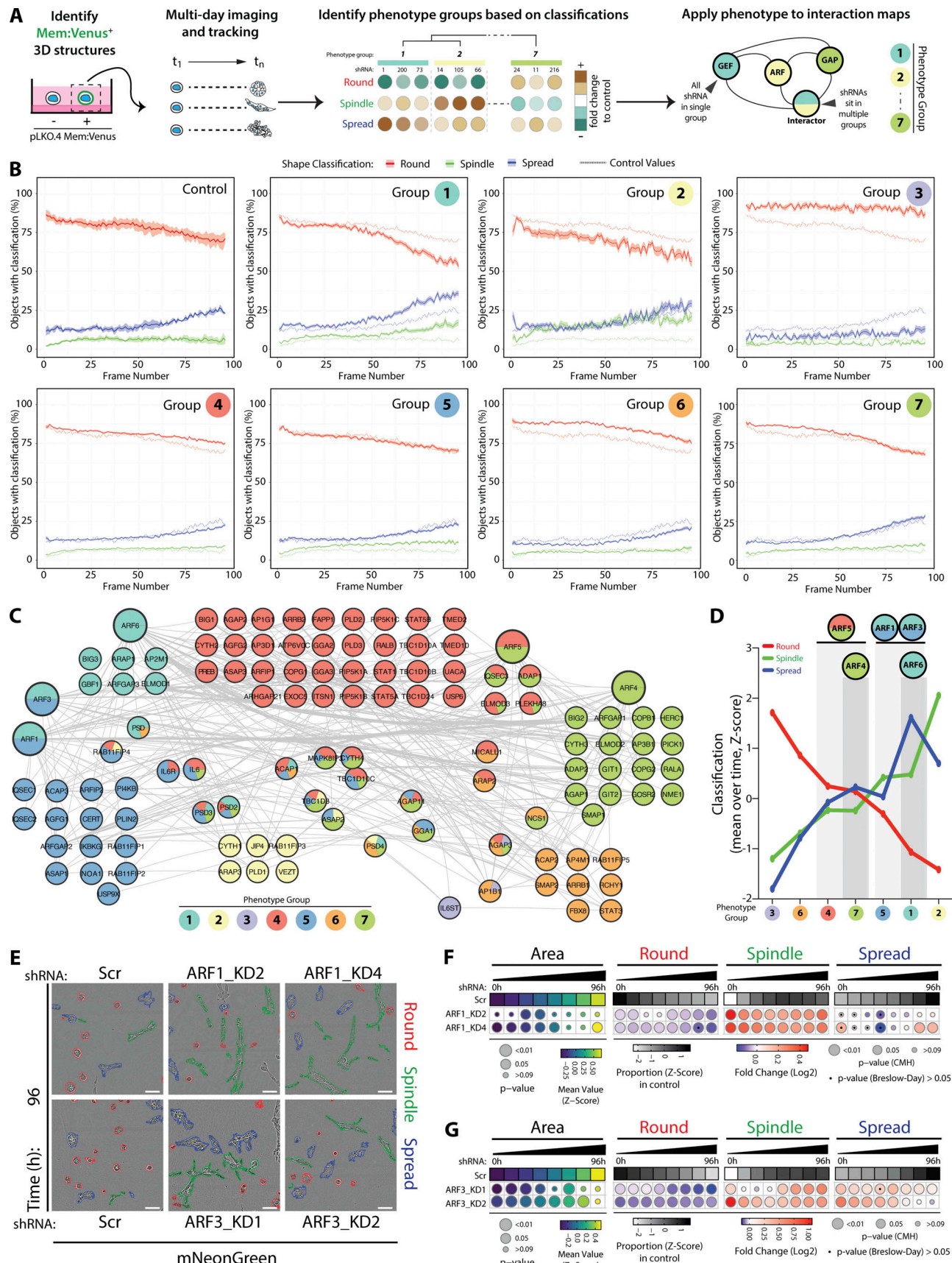

Figure 2. **Contribution of each component of the ARFome to collective cancer cell behavior was examined by individual depletion. (A)** Schema, PC3 acini expressing mem:Venus shRNAs were imaged, tracked, and classified. Phenotype Group 1–7 identified based on frequency of classification into Round,

Spindle, and Spread over time. Interaction map shown, shRNAs color-coded by Phenotype Group. **(B)** Graphs show percentage of acini classified as Round (red), Spindle (green), and Spread (blue) for each Phenotype Group and Scr shRNA (control, on each graph). Data are mean, shaded regions represent SEM. Viral infections and live 3D spheroid assays carried out 3 independent times. Each experimental replicate consisted of 18 technical replicates of Scr (170,674 acini in total) and 1 replicate of 210 ARFome shRNAs (Table S1). **(C)** STRING network analysis of acini visualized using Cytoscape. Phenotype Groups 1–7 identified by frequency of acini classification into Round, Spindle, and Spread. Colors indicate Phenotype Group, and the proportion of shRNAs for each target that sit in different groups is shown. **(D)** Graph is mean percentage of acini, across all time points, classified into Round, Spindle, and Spread per Phenotype Group. **(E)** PC3 acini expressing mNG and Scr, *ARF1*, or *ARF3* shRNA. Outlines: Round (red), Spindle (green), and Spread (blue). Scale bar, 100 µm. n = 4 and 6 experimental replicates for *ARF1* and *ARF3* shRNA, respectively, each with 4 technical replicates/condition. 16,760 (Scr), 21,086 (*ARF1*_KD2), 19,424 (*ARF1*_KD4), and 31,414 (Scr), 40,135 (*ARF3*_KD1), 30,460 (*ARF3*_KD2) acini quantified in total. **(F and G)** Quantitation of E. Heatmaps, Area is mean of Z-score normalized values (purple to yellow). P values, Student's *t* test, Bonferroni adjustment, represented by size of bubble. Heatmaps, Round, Spindle, or Spread is Log$_2$ fold change from control (Scr; blue to red). Proportion of control at each time is Z-score normalized (white to black). P values, CMH test, Bonferroni adjusted, represented by size of bubble. Dot indicates P value (Breslow–Day test, Bonferroni-adjusted) for consistent effect magnitude.

clustering of fluorescent puncta toward the cell periphery in 2D single cells (Fig. S3 E; black arrows; Fig. S3 F) and abrogated the ARF3-mNG induced increase in Spread phenotype observed in 2D (Fig. S3 G). A similar cell–cell contact-proximal localization was observed in 3D acini (Fig. S3 H). In contrast, the ARF 1N/3C-mNG chimera displayed enlarged puncta that were nonetheless reminiscent of the distribution of ARF3 in 2D single cells and 3D acini (Fig. S3, E and F). In acini, both ARF3-mNG and the ARF 1N/3C-mNG chimera puncta extensively co-localized with the Golgi marker GM130 and the recycling endosome marker RAB11 (Fig. S3 I), consistent with previous reports (Cavenagh et al., 1996; Kondo et al., 2012; Manolea et al., 2010). The ARF 3N/1C-mNG chimera, in contrast, maintained some colocalization with GM130 and RAB11, but the majority of localization occurred at cell–cell junctions. Phenotypically, the ARF3 N-terminus was dispensable, and C-terminus indispensable, to maintain sheet-type invasion (Fig. 4, J and K), and Spread-type acinus formation to levels reminiscent of ARF3 wild type (Fig. S3, J and K). These data indicate that ARF3 acts as a rheostat for the modality of invasion and that this function is dictated to the Class I ARFs by three unique residues in the ARF3 C-terminus (A174/K178/K180; Fig. 4 L).

### Identification of co-acting partnerships in the ARFome that regulate collective morphogenesis

We explored potential regulators and effectors of ARF3. In the morphogenesis ARFome screen, the ARFGEF PH and Sec7 Domain (PSD) displayed phenotypes similar to ARF3 (Fig. 2 C and Fig. S1 K). PSD is also known as Exchange Factor for ARF6 (EFA6A), due to its ability to strongly activate ARF6 GTP loading in solution; however, on membranes PSD is also a potent GEF for the Class I ARF, ARF1 (Padovani et al., 2014). Independent validation revealed that total levels of ARF3, but not ARF6, were increased upon PSD depletion (Fig. 5 A). Moreover, PSD KD resulted in a significant reduction of ARF3, but not ARF6, GTP loading (Fig. 5 B). PSD depletion mirrored the ARF3 depletion phenotype, resulting in increased Spindle and Spread behaviors in 3D and increased 3D chain-type invasion (Fig. 5, C and D; and Fig. 5 E, arrowhead; compare to Fig. 2, E–G and Fig. 3 E). Collectively, this suggests that in these cells PSD controls GTP loading of ARF3.

To identify a potential effector for ARF3, we examined the dual RAB11-GTP and ARF-GTP binding protein RAB11FIP4 (also known as Arfophilin-2), which displayed a similar phenotype to

ARF3 in the morphogenesis screen (Fig. 2 C and Fig. S1 K). We confirmed that RAB11FIP4 associated with both endogenous and mNG-tagged ARF3 (Fig. 5, F and G) and colocalized in puncta with ARF3 (Fig. 5 H). Given the association of RAB11FIP4 with both ARF3 and RAB11, we examined whether ARF3 depletion affected RAB11FIP4 endosomal distribution. ARF3 depletion had no effect on the number or size of RAB11FIP4 puncta, nor of their distribution when segmenting the cells into periphery ("Periph"), juxtanuclear ("Juxta"), or the regions between ("Cyto"; Fig. 5, I and J). RAB11FIP4 depletion, however, mirrored ARF3 KD, inducing both Spindle and Spread phenotypes in 3D, and Chain-type ECM invasion (Fig. 5, K–N, arrowhead; compare to Fig. 2, E–G and Fig. 3 E). This allows us to propose a model wherein while the endosomal recruitment of Rab11FIP4, which is likely controlled by Rab11 (Hickson et al., 2003; Wallace et al., 2002a; Wallace et al., 2002b), is independent to ARF3 binding. PSD activation of ARF3-GTP loading facilitates ARF3 association with Rab11FIP4 on endosomes to suppress invasive 3D behaviors (Fig. 5 O).

### N-cadherin is a key interactor of ARF3 that controls morphogenesis

As the PSD-ARF3-RAB11FIP4 module levels controlled the modality of collective movement in 3D, we examined whether ARF3 contributed to junctional organization between cells. Compared to mNG-expressing acini alone, ARF3-depleted acini displayed lowered overall F-actin intensity and a robust decrease of F-actin specifically at cell–cell, but not cell–ECM, junctions (Fig. 6 A, arrowheads; Fig. S4, A and B). In contrast, ARF3-mNG overexpression resulted in increased overall F-actin intensity, which was observed at the cell cortex (Fig. 6 A, arrows; Fig. S4, A and B). Analysis of the major cell–cell adhesion molecules E-cadherin and N-cadherin, which are co-expressed in PC3 cells, revealed that ARF3 levels associated with altered N-cadherin, but not E-cadherin, protein levels; ARF3 depletion decreased, while ARF3-mNG overexpression strongly increased, N-cadherin protein levels (Fig. 6, B–D). This decrease in N-cadherin protein levels upon ARF3 KD was mirrored by a decrease in N-cadherin mRNA levels (Fig. 6 E), initially suggesting that the ARF3 depletion phenotype could be explained by a transcriptional effect on N-cadherin. However, ARF3-mNG overexpressing cells also had decreased N-cadherin mRNA but displayed a robust elevation of N-cadherin protein. This suggests that ARF3 controls N-cadherin protein levels, with ARF3 levels uncoupling N-cadherin protein levels from mRNA levels.

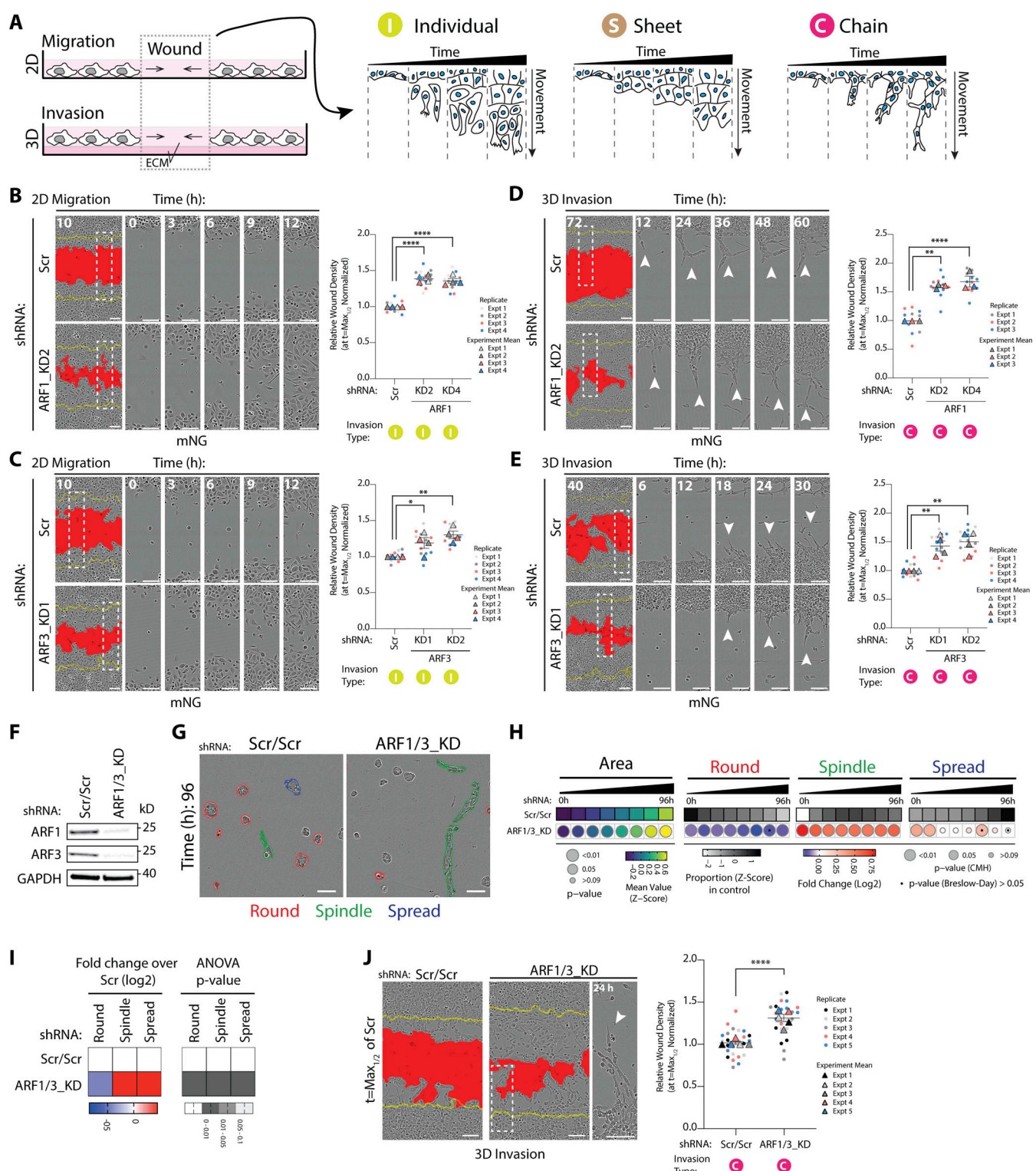

Figure 3. **Class I ARF GTPases regulate migration and invasion of prostate cancer cells. (A)** Schema, 2D migration or 3D invasion (+ECM) of wounded PC3 monolayer. Three modes of movement observed, cells moving individually (I), as a sheet (S), or as chains (C). **(B and C)** Phase images of cells expressing mNG and Scr or (B) *ARF1* or (C) *ARF3* shRNA in 2D migration assay. Yellow lines, initial wound, and red pseudo color, wound at $t$ = Max$_{1/2}$. Scale bars, 100 µm. Magnified images of boxed regions at different times shown. Graph is RWD at $t$ = Max$_{1/2}$ (Scr = 50% closed). Data is mean ± SEM (4 experimental replicates, triangles, 2–4 technical replicates, circles). P values (Student's two-tailed $t$ test), *P ≤ 0.05, **P ≤ 0.01, and ****P ≤ 0.0001. **(D and E)** Cells in a wounded monolayer overlaid with 25% ECM were imaged to observe 3D invasion. Phase images of cells expressing mNG and Scr, (D) *ARF1* or (E) *ARF3* shRNA shown. Yellow lines, initial wound, and red pseudo color, wound at $t$ = Max$_{1/2}$. Scale bars, 100 µm. Magnified phase images of boxed regions at different times shown. White arrowheads, invasive chains. Graph is RWD at $t$ = Max$_{1/2}$, normalized to Scr. Data is mean ± SEM. (3–4 experimental replicates, triangles, 3–5 technical replicates, circles). P values (Student's two-tailed $t$ test), **P ≤ 0.01 and ****P ≤ 0.0001. **(F)** Western blot analysis of PC3 cells expressing Scr/Scr or *ARF1/3*_KD

shRNA for ARF1 or ARF3. GAPDH is loading control for ARF3 and sample control for ARF1. Panels shown are representative of 3 independent lysate preparations. **(G)** Phase images of acini expressing Scr/Scr or *ARF1/3*_KD shRNA. Outlines: Round (red), Spindle (green), and Spread (blue). Scale bar, 100 μm. *n* = 5 experimental replicates each with 3–4 technical replicates/condition. 20,645 (Scr/Scr), 8,601 (*ARF1/3*) mem:Venus-positive acini quantified in total. **(H)** Quantitation of G. Heatmaps, Area is mean of Z-score normalized values (purple to yellow). P values, Student's *t* test, Bonferroni adjustment, represented by size of bubble. Heatmaps, Round, Spindle, or Spread is Log$_2$ fold change from control (Scr/Scr; blue to red). Proportion of control at each time is also Z-score normalized (white to black). P values, CMH test, Bonferroni adjusted, represented by size of bubble. Dot indicates P value (Breslow–Day test, Bonferroni-adjusted) for consistent effect magnitude. **(I)** 2D PC3 cells expressing Scr/Scr or *ARF1/3*_KD shRNA classified into Round, Spindle, and Spread. Heatmaps, Log$_2$ fold change over Scr/Scr. P values, one-way ANOVA, grayscale values as indicated. *n* = 2 experimental replicates with 4 technical replicates/condition. 3,323 (Scr/Scr) and 1,847 (*ARF1/3*) mem:Venus-positive cells quantified in total. **(J)** Phase images of cells expressing Scr/Scr or *ARF1/3*_KD shRNA in 3D invasion assay shown. Yellow lines, initial wound, and red pseudo color, wound at *t* = Max$_{1/2}$. Scale bars, 100 μm. Magnified image of boxed region shown. White arrowhead, invasive chain. Graph is RWD at *t* = Max$_{1/2}$, normalized to Scr/Scr. Data is mean ± SEM (5 experimental replicates, triangles, 4–5 technical replicates, circles). P values (Student's two-tailed *t* test), ****P ≤ 0.0001. Source data are available for this figure: SourceData F3.

N-cadherin could be recovered in ARF3 immunoprecipitants (Fig. 6 F) and co-localized with a subset of intracellular puncta positive for ARF3 (Fig. 6 G). Consistent with a decrease in total N-cadherin levels upon ARF3 depletion (Fig. 6, B and D), ARF3 KD did not affect the total cell area, but instead decreased both the number and the size of N-cadherin–positive puncta distributed throughout the cell (Fig. 6 H). Conversely, ARF3-mNG overexpression increased the total cell area, consistent with increased cell spreading (Fig. S2 N), and increased N-cadherin puncta number and size, particularly in non-peripheral regions (Fig. 6 I). This indicates that ARF3 controls the endosomal levels of N-cadherin.

We examined the identity of endosomes containing N-cadherin. Puncta positive for N-cadherin overlapped at a frequency of ∼20% with each of RAB4, RAB11, RAB11FIP4, and LAMP2 (Fig. 6, J and K). Depletion of ARF3 did not significantly affect localization of N-cadherin to RAB4 early recycling endosomes, LAMP2 late endosomes, or RAB11 recycling endosomes in general, but strongly increased localization of N-cadherin to RAB11FIP4-positive endosomes (Fig. 6 K). The percentage of RAB11 (P = 0.0111) and RAB11FIP4 (P = 0.0733) that co-localized with N-cadherin was decreased upon ARF3 depletion (Fig. 6 L). This suggests that ARF3 functions to couple N-cadherin specifically to a RAB11FIP4-positive subpopulation of recycling endosomes, to control total levels of N-cadherin protein.

To determine the consequence of ARF3 depletion-mediated uncoupling of N-cadherin trafficking at RAB11-RAB11FIP4 endosomes we examined N-cadherin surface distribution and turnover from the surface. While ARF3 reduction did not decrease surface N-cadherin, as determined by flow cytometry, ARF3-mNG overexpression robustly increased the steady-state N-cadherin surface levels (Fig. 6, M and N). Comparison of biotinylated N-cadherin at the cell surface (0 h, at 4°C without internalization) versus after 4 h of internalization (4 h, at 37°C), revealed that ARF3 depleted cells had accelerated, while ARF3-mNG overexpressing cells had delayed, turnover of N-cadherin from the surface (Fig. 6 O). ARF3 has been reported as part of the N-cadherin interactome (Li et al., 2019) and our data elucidate that ARF3 controls N-cadherin turnover from the cell surface by regulating association of internalized N-cadherin with RAB11-RAB11FIP4 recycling endosomes.

N-cadherin appeared to be a key cargo of ARF3 controlling morphogenesis. Unexpectedly, ARF3 and N-cadherin acted to mutually stabilize each other's level; while N-cadherin levels decreased or increased upon ARF3 depletion or overexpression,

respectively (Fig. 6, B and D), depletion of N-cadherin also decreased endogenous ARF3 levels (Fig. S4, C–F). In contrast, E-cadherin levels were not consistently changed upon alteration of either ARF3 or N-cadherin (Fig. 6, B and C; and Fig. S4 C). N-cadherin was essential for the switch between chain-type and sheet-type invasion, as depletion of N-cadherin increased 2D Spindle shape, chain-type invasion of 3D monolayers expressing mNG alone and completely reversed the sheet-type invasion of ARF3-mNG-expressing monolayers (Fig. S4 G; and Fig. 7, A and B). This was not simply due to a decrease in ARF3 levels upon N-cadherin depletion, as total levels of ARF3 were maintained upon ARF3-mNG expression in N-cadherin KD cells (Fig. S4, D and F). Moreover, N-cadherin depletion in control mNG-expressing cells phenocopied ARF3 depletion in 3D acini phenotypes (adoption of Spindle and Spread phenotypes; compare Fig. S4, H and I to Fig. 2, E and G), and completely reversed the spread-type phenotype of ARF3-mNG-expressing 3D acini (compare Fig. 4 F to Fig. 7, C and D). This effect on N-cadherin levels was also mirrored by depletion of the ARF3 effector RAB11FIP4 (Fig. S4 J). Taken together, these data indicate that ARF3, at Rab11-RAB11FIP4 endosomes, acts as a rheostat to control the turnover and total levels of N-cadherin to influence the modality of invasion (Fig. 7 E).

## ARF3 regulates metastasis in vivo

We examined the function of ARF3 in tumorigenesis in vivo through orthotopic intraprostatic xenograft of PC3 cells in control (mNG and Scr shRNA), ARF3-depleted (mNG and *ARF3* shRNA), or ARF3-elevated (ARF3-mNG and Scr shRNA) conditions (Fig. 8 A). Mice were examined at timed endpoint of 8 wk, which allowed for examination of effects on both primary tumor formation and metastasis (Nacke et al., 2021). No difference in cell engraftment or prostate weights at timed endpoint were detected between any conditions (Fig. 8, B and C) suggesting no effect on primary tumor growth, similar to a lack of effect on 2D or 3D proliferation in vitro upon ARF3 manipulation (Fig. S2, A–D and Fig. S2, J–M). Mirroring the effect on switching collective movement modality in vitro, ARF3 depletion versus overexpression showed robust and alternate effects on metastasis in vivo. ARF3 depletion induced a fully penetrant metastatic incidence (the number of mice with a primary tumor that also possessed at least one metastasis) compared to a reduction in metastatic incidence in ARF3-overexpressing cells (100% in ARF3 KD, 67% in ARF3 overexpression, compared to 75% in control; Fig. 8 D, P = 0.0308).

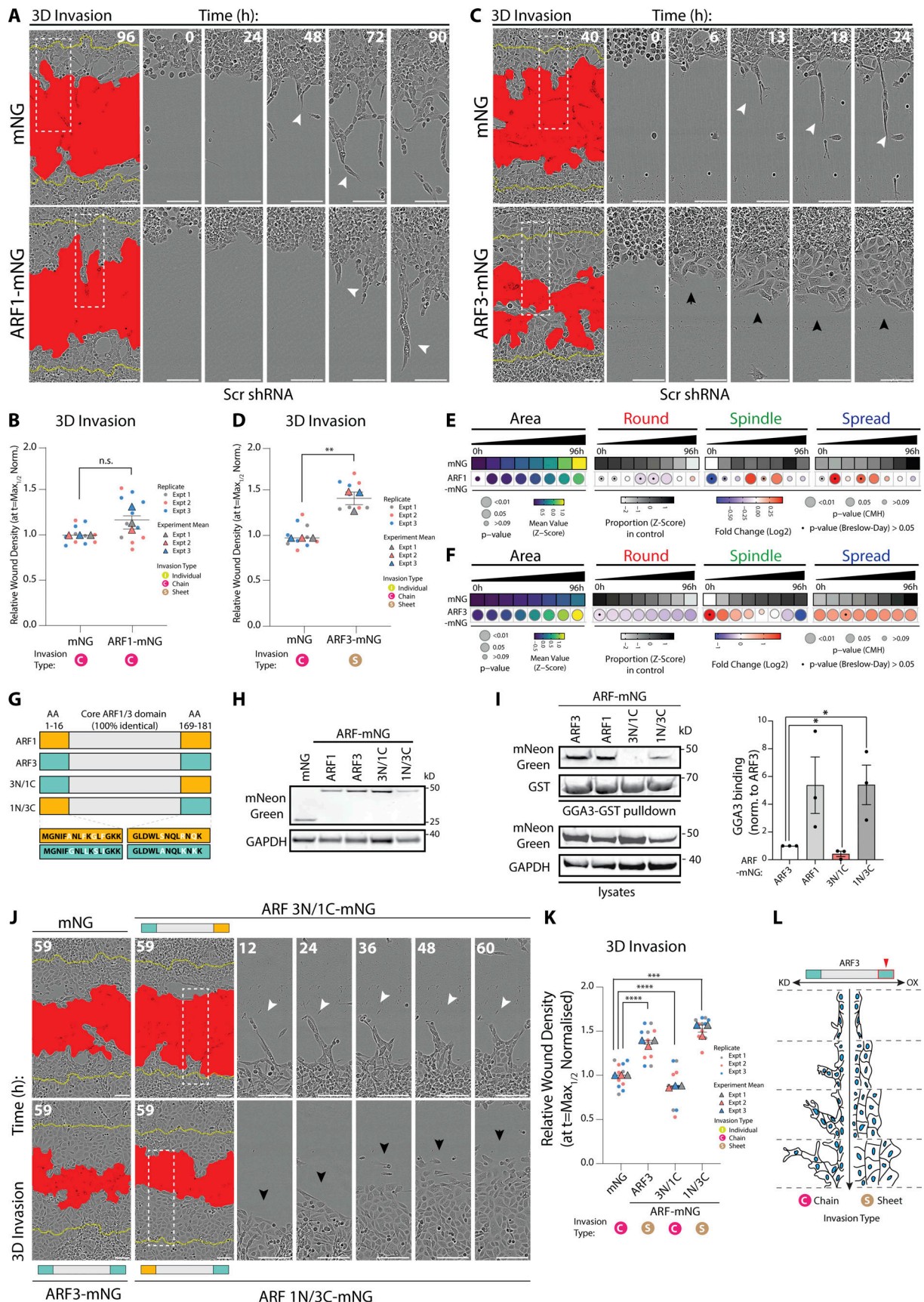

Figure 4. **ARF3 is a rheostat for the modality of collective invasion. (A–D)** PC3 cells expressing mNG, (A) ARF1-mNG, or (C) ARF3-mNG and Scr shRNA in 3D invasion assay. Yellow lines, initial wound, and red pseudo color, wound at $t = Max_{1/2}$. Scale bars, 100 μm. Magnified images of boxed regions shown. White

and black arrowheads, invasive chain or sheet, respectively. RWD at $t$ = Max$_{1/2}$, normalized to mNG is shown in graphs (B and D). Data are mean ± SEM (3 experimental replicates, triangles, 2–5 technical replicates, circles). P values (Student's two-tailed $t$ test), **P ≤ 0.01. **(E and F)** PC3 acini expressing mNG, (E) ARF1-mNG, or (F) ARF3-mNG and Scr shRNA were classified into Round, Spindle, and Spread. $n$ = 6 and 4 experimental replicates for ARF1-mNG and ARF3-mNG, respectively, each with 2–4 technical replicates/condition. (E) 5,005 (mNG), 1,938 (ARF1-mNG) and (F) 9,320 (mNG), 8,699 (ARF3-mNG) mNG-positive acini quantified in total. Heatmaps, Area is mean of Z-score normalized values (purple to yellow). P values, Student's $t$ test, Bonferroni adjustment, represented by size of bubble. Heatmaps, Round, Spindle, or Spread is Log$_2$ fold change from control (mNG; blue to red). Proportion of control at each time is Z-score normalized (white to black). P values, CMH test, Bonferroni adjusted, represented by size of bubble. Dot indicates P value (Breslow–Day test, Bonferroni-adjusted) for consistent effect magnitude. **(G)** Schema, Class 1 ARFs share 100% identical core region but differ in seven amino acids (AA) in N and C termini. ARF chimeras with ARF3 N-terminal and ARF1 C-terminal (3N/1C) and ARF3 C-terminal and ARF1 N-terminal (1N/3C) created. **(H)** Western blot of PC3 cells expressing mNG, ARF1-mNG, ARF3-mNG, and ARF-mNG chimeras for mNG and GAPDH, as loading control. Panels shown are representative of 3 independent lysate preparations. **(I)** ARF-GTP pulldown and representative Western blot for mNG, GST, and GAPDH, as loading control for both. $n$ = 3 independent lysate preparations and pulldowns. Graphs show mean GGA3 binding ± SEM normalized to ARF3. P values (Student's two-tailed $t$ test), *P ≤ 0.05. **(J and K)** PC3 cells expressing mNG, ARF3-mNG, or ARF-mNG chimeras plated in 3D invasion assay. Yellow lines, initial wound, and red pseudo color, wound at $t$ = Max$_{1/2}$. Scale bars, 100 µm. Magnified images of boxed regions shown. White and black arrowheads, invasive chain or sheet, respectively. RWD at $t$ = Max$_{1/2}$, normalized to mNG shown in K. Data is mean ± SEM (3 experimental replicates, triangles, 2–5 technical replicates, circles). P values (Student's two-tailed $t$ test), ***P ≤ 0.001 and ****P ≤ 0.0001. **(L)** Schema, ARF3 expression levels affect mode of invasion. Source data are available for this figure: SourceData F4.

Moreover, ARF3 depletion also increased the number of organs with metastasis per mouse compared to ARF3 overexpression (Fig. 8 E, P = 0.01), as well as expanded the metastatic tropism to all organs examined, bar the stomach, while ARF3 overexpression resulted in metastasis to only very proximal organs (lumbar lymph nodes, mesentery, and spleen; Fig. 8 F).

We examined whether N-cadherin localization or levels were altered in ARF3-manipulated tumors, similar to 2D or 3D PC3 cells. Primary tumors from control mice displayed variable regions of both N-cadherin–positive and –negative labeling, as well as regions of N-cadherin with different intensity (Fig. 8 G, upper panels). While the average region of tumor positive for N-cadherin as well as weighted histoscore for N-cadherin intensity trended toward the corresponding effects observed in vitro for ARF3 depletion vs. overexpression, these did not reach statistical significance (Fig. 8, H and I). Rather, ARF3 manipulation affected the homogeneity of N-cadherin distribution across tumors. While the aforementioned patches of N-cadherin expression occurred in control tumors, ARF3-depleted tumors displayed larger patches of weak or no N-cadherin labeling, with some regions of often smaller N-cadherin positivity (Fig. 8 G, middle panels). In contrast, ARF3-overexpressing tumors displayed large areas of somewhat homogeneous N-cadherin expression with 40% of tumors exhibiting > 40% N-cadherin positivity in comparison to 25% of ARF3-depleted tumors (Fig. 8 G, bottom panels, H). Collectively, this suggests that rather than controlling set levels of high or low N-cadherin (from overexpression or KD, respectively) ARF3 overexpression effects an even distribution of N-cadherin expression across tumor cells.

These data suggest that ARF3 is an in vivo regulator of metastasis, not primary tumor formation, through control of N-cadherin levels. Moreover, this suggests that of the alternate modalities of collective movement that ARF3 can influence in vitro, while sheet-type movement conditions may allow local metastasis, only the spindle-type chain-based invasive modality is able to induce distant and widespread metastasis.

### N-cadherin and ARF3 expression identify poor-outcome prostate cancer patients

We examined whether levels of ARF3 and/or N-cadherin may identify patients with poor outcome or metastatic disease. We first compared normal and tumor tissue mRNA levels of *ARF3* across tumor types from The Cancer Genome Atlas (TCGA) and the Gene Expression for Normal and Tumor database (which allows combination of multiple independent datasets; Park et al., 2019). This revealed that while *ARF3* mRNA levels are widely altered in tumor versus normal tissue, the directionality of *ARF3* mRNA alternation in tumors is dependent on tissue type and that *ARF3* mRNA expression is not a consistent indicator of clinical characteristics (Fig. S5, A and B). This was corroborated in prostate cancer by a lack of consistent alteration in independent datasets comparing *ARF3* mRNA levels across prostate normal tissue, primary tumor or metastasis (Fig. 9, A–E), or in the TCGA prostate adenocarcinoma dataset (PRAD Prostate; normal, $n$ = 86; tumor, $n$ = 323) for normal versus primary tumor or across Gleason Grades (Fig. S5, C and D).

Given the requirement of N-cadherin for ARF3 function in vitro, we examined whether the effect of ARF3 requires consideration of N-cadherin. Only four tumor types showed a consistent alteration in mRNA levels of N-cadherin (gene: *CDH2*) across the independent datasets: increased *CDH2* levels in thyroid (THCA) and kidney (KIRP), decreased *CDH2* mRNA levels in colon (COAD) and prostate (PRAD) adenocarcinoma (Fig. 9 F; and Fig. S5, E and F). Decreased *CDH2* mRNA levels were also observed.

In our studies, the levels of ARF3 uncoupled N-cadherin protein from mRNA level (Fig. 6, B, D, and E). We therefore surveyed N-cadherin protein levels compared to mRNA from the TCGA prostate cohort (from Reverse Phase Protein Array; note ARF3 not profiled). In prostate cancer patients, *CDH2* mRNA only modestly correlated with N-cadherin protein (Fig. S5 G). N-cadherin protein, but not mRNA levels, levels showed a modest decrease with progressive Gleason Grade (Fig. 9 H and Fig. S5 H). When divided into quartiles of expression, ascending N-cadherin protein expression, but not *CDH2* mRNA levels, showed a significant, inverse decrease in the frequency of patients presenting a new neoplasm following initial therapy (protein, P = 0.0074; mRNA, not significant; Fig. 9 I), whether patients were with or without tumor (protein, P = 0.0228; mRNA, not significant; Fig. 9 J), and lymph node metastasis positivity (protein, P = 0.0213; mRNA, not significant; Fig. 9 K). This further confirms that N-cadherin protein is partially

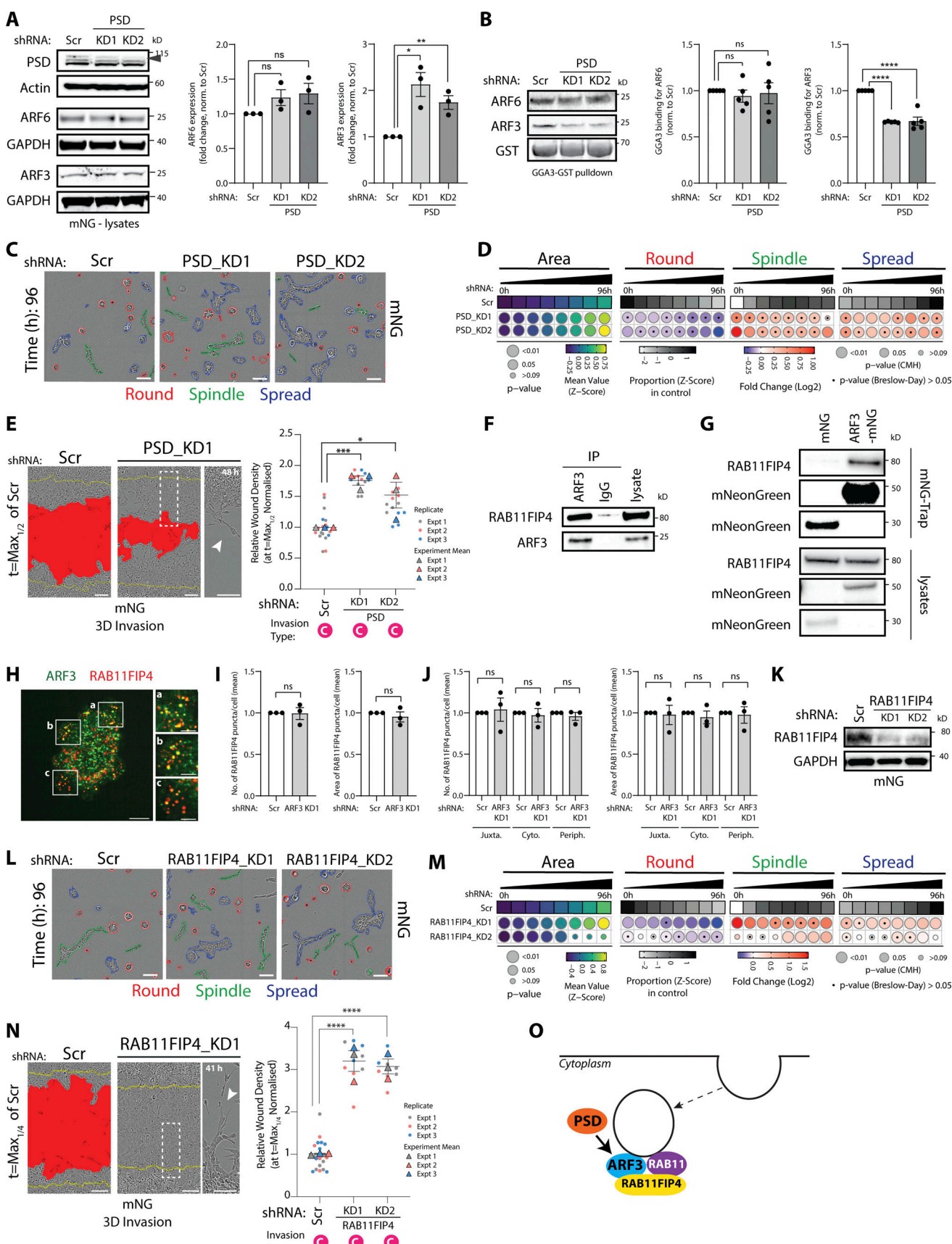

Figure 5. **Identification of co-acting partnerships in the ARFome that regulate collective morphogenesis. (A)** Western blot of PC3 cells expressing mNG and Scr or *PSD* shRNA for PSD, ARF6, ARF3, and actin or GAPDH, as loading control. Panels shown are representative of 3 independent lysate preparations.

Quantitation of ARF6 and ARF3 expression is shown as mean fold change ± SEM normalized to Scr. P values (Student's two-tailed t test), *P ≤ 0.05 and **P ≤ 0.01. **(B)** ARF-GTP pulldown and representative Western blot for ARF6, ARF3, and GST as loading control for ARFs in Scr or *PSD* shRNA cells. n = 5 independent lysate preparations and pulldowns. Graphs show mean GGA3 binding ± SEM normalized to ARF. P values (Student's two-tailed t test), ****P ≤ 0.0001. **(C)** Phase images of PC3 acini expressing mNG and Scr or *PSD* shRNA. Outlines: Round (red), Spindle (green), and Spread (blue). Scale bar, 100 µm. n = 2 experimental replicates each with 4 technical replicates/condition. 10,723 (Scr), 13,920 (*PSD*_KD1), 15,157 (*PSD*_KD2) acini quantified in total. **(D)** Quantitation of C. Heatmaps, Area is mean of Z-score normalized values (purple to yellow). P values, Student's t test, Bonferroni adjustment, represented by size of bubble. Heatmaps, Round, Spindle, or Spread is $Log_2$ fold change from control (Scr; blue to red). Proportion of control at each time is Z-score normalized (white to black). P values, CMH test, Bonferroni adjusted, represented by size of bubble. Dot indicates P value (Breslow–Day test, Bonferroni-adjusted) for consistent effect magnitude. **(E)** PC3 cells expressing mNG and Scr or *PSD* shRNA in 3D invasion assay. Yellow lines, initial wound, and red pseudo color, wound at $t = Max_{1/2}$. Scale bars, 100 µm. Magnified image of boxed region shown. White arrowhead, invasive chains. RWD at $t = Max_{1/2}$ is shown, normalized to Scr. Data is mean ± SEM (3 experimental replicates, triangles, 4–8 technical replicates, circles). P values (Student's two-tailed t test), *P ≤ 0.05 and ***P ≤ 0.001. **(F)** IP was performed using an anti-ARF3 antibody or mouse IgG and samples immunoblotted for RAB11FIP4 and ARF3. Panels shown are representative of 3 IPs from 3 independent lysate preparations. **(G)** IP was performed using mNG-Trap Agarose beads in cells expressing mNG or ARF3-mNG. Samples were immunoblotted for RAB11FIP4 and mNeonGreen. Panels shown are representative of 2 IPs from independent lysate preparations. **(H)** Image of PC3 cell stained with ARF3 (green) and RAB11FIP4 (red). Scale bars, 20 µm. Magnified images of boxed regions shown (a–c). Scale bars, 10 µm. Images representative of phenotypes observed in 3 experimental replicates. 40.2 ± 4% of ARF3 positive puncta overlap with RAB11FIP4 positive puncta in 320 cells quantified. **(I and J)** PC3 cells expressing mNG and Scr or *ARF3* shRNA were stained for RAB11FIP4, High-Content Screening Whole Cell Stain (HCS WCS), and Hoechst. Number and area of RAB11FIP4 puncta was quantified (I) per cell or (J) per sub-cellular region in each cell. n = 3 independent experiments with 508 (Scr) and 613 (*ARF3* KD) cells quantified in total. Data are presented as mean ± SEM normalized to Scr. P values (Student's two-tailed t test). **(K)** Western blot of PC3 cells expressing mNG and Scr or *RAB11FIP* shRNA for RAB11FIP and GAPDH, as a loading control. Panels shown are representative of 3 independent lysate preparations. **(L)** Phase images of PC3 acini expressing mNG and Scr or *RAB11FIP4* shRNA. Outlines: Round (red), Spindle (green), and Spread (blue). Scale bar, 100 µm. n = 3 experimental replicates each with 3–4 technical replicates/condition. 14,551 (Scr), 16,435 (*RAB11FIP4*_KD1), 11,880 (*RAB11FIP4*_KD2) acini quantified in total. **(M)** Quantitation of L. Heatmaps, Area is mean of Z-score normalized values (purple to yellow). P values, Student's t test, Bonferroni adjustment, represented by size of bubble. Heatmaps, Round, Spindle, or Spread is $Log_2$ fold change from control (Scr; blue to red). Proportion of control at each time is Z-score normalized (white to black). P values, CMH test, Bonferroni adjusted, represented by size of bubble. Dot indicates P value (Breslow–Day test, Bonferroni-adjusted) for consistent effect magnitude. **(N)** PC3 cells expressing mNG and either Scr or *RAB11FIP4* shRNA in 3D invasion assay. Yellow lines, initial wound, and red pseudo color, wound at $t = Max_{1/4}$. Scale bars, 100 µm. Magnified image of boxed region shown. White arrowhead, invasive chains. RWD at $t = Max_{1/4}$ is shown, normalized to Scr. Data is mean ± SEM. (3 experimental replicates, triangles, 3–8 technical replicates, circles). P values (Student's two-tailed t test), ****P ≤ 0.0001. **(O)** Schema, relationship between PSD, ARF3, and Rab11FIP4. Source data are available for this figure: SourceData F5.

uncoupled from *CDH2* mRNA levels, and that low N-cadherin levels identify patients with recurrent, metastatic tumors.

Analysis of progression-free survival of prostate cancer patients (TCGA) indicated that neither *ARF3* nor *CDH2* mRNA levels could stratify patient groups with altered survival (Fig. 9, L and M). In contrast, patients with lowest N-cadherin protein levels showed drastically reduced progression-free survival (compare lowest quartile [Q1] to all other patients, Q2–4; P = 0.0007; Fig. 9 N).

We examined whether combining N-cadherin protein expression with *ARF3* mRNA levels would further stratify patient survival by comparing patient groups segregated by expression based on a median split (M1, low; M2, high; Fig. 9 O). Patients with low N-cadherin protein (M1) showed similar survival regardless of *ARF3* mRNA levels (yellow and blue lines). In high N-cadherin expressing patients (M2) the mRNA levels of *ARF3* divided survival patterns. Low *ARF3* (M1) mRNA despite high N-cadherin expression (M2; green line) reduced survival to levels mirroring low N-cadherin protein. Conversely, having both high N-cadherin protein (M2) and *ARF3* mRNA (M2; red line) identified patients with best progression-free survival. Similarly, high levels of CDH2 protein and *ARF3* mRNA identified a patient group with lowest levels of new tumor formation after initial therapy, while all other groups showed similar rates (Fig. S5 I).

This clinical data is consistent with our in vitro data identifying a co-operation between N-cadherin and ARF3, wherein ARF3 and N-cadherin mutually control each other's levels and function in tumorigenesis, with reduced N-cadherin protein associated with metastatic, recurrent prostate cancer. These

data also indicate ARF3 as a contextual regulator of N-cadherin protein levels during tumorigenesis in prostate cancer patients.

## Discussion

The use of 3D culture allows the assessment of how individual genes or entire pathways contribute to collective cell behaviors. The application of a screening approach to 3D requires a number of adaptations not directly transferrable from the screening of 2D cell cultures. First, collective morphogenesis occurs over multiple days, requiring stable genetic manipulations to which siRNA transfections are poorly suited. Moreover, 3D phenotypes can be stereotyped but asynchronous or multiple phenotypes can occur in parallel in the same well. This requires live imaging of 3D morphogenesis to capture and quantify these considerations.

A bottleneck in the systematic screening of collective morphogenesis (arrayed, one manipulation per well) is the plating of multiple parallel manipulations into 3D culture at the same starting density. This is essential to ensure that phenotypes quantified do not simply represent morphogenesis from a different starting point, such as altered density. If the genetic manipulation of interest alters proliferation before plating into 3D culture, this is a technical challenge in assuring similar plating. We have overcome these obstacles using an arrayed shRNA library vector that co-encodes mem:Venus to allow plating at similar density in 3D culture. Through the use of phase contrast and fluorescent imaging of 96-well plates of 3D culture, tracking of morphogenesis of mem:Venus-positive acini, and machine learning–based classifications of distinct phenotypes,

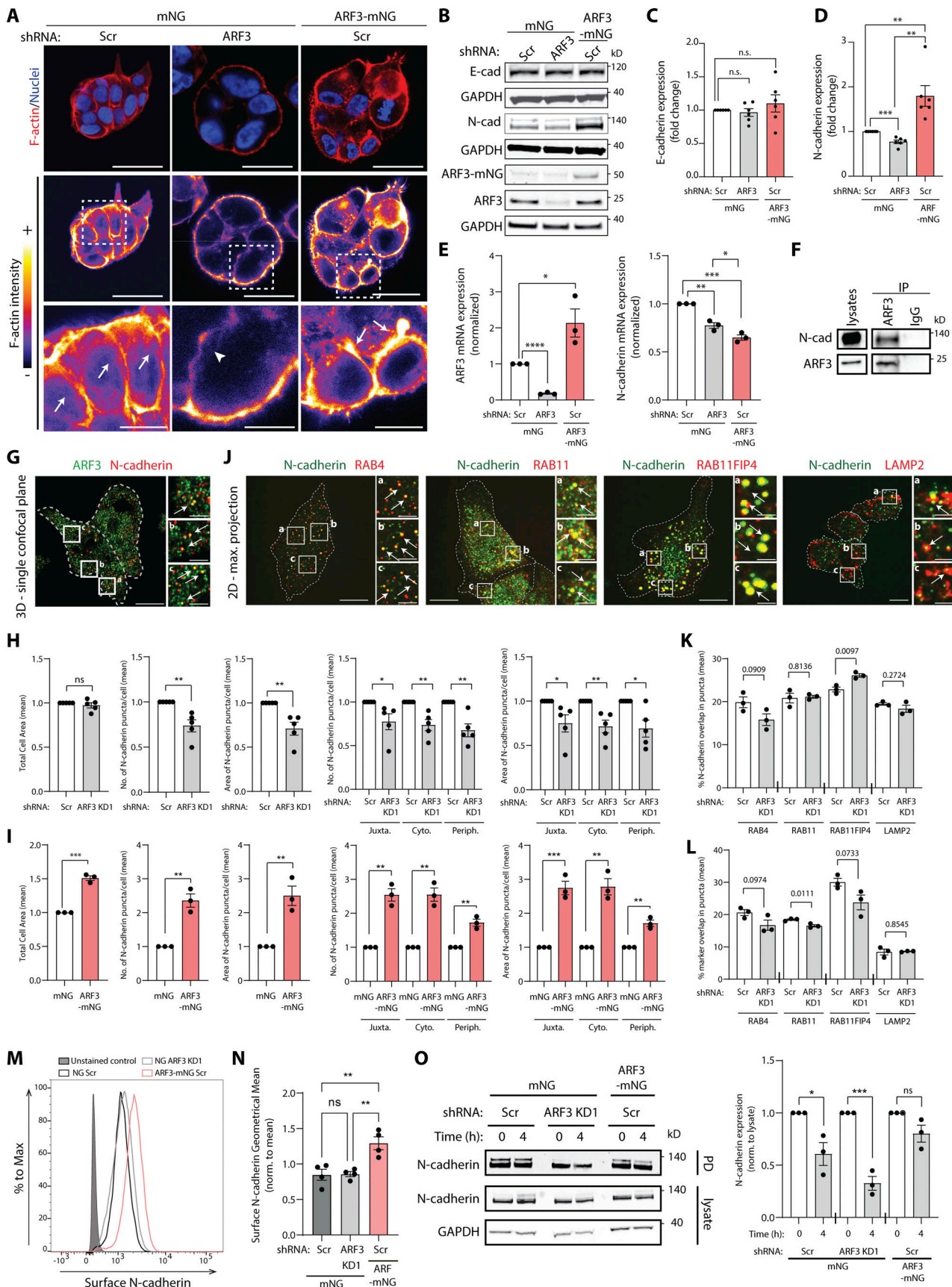

Figure 6. **ARF3 controls N-cadherin turnover from the cell surface by regulating association of internalized N-cadherin with recycling endosomes.**
**(A)** Confocal images of PC3 acini expressing mNG or ARF3-mNG and either Scr or ARF3 shRNA stained with F-actin (red) and Hoechst (nuclei, blue). F-actin

intensity, FIRE LUT. Scale bars, 20 µm. Magnified images of boxed regions shown. White arrows or arrowheads, presence or absence of intense F-actin staining in junctions, respectively. Scale bars, 7 µm. Images representative of phenotypes observed in 3 experimental replicates. **(B–D)** Representative Western blot of PC3 cells (B) expressing mNG or ARF3-mNG and Scr or *ARF3* shRNA for E-cadherin, N-cadherin, and ARF3. GAPDH loading control is shown for each blot. *n* = 6 independent lysate preparations. Data is presented in C and D as mean fold change ± SEM normalized to Scr. P values (Student's two-tailed *t* test), **P ≤ 0.01. **(E)** *ARF3*, N-cadherin, and GAPDH mRNA expression in PC3 expressing mNG or ARF3-mNG and either Scr or *ARF3* shRNA was determined by RT-qPCR. *n* = 3 independent RNA and cDNA preparations with 4 technical replicates/condition. Data is mean fold change ± SEM normalized to GAPDH then to Scr. P values (Student's two-tailed *t* test), *P ≤ 0.05, **P ≤ 0.01, ***P ≤ 0.001, and ****P ≤ 0.0001. **(F)** IP was performed using an anti-ARF3 antibody or mouse IgG and samples immunoblotted for N-cadherin and ARF3. Panels shown are representative of 3 IPs from 3 independent lysate preparations. **(G)** Confocal image of PC3 acini stained for ARF3 (green) and N-cadherin (red). Scale bars, 20 µm. Magnified images of boxed regions shown (a–c). Scale bars, 5 µm. White arrows, co-localization in subset of puncta. Image is representative of co-localization observed in cells in 3 experimental replicates. **(H and I)** PC3 cells expressing (H) mNG and Scr or *ARF3* shRNA or (I) mNG and ARF3-mNG with Scr were stained for N-cadherin, HCS WCS, and Hoechst. Cell area, number, and area of N-cadherin puncta was quantified (H) per cell or (I) per sub-cellular region in each cell. *n* = 5 (1 technical replicate) or 3 (5 technical replicates) independent experiments, respectively. 542 (Scr), 664 (*ARF3* KD1), 5,889 (mNG), and 4,353 (ARF3-mNG) cells quantified in total. Data is presented as mean ± SEM normalized to Scr. P values (Student's two-tailed *t* test), *P ≤ 0.05, **P ≤ 0.01, and ***P ≤ 0.001. **(J)** Images of PC3 cells stained with N-cadherin (green) and RAB4, RAB11, RAB11FIP4, or LAMP2 (red). Scale bars, 20 µm. Magnified images of boxed regions shown (a–c). White arrows, co-localization in subset of puncta. Scale bars, 5 µm. Images representative of phenotypes observed in 3 experimental replicates. **(K and L)** Quantitation of percent overlap of N-cadherin positive puncta with puncta positive for various sub-cellular markers (K) and % overlap of markers with N-cadherin puncta in Scr or *ARF3* KD1 cells are shown (L). Data is presented as mean ± SEM *n* = 3 independent experiments. 307 and 388 (RAB4), 243 and 195 (RAB11), 309 and 369 (RAB11FIP4), and 328 and 369 (LAMP2) cells quantified for Scr and *ARF3* KD1, respectively, in total. P values stated, (Student's two-tailed *t* test). **(M and N)** Flow cytometry was performed on PC3 cells with anti-N-cadherin antibody. Representative plot (M) and geometrical mean ± SEM (N) of surface N-cadherin levels are presented. *n* = 4 independent experiments. P values (Student's two-tailed *t* test), **P ≤ 0.01. **(O)** Surface proteins in PC3 cells expressing mNG or ARF3-mNG and either Scr or ARF3 shRNA were biotinylated and N-cadherin levels were analyzed by Western blot after internalization at 0 or 4 h. GAPDH was used as loading control for lysates. Data is presented as mean ± SEM with pulldowns (PD) normalized to lysates relative to control cells (0 h). *n* = 3 independent experiments. P values (Student's two-tailed *t* test), *P ≤ 0.05, and ***P ≤ 0.001. Source data are available for this figure: SourceData F6.

we were able to perform a functional genomic characterization of the ARFome contribution to collective morphogenesis.

It is important to note that despite using a library with high independent validation of target depletion from shRNAs, our approach is not an exhaustive analysis of every ARFome member. Rather, shRNAs are assigned into Phenotype Groups based on their relative change in Round, Spindle, or Spread acinus phenotype over time compared to a control (Scr) shRNA (Fig. S1 K). While this can detect phenotypes such as being highly Round (Group 3, e.g., *IL6ST*) or highly Spindle (Group 2, e.g., *RAB11FIP3*), phenotypes with modest change to control (Group 7, e.g., *ARF4*) were also identified. Modest changes can occur due to a bona fide lack of strong phenotype or could be due to inconsistent effect across the three independent instances we performed the screen. Therefore, lack of robust effect should be interpreted through the lens of such limitations of large-scale screens, rather than definitive demonstration of a lack of function of such ARFome members.

Of those ARFome members that exhibited notable phenotypes, our screen identified the Class I ARFs, the GEF PSD, and the effector RAB11FIP4 as repressors of Spindle- and Spread-type collective invasion. Particularly, we identify that loss of ARF3, not ARF1, in 3D culture phenocopies PSD and RAB11FIP4 loss of function. That depletion of Class I ARFs induced invasive activity was somewhat unexpected, as numerous studies, particularly in breast cancer cells, report a pro-invasive and pro-tumorigenic function of ARF1 (Boulay et al., 2008; Boulay et al., 2011; Haines et al., 2015; Lewis-Saravalli et al., 2013; Schlienger et al., 2014; Schlienger et al., 2016; Schlienger et al., 2015; Wang et al., 2020; Xie et al., 2016).

In our studies, we define that ARF3 has a function distinct from that of ARF1. Notably, these 3D phenotypes, which require multicellular collective function, were not recapitulated when looking at the shape of single cells in 2D, which may explain

some of the differences to observations using 2D culture. Moreover, ARF3 expression was strongly induced in 3D culture, suggesting a requirement for collective function. That we identify such co-acting modules using a functional morphogenesis perturbation approach is notable as this would be difficult to predict from studies of single cells in the literature. Of ARFome expression in PC3 cells, PSD was the lowest expressed GEF, while RAB11FIP4 sits at approximately the mid-point of the ARFome interactors screened, which makes selection of these candidates non-obvious. PSD is also known as EFA6-A, which represents its prior consideration as a GEF for mostly ARF6. This is partially due to a low exchange activity of PSD from studies in solution (Hanai et al., 2016; Padovani et al., 2014). However, PSD is a potent GEF for ARF1 when both are present at membranes (Padovani et al., 2014). Indeed, in our studies, PSD was required for efficient GTP loading ARF3, but not ARF6. This underscores the power of our morphogenesis-based approach to identify co-acting molecules, rather than based solely on in vitro biochemical approaches.

We identified that RAB11FIP4 is a key ARF3 effector, and that N-cadherin is a key cargo protein of ARF3-RAB11FIP4 complex on RAB11 recycling endosomes. RAB11FIP4, also known as Arfophilin-2, is a dual Rab11- and ARF-binding protein that controls the organization of, and trafficking through, the Rab11 recycling endosome (Hickson et al., 2003; Wallace et al., 2002a; Wallace et al., 2002b). Indeed, ARF1 and ARF3 have been reported to control recycling endosome function independent to effects on the Golgi apparatus (Kondo et al., 2012). Notably, ARF3 was not required for RAB11FIP4 localization or distribution, concomitant with RAB11 being the major regulator of RAB11FIP4 endosomal recruitment (Hickson et al., 2003; Wallace et al., 2002a; Wallace et al., 2002b). Rather, ARF3 depletion increased N-cadherin retention in RAB11-RAB11-FIP4 endosomes. Similarly, KD of RAB11FIP3 in neuronal cells causes the

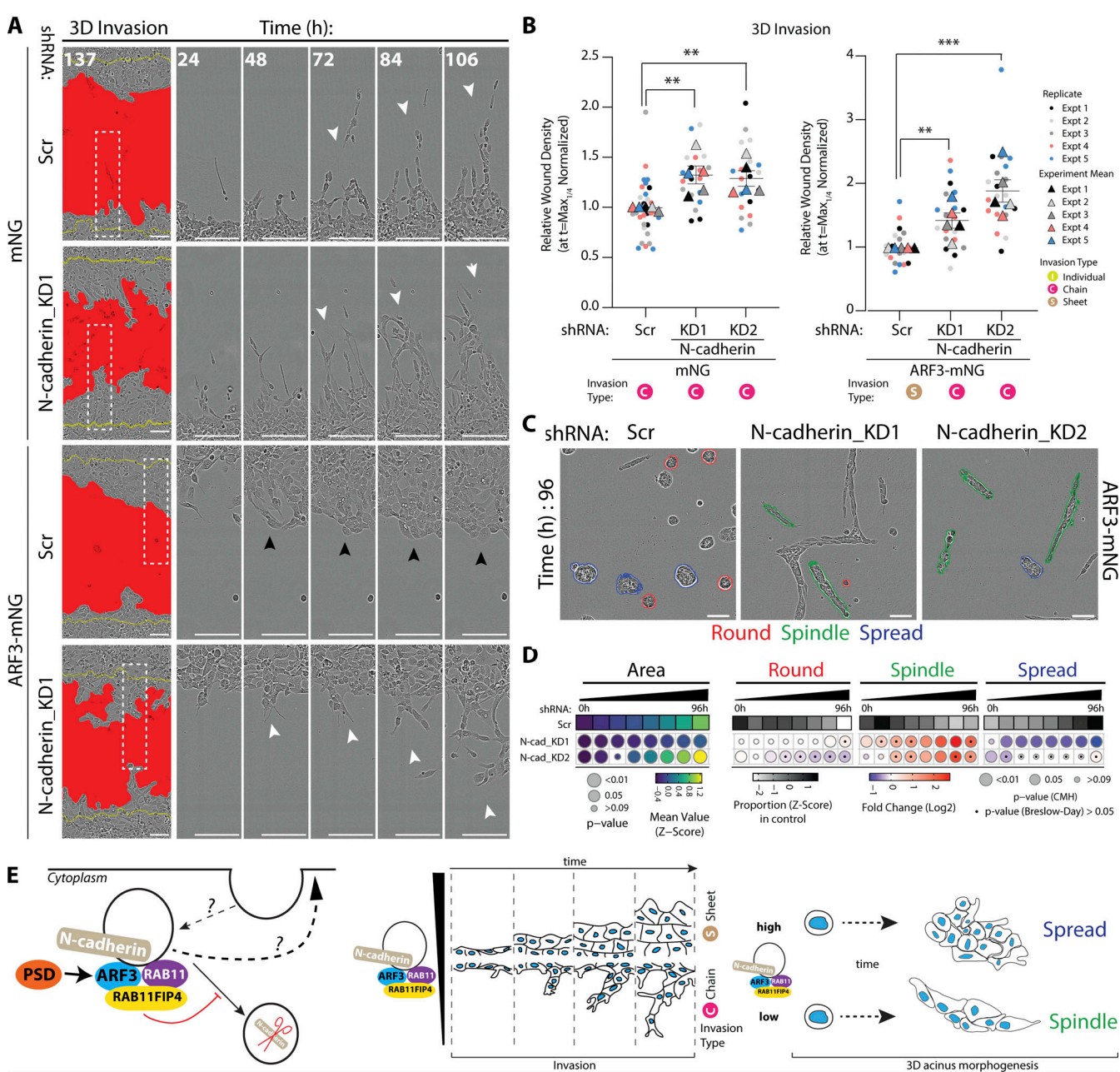

Figure 7. **N-cadherin is a key interactor of ARF3 that controls morphogenesis. (A and B)** PC3 cells in 3D invasion assay. Yellow lines, initial wound, and red pseudo color, wound at $t = \text{Max}_{1/4}$. Scale bars, 100 μm. Magnified images of boxed regions shown. White and black arrowheads, invasive chain or sheet, respectively. RWD at $t = \text{Max}_{1/4}$, normalized to Scr is shown in graphs (B). Data is mean ± SEM (5 experimental replicates, triangles, 3–8 technical replicates, circles. P values (Student's two-tailed $t$ test), **$P \leq 0.01$ and ***$P \leq 0.001$. **(C)** Phase images of PC3 acini expressing ARF3-mNG and Scr or N-cadherin shRNA. Outlines: Round (red), Spindle (green), and Spread (blue). Scale bar, 100 μm. $n$ = 3 experimental replicates each with 3 technical replicates/condition. 4,039 (Scr), 4,814 (N-cadherin KD1), and 3,454 (N-cadherin KD2) mNG acini quantified in total. **(D)** Quantitation of C. Heatmaps, Area is mean of Z-score normalized values (purple to yellow). P values, Student's $t$ test, Bonferroni adjustment, represented by size of bubble. Heatmaps, Round, Spindle, or Spread is $\text{Log}_2$ fold change from control (Scr; blue to red). Proportion of control at each time is Z-score normalized (white to black). P values, CMH test, Bonferroni adjusted, represented by size of bubble. Dot indicates P value (Breslow–Day test, Bonferroni-adjusted) for consistent effect magnitude. **(E)** Model, ARF3-RAB11FIP4 complex interacts with and regulates levels of N-cadherin in intracellular vesicles to control different modes of invasion.

intracellular retention of N-cadherin, which can only be rescued by expression of RAB11FIP3 with functional ARF-binding capacity (Hara et al., 2016). The related protein RAB11FIP1 regulates recycling of endocytosed N-cadherin to promote cell migration (Lindsay and McCaffrey, 2016). Therefore, the endosomal traffic of N-cadherin appears to be a major target of PSD-

ARF3-RAB11FIP3/4 in our system. It is unexpected to identify RAB11FIP3/4 proteins as phenocopying ARF3 function, as previous reports suggest Arfophilins to be Class II or Class III ARF effectors, with only modest binding of Class I ARFs (Shiba et al., 2006; Shin et al., 1999). Given the widespread overlap in binding capabilities within the ARFome, this emphasizes the need for

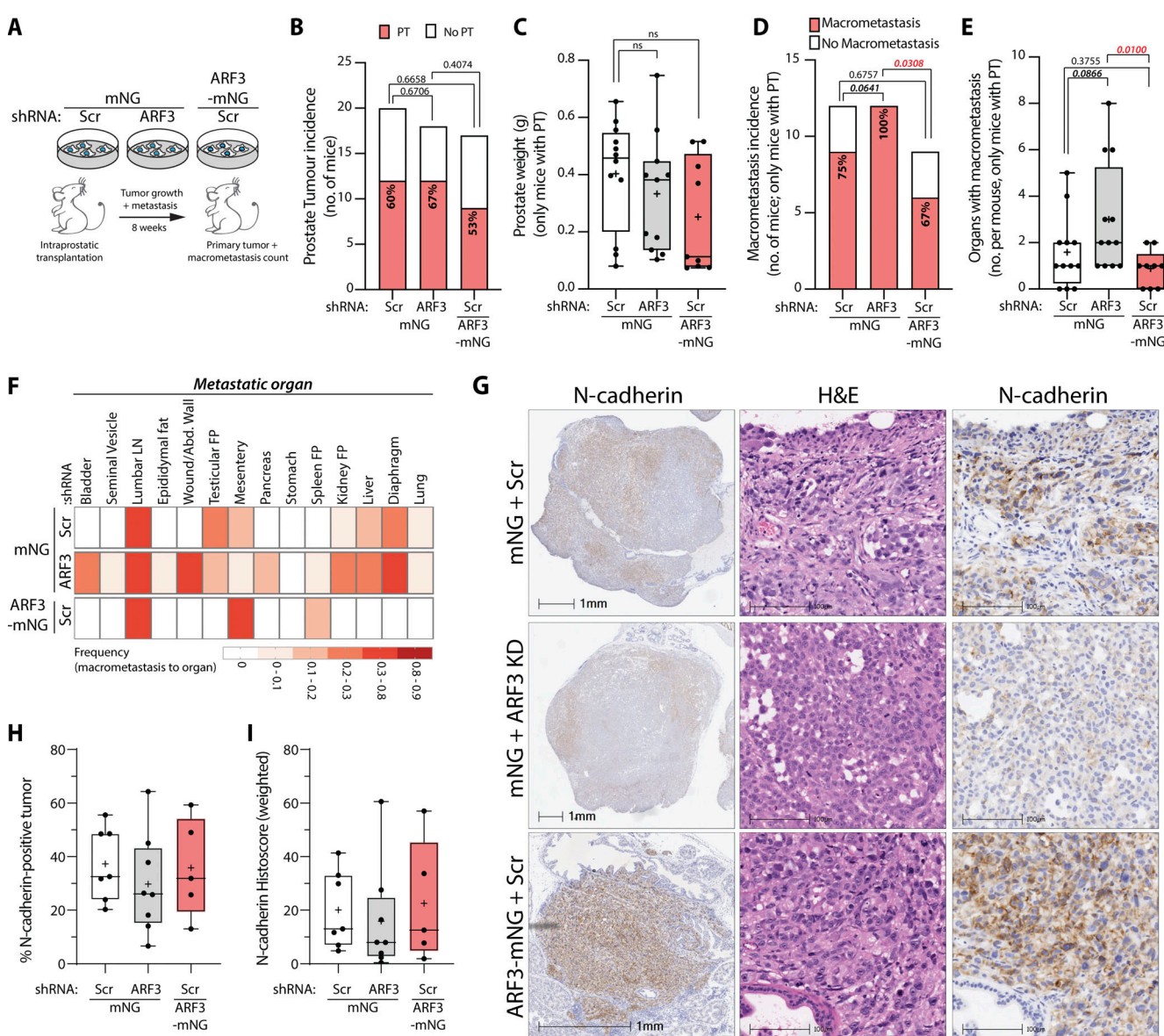

Figure 8. **ARF3 regulates metastasis in vivo. (A)** Schema, intraprostatic transplantation of PC3 cells expressing mNG or ARF3-mNG and Scr or *ARF3* shRNA into CD-1 nude male mice. PT formation and incidence and location of MM determined after 8 wk. **(B)** PT incidence in mice (total number) transplanted with PC3 cells expressing mNG, Scr shRNA (control, 20 mice), mNG, *ARF3* shRNA (18 mice), or ARF3-mNG, Scr shRNA (17 mice). P values (Chi-squared test), indicated. **(C)** Prostate weight in mice (with PT only) transplanted with PC3 cells expressing mNG, Scr shRNA (control, 12 mice), mNG, *ARF3* shRNA (12 mice, weight of 1 mouse prostate not recorded), or ARF3-mNG, Scr shRNA (9 mice). Box and whiskers plot, min–max percentile; +, mean; dots, outliers; midline, median; boundaries, quartiles. P values (Student's two-tailed *t* test). **(D)** MM incidence in mice (with PT only) transplanted with PC3 cells expressing mNG, Scr shRNA (control, 12 mice), mNG, *ARF3* shRNA (12 mice), or ARF3-mNG, Scr shRNA (9 mice). P values (Chi-squared test), indicated. **(E)** MM count/mouse in mice (with PTs only) transplanted with PC3 cells expressing mNG, Scr shRNA (control, 12 mice), mNG, *ARF3* shRNA (12 mice), or ARF3-mNG, Scr shRNA (9 mice). Box and whiskers plot, min–max percentile; +, mean; dots, outliers; midline, median; boundaries, quartiles. P values (Mann–Whitney test [two-tailed]), indicated on graph. **(F)** MM frequency to indicated organs for mice (with PT only) transplanted with PC3 cells expressing mNG, Scr shRNA (control, 12 mice), mNG, *ARF3* shRNA (12 mice), or ARF3-mNG, Scr shRNA (9 mice). FP, fat pad. **(G)** Serial sections representative of a primary tumor from a mouse transplanted with PC3 cells expressing mNG, Scr shRNA, mNG, *ARF3* shRNA, or ARF3-mNG, Scr shRNA, stained for H&E and N-cadherin (left panels and higher magnification). Scale bars, 1 mm or 100 µm. **(H and I)** Percentage of N-cadherin–positive PT (H) or weighted Histoscore for N-cadherin in PTs (I) in 8 mNG Scr shRNA, 7 mNG *ARF3* shRNA, and 5 ARF3-mNG, Scr shRNA mice. Box and whiskers plot, 10–90 percentile; +, mean; dots, outliers; midline, median; boundaries, quartiles.

phenotypic screening to identify co-acting modules within the ARFome.

In our system, ARF3 interacted with N-cadherin and acted as a rheostat to control turnover of N-cadherin from the cell surface without affecting E-cadherin levels. Reciprocally, decreased N-cadherin levels also reduced ARF3 levels, suggesting a mutual regulation of the complex. Notably, in cardiomyocytes ARF3 interacts with N-cadherin but not E-cadherin, while conversely ARF4 interacts with E-cadherin but not N-cadherin (Li et al., 2019), identifying specificity of different ARFs for different cadherins. The regulation of recycling of N-cadherin by ARF3-RAB11FIP4 may be a mechanism by which the latter complex

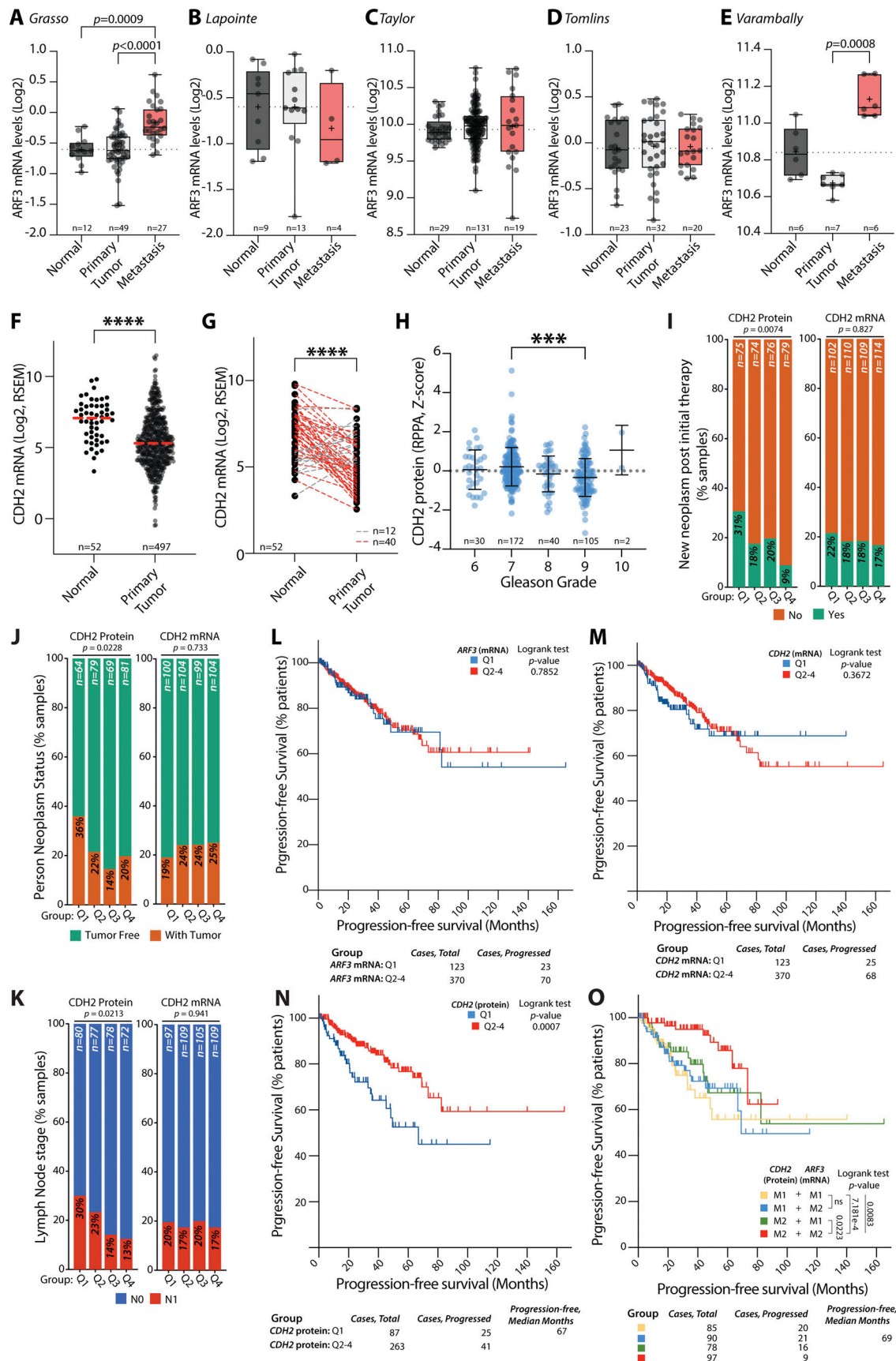

Figure 9. **N-cadherin and ARF3 expression identify poor-outcome prostate cancer patients. (A–E)** *ARF3* mRNA expression in normal prostate, PT, and metastasis samples from prostate cancer patients. Patient numbers, on graph. Glinsky (Glinsky et al., 2004), Grasso (GSE35988), Lapointe (GSE3933), Taylor

(GSE21032), TCGA, Tomlins (GSE6099), Varambally (GSE3325). P values, Kruskal–Wallis test with Dunn's multiple comparison test. **(F and G)** *CDH2* mRNA (Log$_2$, RNA-Seq by Expectation-Maximization [RSEM]) from Normal vs. Primary Tumor prostate samples (dataset, TCGA). Patient numbers, on graph. (F), all samples; (G), matched samples from same patients. Lines, directionality of change in normal compared to tumor; red, elevated in tumor; gray, elevated in normal. P values, two-tailed unpaired *t* test with Welch's correction, ****P ≤ 0.0001. **(H)** CDH2/N-cadherin protein from prostate tumor samples with different Gleason Grade scores (dataset, TCGA). Patient numbers, on graph. P value, one-way ANOVA with Tukey corrections for multiple testing and ***P ≤ 0.001. **(I–K)** Clinical parameters upon grouping of patients based on quartiles of expression (Q1–4) for N-cadherin protein or *CDH2* mRNA in prostate tumor samples. **(I)** New neoplasm post initial therapy. **(J)** Neoplasm status. **(K)** Lymph node stage. Dataset, TCGA. Patient numbers, on graph. Data presented as percentage of samples in each quartile grouping in presented categories. P values, chi-squared test. **(L–N)** Progression-free survival of prostate cancer patient groups based on quartiles (Q1 versus Q2–4 combined) of expression in tumor samples of (L) *ARF3* mRNA, (M) *CDH2* mRNA, or (N) CDH2/N-cadherin protein. P values, log rank test. Dataset, TCGA. Patient numbers, on graph. **(O)** Progression-free survival of prostate cancer patient groups based on median split (M1, low versus M2, high) of CDH2/N-cadherin protein and *ARF3* mRNA expression in prostate tumors. P values, log rank test. Dataset, TCGA. Patient numbers, on graph.

acts as a rheostat to control levels of the former. How N-cadherin in turn controls ARF3 levels is unclear.

The functional consequence of altering ARF3 was to control the modality of invasion in cells through influencing junctional F-actin levels and N-cadherin–dependent collective sheet-type movement. This activity, unique to ARF3 compared to ARF1, required three residues in the ARF3 C-terminus (A174/K178/K180). How these residues uniquely couple ARF3 to N-cadherin function is the focus of future work.

Although conceptually it is understandable that increased levels of a cadherin could increase junctional stability and induce sheet-type movement, that this was conferred by N-cadherin was unexpected. Conversely to our findings, N-cadherin overexpression has been reported as a targetable and prominent poor prognostic indicator of prostate cancer outcome and metastasis (Tanaka et al., 2010). Some of this discrepancy may be due to experimental system; some of these studies involve ectopic overexpression of N-cadherin in E-cadherin–negative cells. Notably, in our studies both N-cadherin and E-cadherin are expressed in PC3 cells and alteration of N-cadherin levels occurred without changing E-cadherin levels, indicating that this was not a transcriptional change between cadherin types (i.e., a "cadherin switch"). Effects on N-cadherin on collective behaviors are likely to differ greatly if other Type-I classical cadherins are not present.

The notion that E-cadherin without N-cadherin expression is anti-invasive or non-metastatic has been challenged, as E-cadherin is an active participant and requisite for metastasis in a number of model systems (Padmanaban et al., 2019; Shamir et al., 2014). In our system, reduced N-cadherin and unaltered E-cadherin was associated with widespread metastasis, suggesting that these cells may follow a similar E-cadherin–led metastatic pathway. To this end, examination of prostate cancer patients indicated that, similarly to our findings, low protein levels of N-cadherin in prostate tumors were associated with metastatic tumors with poor progression-free survival. Notably, when N-cadherin levels were high in patients, combining this with *ARF3* levels further stratified patients, wherein *ARF3*[HI]-N-cad[HI] patients presented with the best clinical outcome. Moreover, the ability of N-cadherin to stratify survival was specific to protein, not mRNA, levels, indicating that the trafficking of N-cadherin, such as by the ARF3-RAB11FIP4 rheostat identified here, is essential for N-cadherin influence on tumorigenesis.

Our studies indicate that PSD-ARF3-RAB11FIP3/4 function to control the modality of invasion in vitro. By using a murine model of intraprostatic xenograft of ARF3-manipulated cells, we could examine which modality was required for metastasis in vivo. While both ARF3 depletion and overexpression conditions still grew tumors and metastasized, the type and frequency of metastasis was dramatically altered. ARF3 depletion resulted in completely penetrant, widespread metastasis, while ARF3-overexpressing cells manifested low efficiency, local metastases. This was mirrored in analysis of prostate cancer patients who possessed metastatic disease when the key ARF3-RAB11FIP4 cargo N-cadherin protein levels were low. In breast cancers, ARF3 upregulation was reported as associated with pro-proliferative functions (Huang et al., 2019), while in gastric cancer, similar to our findings in prostate cells, ARF3 suppressed proliferative function and in vivo tumorigenesis (Liu et al., 2021). Indeed, in our own analysis while ARF3 expression was commonly altered in tumor compared to normal tissue, the directionality of expression change was dependent on tumor type. Given that control of N-cadherin levels was a major function of ARF3, this seeming incongruent effect of ARF3 may be due to a differential effect of N-cadherin in alternate tissue types. Therefore, rather than being a generalized good or bad indicator of tumorigenesis, the ARF3–N-cadherin rheostat may be a contextual regulator of tumorigenesis in different tumor types by controlling the homogeneity of N-cadherin turnover across the tumor. Collectively, this indicates that our screening approach to identify co-acting ARFome modules using 3D live-imaging approaches identifies an unexpected function for the Class I ARF, ARF3, in regulation of N-cadherin influence on collective invasion dynamics during metastasis.

## Materials and methods

### Cell culture
HEK293-FT cells (Thermo Fisher Scientific) were cultured in DMEM with 10% FBS, 2 mM *L*-glutamine, and 0.1 mM Non-Essential Amino Acids (all Gibco). PC3 cells (ATCC) were maintained in Ham's F-12K (Kaighn's) Medium (Gibco) supplemented with 10% FBS. RWPE-1, RWPE-2, WPE-NB14, and CA-HPV-10 cell lines (ATCC) were grown in Keratinocyte Serum Free Medium supplemented with 50 μg/ml Bovine Pituitary Extract and 5 ng/ml EGF (all Gibco). LNCaP and VCaP cells (ATCC) were cultured in RPMI-1640 or DMEM, respectively (Gibco), containing 10% FBS and 2 mM *L*-glutamine. DU145 cells (ATCC) were maintained in Minimum Essential Medium (MEM; Gibco) with 10% FBS and 2 mM *L*-glutamine. 22Rv1 cells (ATCC)

were grown in phenol free RPMI-1640 containing 10% charcoal stripped FBS and 2 mM *L*-glutamine (all Gibco). Cells were routinely checked for mycoplasma contamination (all negative) and HEK293-FT, PC3, RWPE-1, RWPE-2, LNCaP, and DU145 cells were authenticated using short tandem repeat profiling.

## ARFome shRNA screen
### Generation of shRNA
Oligonucleotides for shRNAs were synthesized for 210 ARFs, GEFs, GAPs, and effectors (Thermo Fisher Scientific) based on short hairpin sequences available from the RNAi Consortium (Broad Institute). Sequences and validation data are shown in Table S1. We selected one or two shRNAs per target if validation data was available or five shRNAs per target if unvalidated. shRNA oligonucleotides were cloned into a modified lentiviral vector, pLKO.4-mem:Venus. This was constructed from pLKO.1-puro and the puromycin-resistance cassette replaced with an mVenus fluorescent protein with an N-terminal membrane-targeting domain of GAP43 (derived from approach used in Beier et al., 2011), and was also modified to contain a single XhoI site. This allowed for screening of successful ligation of shRNA cassettes into the vector, which introduced a second XhoI site in the form of the shRNA loop (5′-CTCGAG-3′). The library generated was therefore improved by allowing for XhoI restriction digest screening of shRNA-containing sequences prior to plasmid sequencing (University of York). The resultant plasmid DNA was mini-prepped and plated into 96-well shRNA source plates at 100 ng/µl for use in transient transfections.

### Lentiviral infection of PC3
HEK293-FT cells were washed in PBS then dissociated by addition of 0.25% Trypsin in PBS/EDTA. Detached cells were collected in medium, cell number determined, and 13,000 cells added per well of 96-well plates. Plates were then incubated at 37°C, 5% $CO_2$ overnight. Lentiviral packaging vectors psPAX2 (plasmid 12260; Addgene; 100 ng/well) and VSVG (pMD2.G; plasmid 12259; Addgene; 10 ng/well) were diluted in OPTI-MEM (10 µl/well; Thermo Fisher Scientific) and dispensed into 96-well plates. Lipofectamine 2000 (Thermo Fisher Scientific) was also diluted in OPTI-MEM (0.6 and 10 µl per well) and dispensed into 96-well plates. After 5 min at room temperature the contents of these plates were combined and 1.25 µl of each shRNA added (final concentration of 125 ng/well). Plates were mixed briefly using a shaker, incubated at room temperature for 5 min and then added to HEK293-FT cells overnight at 37°C, 5% $CO_2$. Supernatant was removed and replaced with HEK293-FT medium containing an additional 10% FBS and plates incubated for a further 24 h at 37°C, 5% $CO_2$. Viral supernatant was then removed and replaced again after 24 h. The viral supernatant collected after 24 and 48 h was combined in 96-well plates and centrifuged at 300 *g* for 4 min. 150 µl of supernatant was decanted from each well and added to 96-well plates containing 50 µl Lenti-X Concentrator (Clontech) at 4°C for 1 h. Plates were then centrifuged at 1,100 *g* for 45 min at 4°C and the resultant pellets re-suspended in PC3 medium and added to PC3 cells plated in 96-well plates (at 6,000 cells/well) 24 h prior to infection. Additional PC3 medium (50 µl/well) was added after

24 h and the plates incubated for a further 48 h at 37°C, 5% $CO_2$. PC3 cells were then washed in PBS and 50 µl of 0.25% Trypsin in PBS/EDTA added slowly to each well. This was removed immediately and replaced with 5 µl of Trypsin per well for 7 min with gentle shaking. A further 5 µl of Trypsin was added to the center of each well for 1 min, without shaking, to help break up any cell clumps. PC3 medium was added, and the cells split into 4 96-well tissue culture–treated plates (655090; Greiner) which were incubated at 37°C, 5% $CO_2$ for use in assays.

### Live imaging of PC3 in 3D
A visual inspection was carried out after 24 h to identify plates with consistent confluence across wells. Cells were washed in PBS, trypsinized as described above and resuspended in PC3 medium. Typically, one-half to two-thirds of cells from each well were used for subsequent 3D cultures depending upon the initial confluence and effectiveness of trypsinization. Medium was supplemented with 2% Growth Factor Reduced Matrigel (GFRM; BD Biosciences) and added to 96-well ImageLock plates (Satorius) pre-coated with 10 µl of GFRM for 15 min at 37°C. For live imaging, ImageLock Plates were incubated at 37°C for 4 h, then imaged using an IncuCyte ZOOM (Satorius) with IncuCyte ZOOM Live Cell Analysis System Software 2018A. Phase and GFP images were taken every hour for 4 d at two positions per well with ×10 objective lens.

### Analysis
CellProfiler (Version 3.1.8) was used to design a pipeline to process the phase images acquired on the IncuCyte, including retention of only mem:Venus-positive objects. This pipeline identified and tracked acini at each time point and generated a database containing size, shape, and movement measurements. CellProfiler Analyst (Version 2.2.0) was then used to apply iterative, user-supervised machine learning to spheroid measurements in the resulting database. Specifically, PC3 acini were classified into bins based on their morphology: Round, Spindle, or Spread. Machine learning to differentiate between these classes, using a maximum of 20 rules with Fast Gentle Boosting, was used until accuracy for each class was >90%, as assessed using a confusion matrix. Once generated, these classification rules were saved as a .txt file and imported into CellProfiler for classification without need for re-training of new datasets (described in Freckmann et al., 2022).

Custom pipelines in KNIME Data Analytics Platform (Version 3.3.1; adapted from those described in Freckmann et al., 2022) were then used to collate data from experimental and technical replicates, filter datasets, and apply tracking label corrections. For user-defined phenotypes, heatmaps show phenotype frequency over time in comparison to control samples. The relative proportion of the total acini in each sample is presented as Z-score normalized data in 12-h time intervals using ggpot2 R package. P values correspond to circle size. Statistical comparison was performed using the Cochran–Mantel–Haenszel (CMH) test, which takes into account experimental replicates and is only statistically significant when the effect was present across all replicates. We also used the Breslow-Day statistic to test whether the magnitude of effect was homogeneous across all

experimental replicates—a non-significant P value indicated homogeneity and is represented by a black dot in the heatmap. For this, we used the DescTools R package. In both statistical tests, a Bonferroni adjustment was applied to account for multiple testing. For detection of phenotype classes, shRNAs were grouped into seven groups based on a dendrogram from clustering the fold change over time for each shRNA in Round, Spindle, and Spread phenotypes compared to the control (Scr) value. The dendrogram was generated using hierarchical clustering of heatmap data by complete linkage of Euclidian distances between samples. Line plots are presented which show the mean ± SEM of Round, Spindle, and Spread phenotypes over time for each group and for Scr shRNA controls. The average proportion of acini classified as Round, Spindle, or Spread across all time points is also shown for each group.

The entire shRNA screen, all the way from virus production to phenotype quantitation, was performed three independent times. Each replicate consisted of 18 technical replicates (three per 96-well plate) of control (Scr) shRNA and one replicate of each of the 210 ARFome shRNAs. 170,674 Scr mem:Venus shRNA-expressing acini were quantified in total. Table S1 shows the total number of acini quantified for each mem:Venus shRNA.

**Generation of stable cell lines**
Short hairpin sequences in pLKO.1-puromycin lentiviral vector (Sigma-Aldrich) were used to generate stable KD. The specific shRNA sequences used are shown in Table S2. GFP-tagged ARFs were kind gifts from P. Melançon (University of Alberta, Edmonton, Canada) and alternate fluorescent tags, mutations, and chimeras were generated by sub-cloning. All RNAi-resistant variants and chimeras were made by mutagenesis or subcloning using fragment synthesis (GeneArt). Plasmids will be deposited with Addgene upon publication.

Stable cell lines were made by co-transfecting lentiviral packaging vectors VSVG (pMD2.G; plasmid 12259; Addgene) and psPAX2 (plasmid 12260; Addgene) with plasmid of interest, into HEK293-FT using Lipofectamine 2000 (Thermo Fisher Scientific) as per manufacturer's instructions. Viral supernatants were filtered using PES 0.45 μm syringe filters (Starlab) and concentrated using Lenti-X Concentrator (Clontech) as per the manufacturer's instructions. PC3 cells were transduced with lentivirus for at least 3 d prior to FACS sorting or selection with 300 μg/ml G418, 2.5 μg/ml puromycin (both Thermo Fisher Scientific) or 10 μg/ml blasticidin (InvivoGen). To allow direct comparison PC3 cells expressing mNG were used for shRNA expression and Scr shRNA was added to cells over-expressing mNG-tagged plasmids.

**Live 3D culture and analysis**
Stable cells lines were used to form acini with 2,250 cells plated per well in a 96-well ImageLock plate pre-coated with 10 μl of GFRM for 15 min at 37°C. Medium was supplemented with 2% GFRM. Plates were incubated at 37°C for 4 h, then imaged using an IncuCyte ZOOM or IncuCyte S3 (Satorius). Images were taken every hour for 4 d at two positions per well using a ×10 objective lens.

The CellProfiler pipeline and classification rules generated in CellProfiler Analyst previously described for the ARFome shRNA

screen were also used to analyze these experiments. A Custom pipeline in KNIME Data Analytics Platform (Version 3.3.1) was used to collate data from experimental and technical replicates, filter datasets, apply tracking label corrections, and overlay colored outlines onto phase images. This pipeline then generated a heatmap of mean features over time (i.e., area) with statistical comparison to control sample. Data are presented in heatmaps as Z-score normalized in 12-h time intervals using ggpot2 R package. P values correspond to circle size. Statistical comparison was performed by Student's *t* test, two-tailed, and a Bonferroni adjustment was applied to account for multiple testing. For user-defined phenotypes, heatmaps show phenotype frequency over time in comparison to control samples as described for ARFome shRNA screen. Number of experimental replicates ($n$), the number of technical replicates per experiment, and the number of acini quantified in total per condition are stated in figure legends.

**Fixed 3D culture and analysis**
PC3 acini were cultured in GFRM as described in "Live 3D culture and analysis" section and incubated for 3 d at 37°C, 5% CO$_2$. Acini were gently washed with PBS prior to addition of 4% paraformaldehyde for 15 min. Samples were blocked in PFS (0.7% fish skin gelatin/0.025% saponin/PBS) for 1 h then stained with primary antibodies overnight at 4°C with gentle agitation. After 3 × 5 min washes in PFS, secondary antibodies (all Thermo Fisher Scientific) were added for 45 with gentle agitation at room temperature. Acini were washed 3× in PBS for 5 min each and maintained in PBS at 4°C until imaging was carried out. Antibodies and fluorochromes used are described in Table S3.

All images were acquired at room temperature in PBS. Confocal images were taken using A1R microscope (Nikon) with ×40 oil objective, exported as TIFF files and processed in Fiji. Other images were taken using an Opera Phenix High Content Screening System (Perkin Elmer). 35 images and 15 planes were taken per well with a 20× objective for PC3 acini. Columbus Image Data Storage and Analysis System (PerkinElmer, version 2.9.1) was used to design a custom pipeline to measure F-actin intensity per acini. F-actin staining was used to detect individual acini in maximum projection images of all planes and any acini touching the image border were excluded from further analysis. Data is presented in box and whiskers plot as total F-actin intensity per acinus. The percentage of acini with visibly reduced F-actin intensity at junctions in maximum projection images is also shown. Values are mean ± SEM and P values and statistical test used are described in figure legends. Number of experimental replicates ($n$), number of technical replicates per experiment, and the number of acini quantified in total per condition are stated in figure legends.

**2D morphology assays**
Method described previously (Nacke et al., 2021), briefly cells were plated on 96-well plates for 48 h, fixed, and then stained, as described in "Fixed 3D culture and analysis," using Hoechst 34580, Alexa Fluor 568 Phalloidin, and HCS CellMask Deep Red Stain. Samples were maintained in PBS and imaged at room temperature using an Opera Phenix High-Content Screening

System (×10 or ×63 objective) and Columbus Image Data Storage and Analysis System (PerkinElmer, version 2.9.1) used to design a custom pipeline for analysis.

Cells were detected based on nucleus localization (Hoechst) and the shape of each cell defined by either F-actin or HCS CellMask Deep Red Stain staining. Any cells touching the image border were excluded from further analysis. Where appropriate, cells expressing mNG-tagged proteins were detected using fluorescence intensity properties and any cells not expressing protein of interest were filtered from further analysis. Morphology properties of each object were calculated to classify them into three different categories (Round, Spindle, and Spread) using machine learning following manual training. A custom pipeline was generated using KNIME Data Analytics Platform (Version 3.3.1) to collate data from independent and technical replicates, calculate the $\log_2$ fold change of each phenotype over control and to calculate statistical significance using one-way ANOVA. Number of experimental replicates ($n$), number of technical replicates per experiment, and total number of cells quantified per condition are stated in each figure legend.

## 2D immunofluorescence
PC3 cells were plated on 96-well plates for 48 h at 37°C, 5% $CO_2$, fixed in 4% paraformaldehyde for 15 min and blocked in PFS (0.7% fish skin gelatin/0.025% saponin/PBS) for 1 h. Cells were stained with primary antibodies overnight at 4°C with gentle agitation. After 3 × 5-min washes in PFS, either Alexa-Fluor secondary antibodies, HCS CellMask Deep Red Stain and Hoechst 34580 (all Thermo Fisher Scientific) were added for 45 with gentle agitation at room temperature. Cells were washed 3 times in PBS for 5 min each and maintained in PBS at 4°C until imaging at room temperature was carried out. Antibodies used described in Table S3.

Plates were imaged using an Opera Phenix High-Content Screening System (×63 objective) and Harmony High-Content Imaging and Analysis Software v4.9 (PerkinElmer) used to design custom pipelines for analysis. The total number and area of puncta positive for each antibody was detected per cell in the presence and absence of ARF3. Using mNeonGreen and Hoechst 34580 to identify sub-cellular regions cells were also segmented into specific sub-cellular regions e.g., juxtanuclear, cytoplasmic, and periphery (defined as 5 pixels from outer edge of cell) and the total number and area of puncta calculated per region per cell. The percentage of puncta area "overlap" between puncta positive for different antibodies was also calculated per cell per condition as a measure of co-localization. Values are presented as mean ± SEM and cell numbers, P values, and statistical tests used are described in figure legends. $n = 3$ independent experiments.

## 2D proliferation assays
1,000 PC3 cells were plated per well in a 96-well ImageLock plate then imaged using an IncuCyte ZOOM (10× objective, two images per well) each hour for 96 h. Confluence per well was then measured using the IncuCyte ZOOM analysis software. Mean confluence for each condition at each time point (from four technical replicates) was calculated and normalized to time 0 ($t = 0$) for three experimental replicates. Mean data ± SEM is presented in a line graph (PRISM 7, GraphPad) as confluence normalized to $t = 0$ at 12-h time points. P values and the statistical test used are described in figure legends.

## 3D proliferation assays
PC3 acini were set up, as described in "Live 3D culture and analysis" section, on four 96-well tissue culture–treated plates with 2,250 cells plated per well. Plates were maintained at 37°C, 5% $CO_2$ for 4, 24, 48, and 72 h. At these time points 100 µl CellTiter-Glow (Promega) was added to each well and plates placed on an orbital shaker at low speed for 5 min. Plates were then removed and incubated at room temperature for a further 20 min. Luminescence was measured using a Tecan SPARK Microplate Reader (Tecan). Mean luminescence for each condition at each time point (from three technical replicates) was calculated and normalized to time 0 (4 h) for four experimental replicates. Mean data ± SEM is presented in a line graph (PRISM 7, GraphPad) as luminescence fold change to $t = 0$ at 24-h time points. P values and the statistical test used are described in figure legends.

## Invasion and migration assays
Method described previously (Nacke et al., 2021); briefly, ImageLock plates were coated with 20 µl of 10% GFRM diluted in medium overnight at 37°C. 70,000 PC3 cells in 100 µl medium were plated in each well for 4 h at 37°C. The resultant monolayer was wounded using a wound making tool (Satorius), washed three times with medium, and overlaid with either 50 µl of 25% GFRM for invasion assays or 100 µl medium for migration assays. After incubation at 37°C for an hour 100 µl medium was added to each well of invasion assay and plates imaged every hour for 4–6 d using the IncuCyte ZOOM.

For each experimental replicate the average Relative Wound Density (RWD) of all control samples (Scr shRNA) was calculated and each technical replicate normalized to this value. Results are presented as RWD at the time point at which the average RWD of the control samples is 50% ($t = Max_{1/2}$) or 25% ($t = Max_{1/4}$). Mean values ± SEM (triangles) and technical replicates (circles) for each independent experiment are presented. The number of experimental replicate ($n$) and technical replicates per experiment is also stated in the appropriate figure legend. P values and the statistical test used are described in figure legends.

## Immunoblotting
$2 \times 10^5$ cells were plated in 6-well plates for 48 h. Plates were washed twice with ice cold PBS, then RIPA lysis buffer added for 15 min on ice (50 mM Tris-HCl, pH 7.4, 150 mM NaCl, 0.5 mM $MgCl_2$, 0.2 mM EGTA, and 1% Triton X-100 with cOmplete protease inhibitor cocktail and PhosSTOP tablets [Roche]). Cells were scraped and lysates clarified by centrifugation at 13,000 rpm for 15 min at 4°C. BCA Protein Assay kit (Pierce) was used to determine protein concentration as per manufacturer's instructions.

RWPE-1 or PC3 acini were cultured in 3D as described above by plating $2 \times 10^5$ cells supplemented with 2% GFRM into 6-well

plates pre-coated with 180 µl of GFRM for 30 min at 37°C. After 2–3 d, acini were lysed in RIPA as described for 2D samples, then slowly passaged 10× through a 25-27G needle. After centrifugation, 13,000 rpm at 4°C for 15 min, the supernatant was separated from the lower layer of GFRM and debris and used for SDS PAGE.

SDS-PAGE was performed using Bolt 4–12% Bis-Tris gels and buffers (Thermo Fisher Scientific, as per the manufacturer's instructions, typically 20 µg per sample) and proteins transferred to polyvinylidene difluoride membranes using the iBlot 2 transfer system (Thermo Fisher Scientific). Membranes were incubated for 1 h in Rockland blocking buffer (Rockland) and primary antibodies added overnight at 4°C. Antibodies used described in Table S3. After addition of appropriate secondary antibodies for 1 h, membranes were washed in 1× TBS with 0.1% Tween-20 three times and imaged using a ChemiDoc Imager (BioRad) or Odyssey Imaging System (LI-COR Biosciences). Bands were quantified using Image Lab 6.1 (BioRad) or Image Studio Software 6.0 (LI-COR Biosciences) and values normalized to corresponding GAPDH bands. The number of independent lysate preparations ($n$) is stated in the appropriate figure legend, and quantitation of fold change of protein expression is shown as mean ± SEM. P values and the statistical test used are described in figure legends. GAPDH was used as a loading control for each blot, and a representative blot for each sample set is shown where appropriate.

### Immunoprecipitation (IP)
1 mg of cell lysate was immunoprecipitated with 2 µg anti-ARF3 antibody (610784; BD) overnight at 4°C with gentle rotation. Either anti-mouse agarose or mouse agarose (IgG control; both Sigma-Aldrich) were added for 1 h at 4°C with rotation. Samples were washed three times in RIPA lysis buffer, separated by SDS–PAGE, and immunoblotted. $n$ = 3 experiments from independent lysate preparations and IPs.

### mNG Trap
1 mg of lysate from PC3 cells expressing mNG or ARF3-mNG were immunoprecipitated using mNG-Trap Agarose beads (ChromoTek) as per the manufacturer's instructions and immunoblotting performed as described in the "Immunoblotting" section. $n$ = 2 experiments from independent lysate preparations and pulldowns.

### Biotinylation assay
1 × $10^6$ PC3 cells were plated on 120-mm plates for 48 h at 37°C, 5% $CO_2$. Plates were washed 3 times with ice-cold Ham's F-12K Medium supplemented with 10% FBS and 1% 1 M Hepes (H/F-12K). 1.5 ml of H/F-12K containing 2.25 mg of Biotin (Thermo Fisher Scientific) was added to each plate and incubated with gentle agitation for 1 h at 4°C. Excess biotin was quenched by addition of 1.5 ml Quenching Buffer (50 mM $NH_4Cl$, 1 mM $MgCl_2$, 0.1 mM $CaCl_2$/PBS) for 1 min at 4°C. Plates were then washed five times in ice cold PBS and control (0 h) plates lysed in 500 µl RIPA lysis buffer. Remaining plates were incubated for 4 h at 37°C in pre-warmed H/F-12K, washed five times in ice cold PBS, and lysed in 500 µl RIPA lysis buffer. All samples were

nutated for 5 min at 4°C, clarified for 10 min at 14,000 rpm, and supernatant assayed to determine protein concentration. 300 µg of each supernatant was then added to Streptavidin-agarose beads (Merck), which had been washed 3× with ice-cold 10 mM Tris, pH 7 (30 µl beads + 70 µl 10 mM Tris per sample). Samples were immunoprecipitated overnight at 4°C with gentle rotation. Beads were then washed 3× in ice-cold RIPA lysis buffer and once in 10 mM Tris and immunoblotting performed as described in the "Immunoblotting" section. N-cadherin expression levels in pulldowns were quantified using Image Studio Lite v5.2 (LI-COR Biosciences) and normalized to N-cadherin/GAPDH levels in lysates. P values and the statistical test used are described in figure legends.

### ARF-GTP pulldown
Method described previously (Nacke et al., 2021); briefly, PC3 cells were incubated on 120-mm plates for 48 h at 37°C, 5% $CO_2$. Cells were then lysed on ice in pulldown-lysis buffer (50 mM Tris, 100 mM NaCl, 2 mM $MgCl_2$, 0.1% SDS, 0.5% Na-deoxycholate, 1% Triton X-100, 10% glycerol). Lysates were syringed 5× using a 25-27G needle and centrifuged at 4°C 14,000 $g$ for 1 min. Spin columns were equilibrated with 50 µl of Glutathione Agarose resin and washed with pulldown-column wash buffer (1:1 pulldown-lysis buffer and 1×TBS). 80 µg of GST-GGA3-GAT recombinant fusion protein was immobilized on the agarose resin by incubation at 4°C with gentle rocking. After 1 h, 1 mg of each lysate was added onto spin columns and incubated again at 4°C for 2 h with rocking. Pulldown wash buffer (50 mM Tris, 100 mM NaCl, 2 mM $MgCl_2$, 1% NP-40, 10% glycerol) was used to wash unbound proteins off the column. 60 µl of pulldown-elution buffer (10 mM Glutathione in 1×TBS) was added to each spin column and incubated for 5 min at room temperature. Eluted protein was collected at 1,250 g for 1 min and samples prepared for SDS-PAGE and immunoblotting as described above. $n$ = 3 experimental replicates for ARF3 Chimeras and $n$ = 5 for PSD KD. Data is presented as GGA3 binding normalized to ARF3 or ARF6 levels for each experiment. Values are mean ± SEM.

### Flow cytometry
To detect surface N-cadherin protein localization, PC3 cells were washed twice in ice-cold PBS, detached using 2 mM EDTA in PE for 10 min and resuspended in medium. 2 × $10^5$ cells per sample were incubated for 30 min at 4°C with 1 µl anti-N-cadherin (13116; CST) primary antibody in PC3 medium. Samples were washed with PBS and stained with a secondary antibody (1:200, Alexa Fluor 647; Thermo Fisher Scientific) for 30 min at 4°C. After washing in PBS, samples were processed with a BD FOR-TESSA Z6102 (BD FACSDIVA software, v8.0.1), then analyzed with FlowJo software (version 10.1r5) and GraphPad Prism 9.

### Quantitative RT-PCR (RT-qPCR)
RNA was isolated using a RNeasy kit (Qiagen) prior to reverse transcription from 1.5 µg RNA using High-Capacity cDNA Reverse Transcription kit (Applied Biosystems) as per the manufacturer's instructions. cDNA was diluted 1:1 with non-DEPC treated Nuclease-Free Water (Thermo Fisher Scientific), then

1 µl was used per qPCR reaction. qPCR was performed using 10 µl PowerUp SYBR Green Master Mix (Thermo Fisher Scientific) as per the manufacturer's instructions. Custom DNA oligos (IDT) were used at 2 µM per reaction. ARF3 primers for RT-qPCR as follows: 3′-CGACCCCTCTGGAGACACTA-5′ and 5′-TACTCGCTCCCGATCATTGC-3′, and N-cadherin primers used for RT-qPCR: 3′-GGAAAAGTGGCAAGTGGCAG-5′ and 5′-GGA GGGGATGACCCAGTCTCT-3′. $n = 3$ independent RNA samples isolated per biological condition with four technical replicates for each. Applied Biosystems Quant Studio 3 was used with comparative Ct (ΔΔCt) program with standard cycling including melt curve for primers as follows: 50°C 2 min, 95°C 10 min, ×40 cycles, 95°C 15 s, 61.5°C 1 min. Relative expression was calculated by normalizing genes of interest to GAPDH, then to Scr control.

### RNA sequencing (RNA-seq)
Briefly, RNA was extracted using an RNeasy kit, incorporating a DNase digestion step using RNase-Free DNase Set (both Qiagen), followed by reverse transcription using High-Capacity cDNA Reverse Transcription kit (Thermo Fisher Scientific), all per the manufacturer's instructions. Quality control of all RNA samples was performed using a 2200 Tapestation and High-sensitivity RNA screentape (Agilent) and only samples with RNA integrity number values >7.9 were processed. Libraries were then prepared using 1 µg of total RNA as per the manufacturer's instructions (Illumina TruSeq stranded mRNA). Libraries were sequenced using an Illumina NextSeq 500, on a High-Output 75 cycle run with paired-end 36 bp read length.

For RNA-seq analysis, quality checks and trimming on the raw RNA-seq data files were performed using FastQC version 0.11.8, FastP version 0.20 and FastQ Screen version 0.13.0. RNA-seq paired-end reads were aligned to the GRCm38 version of the human genome and annotated using HiSat2 version 2.1.0. Expression levels were determined and analyzed using a combination of HTSeq version 0.9.1, the R environment version 3.6.1, utilizing packages from the Bioconductor data analysis suite and differential gene expression analysis based on the negative binomial distribution using the DESeq2 package version 1.22.2. Pathway Analysis was performed using MetaCore from Clarivate Analytics (https://portal.genego.com/).

### Animal studies
Animal experiments were performed in compliance with all relevant ethical regulations and approvals of the relevant UK Home Office Project Licence (70/8645 and P5EE22AEE) and carried out with ethical approval from the Beatson Institute for Cancer Research and the University of Glasgow under the Animal (Scientific Procedures) Act 1986 and the EU directive 2010 and sanctioned by Local Ethical Review Process (University of Glasgow).

7-wk-old CD1-nude male mice were obtained from Charles River (UK) and acclimatized for at least 7 d. Mice were kept in a barriered facility at 19–22°C and 45–65% humidity in 12 h light/darkness cycles with access to food and water ad lib and environmental enrichment. $2 \times 10^6$ PC3 cells stably expressing mNG and either Scr shRNA (20 mice) or ARF3 KD1 shRNA (18 mice) or expressing ARF3-mNG and Scr shRNA (17 mice) were surgically implanted into one of the anterior prostate lobes of each mouse (under anesthesia and with analgesia). The mice were continually assessed for signs of tumor development (including by palpation) and humanely sacrificed at an 8-wk time point, prior to tumor burden becoming restrictive. Primary tumor (PT) and macrometastasis (MM) incidence were analyzed by gross observation and are presented as number of mice with PT or MM incidence only in mice with a PT. MM count per mouse and weight of prostate is also shown in box and whiskers plots only for mice with PTs (Scr shRNA, 12/20 mice, ARF3 KD1 shRNA, 12/18 mice, and ARF3-mNG and Scr shRNA, 9/17 mice). P values and the statistical test used are described in figure legends.

### Hematoxylin and eosin (H&E) and immunohistochemistry (IHC) staining
4 µm formalin fixed paraffin embedded sections which had been maintained at 60ºC for 2 h were used for H&E and IHC staining. A Leica Bond Rx autostainer was used to stain and de-wax (AR9222; Leica) the formalin fixed paraffin embedded sections sections. Epitope retrieval using ER2 solution (AR9640; Leica) was then carried out for 20 min at 95°C. Sections were washed with Leica wash buffer (AR9590; Leica) and blocked using an Intense R kit (DS9263; Leica). Sections were rinsed with wash buffer and incubated for 30 min with N-cadherin antibody (13116; CST). Sections were again rinsed with wash buffer and rabbit envision secondary antibody applied for 30 min. Wash buffer was used for a final wash prior to visualization using DAB and counterstaining with Hematoxylin from Intense R kit.

A Leica autostainer (ST5020) was used for H&E staining. Sections were dewaxed, graded alcohols applied, and then stained for 13 min with Haem Z (RBA-4201-00A; CellPath). Sections were washed in water, differentiated in 1% acid alcohol, washed again, and the nuclei blu'd in Scotts tap water substitute. Sections were placed in Putt's Eosin for 3 min after a wash in water. Sections were washed in water, dehydrated via graded ethanol, and covered in xylene. The sections were then mounted on a coverslip using DPX mountant (CellPath), scanned at × 20 magnification using a Lecia Aperio AT2 slide scanner, and analyzed using Halo software (Indica Labs).

Halo software was used to quantify the area of PTs. Expression level of N-cadherin was scored using Histoscore (= Σ [1 × % area of weak positive staining] + [2 × % area of moderate positive staining] + [3 × % area of strong positive staining]). All statistical analyses and graphs were made using GraphPad Prism 9.

### Analysis of patient cohorts
Patient data i.e., copy number and RNA-seq data from Cancer Cell Line Encyclopedia (CCLE) was accessed, analyzed, and downloaded using in-platform http://cBioportal.org tools (Cerami et al., 2012; Gao et al., 2013). Copy number and mRNA levels (RNA-seq) for prostate normal and cancer cell lines were downloaded from CCLE (https://sites.broadinstitute.org/ccle/ [Ghandi et al., 2019]). Patient datasets for normal and tumor samples from TCGA (pan-cancer) were downloaded from the TCGA Splicing Variants database (http://TSVdb.com [Sun et al., 2018]). Data for the combination of normal and tumor were downloaded from the Gene Expression for Normal and Tumor

database (http://gent2.appex.kr/gent2/ [Park et al., 2019]), with sample numbers and study IDs in Tables S4 and S5. Datasets for Normal, Tumor, and Metastasis (Glinsky, [Glinsky et al., 2004], Grasso [GSE35988], Lapointe [GSE3933], Taylor [GSE21032], TCGA [obtained from http://cBioportal.org], Tomlins [GSE6099], and Varambally [GSE3325]) were downloaded from CANCER-TOOL (Cortazar et al., 2018) and analyzed using GraphPad Prism 9.

## Online supplemental material

Fig. S1 confirms that expression levels of ARF GTPases vary in different prostate cancer cell lines in 2D and 3D. Fig. S2 shows the effects of manipulation of Class 1 ARF expression on proliferation and phenotype of prostate cancer cells. Fig. S3 compares the effect of over-expression of ARF1 and ARF3 in migration and invasion assays. Fig. S4 confirms that N-cadherin depletion mimics ARF3 depletion in 2D and 3D. Fig. S5 examines N-cadherin and ARF3 expression in prostate cancer patients. Tables S1 and S2 list shRNA sequences used in ARFome screen and in addition to the screen, respectively. Table S3 summarizes antibodies used in this study. Tables S4 and S5 show normal and tumor dataset information for ARF3 and CDH2, respectively. Videos 1 and 2 show examples of PC3 acini formation (phase and mem:Venus, respectively). Videos 3, 4, and 5 provide examples of alternate movement modalities in wounded monolayers.

## Data availability

RNA-seq data have been deposited in the NCBI BioProject database with accession no.: PRJNA897704. Any other data that supports the findings of this study will be available from corresponding author.

## Acknowledgments

The LAMP2 monoclonal antibody (H4B4) developed by J.T. August and J.E.K. Hildreth was obtained from the Developmental Studies Hybridoma Bank, created by the National Institute of Child Health and Human Development of the National Institutes of Health and maintained at The University of Iowa, Department of Biology, Iowa City, IA, USA, and was a kind gift from D.G. McEwan. We would also like to thank K. Campbell, E. Shanks, Barbara Cadden, K. Rakovic, and J.Le Quesne for assistance/advice on immunohistochemistry, high-throughput screening, and pathology, respectively. Thanks to Core Services and Advanced Technologies at the Cancer Research UK (CRUK) Beatson Institute, including Beatson Advanced Imaging Resource, Molecular Technologies, Biological Services, and Histology Units.

This work was supported by the following grants: D.M. Bryant National Institutes of Health K99CA163535, CRUK (C596/A19481). E. Sandilands, A. Román-Fernández, and L. McGarry CRUK C596A17196 and A31287. E.C. Freckmann was supported by a University of Glasgow Industrial Partnership PhD scheme co-funded by Essen Bioscience, Sartorius Group. E.M. Cumming was supported by a University of Glasgow MVLS Doctoral Training Programme PhD Studentship. R. Patel and H.L. Leung funded by CRUK (A22904). S. Mason, J. Anand, L. Galbraith, and K. Blyth CRUK (A29799, A17196, and A31287).

Author contributions: D.M. Bryant conceived the project. Under supervision of D.M. Bryant, lab work was performed by E. Sandilands, E.C. Freckmann, E.M. Cumming, A. Román-Fernández, L. McGarry, C. Nixon, and J. Cartwright. E.C. Freckmann, E. Sandilands, and D.M. Bryant performed computational analysis. J. Anand, L. Galbraith, S. Mason, and R. Patel performed in vivo murine experiments under supervision of K. Blyth and H.Y. Leung. Manuscript was written by D.M. Bryant and E. Sandilands. All authors read and commented on manuscript.

Disclosures: E.C. Freckmann reported grants from Essen Bioscience during the conduct of the study. No other disclosures were reported.

Submitted: 23 June 2022

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

# Supplemental material

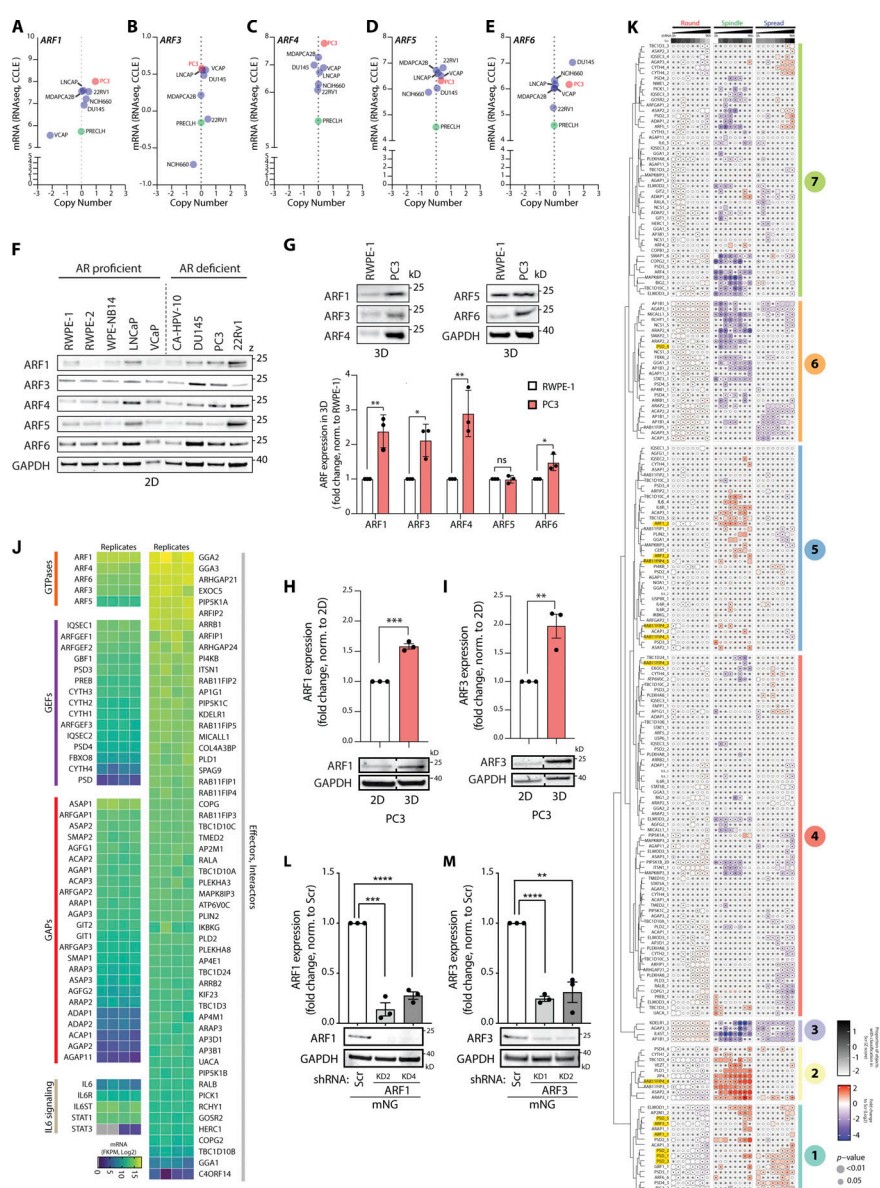

**Figure S1.** **Expression levels of ARF GTPases vary in different prostate cancer cell lines in both 2D and 3D. (A–E)** Graphs generated using RNA-seq data from the CCLE comparing (A) *ARF1*, (B) *ARF3*, (C) *ARF4*, (D) *ARF5*, and (E) *ARF6* gene copy number and mRNA expression levels in multiple prostate cancer and non-transformed cell lines. Metastatic PC3 and normal prostate PLECLH cell lines, red and green, respectively. **(F)** Western blot of androgen receptor (AR) proficient or deficient prostate cell lines for ARF1, ARF3, ARF4, ARF5, and ARF6. GAPDH is loading control for ARF6 and a sample control for all other blots. Panels shown are representative of 3 independent lysate preparations. **(G)** Western blot of RWPE-1 and PC3 acini, formed in GFRM (3D) for 2 d, for ARF1, ARF3, ARF4, ARF5, and ARF6. GAPDH is loading control for ARF4 and a sample control for all other blots. Panels shown are representative of 3 independent lysate preparations. Graph is fold change of ARF expression in PC3 cells, normalized to RWPE-1 cells. Data presented as mean ± SEM. Panels shown are representative of 3 independent lysate preparations. P values (Student's two-tailed $t$ test), *P ≤ 0.05 and **P ≤ 0.01. **(H and I)** Western blot of PC3 cells in 2D or in 3D for (H) ARF1 or (I) ARF3. GAPDH is loading control for each ARF blot. Graph, fold change of ARF expression, normalized to 2D samples. Dashed lines indicate blot was spliced. Data presented as mean ± SEM and panels shown are representative of 3 independent lysate preparations. P values (Student's two-tailed $t$ test), **P ≤ 0.01, ***P ≤ 0.001. **(J)** RNA-seq data from PC3 cells shows mRNA expression (Log$_2$) of genes encoding ARF GTPases, GEFs, GAPs, components of the IL6 signaling pathway, and ARF effectors and interactors (Log$_2$). $n$ = 4 mRNA samples prepared independently. **(K)** PC3 cells expressing ARFome shRNA were plated on ECM with 2% ECM overlay and multi-day high-throughput imaging carried out live in 3D. Mem:Venus-positive acini classified into Round, Spindle, or Spread phenotypes at each time point. Heatmap presents this classification, in 12-h time intervals, as a Log$_2$ fold change from control (Scr; blue to red). The proportion of control at each time point is Z-score normalized for each class (white to black). P values, CMH, Bonferroni adjusted, represented by the size of the bubble. Dot indicates P value (Breslow–Day test, Bonferroni-adjusted) for consistent effect magnitude. shRNAs grouped into seven groups (Phenotype Group 1–7) based on dendrogram generated using hierarchical clustering by complete linkage of Euclidian distances between samples. Viral infections and live 3D assays carried out 3 independent times. Each experimental replicate consisted of 18 technical replicates of Scr shRNA (170,674 acini in total) and 1 replicate of each of the 210 ARFome shRNAs (Table S1). **(L and M)** Western blot of PC3 cells expressing mNG and Scr, (L) *ARF1*, or (M) *ARF3* shRNA for ARF1, ARF3, and GAPDH as a loading control for each. Panels shown are representative of 3 independent lysate preparations. Graph is fold change of ARF expression normalized to Scr. Data is mean ± SEM. P values (Student's two-tailed $t$ test), **P ≤ 0.01, ***P ≤ 0.001, and ****P ≤ 0.0001. Source data are available for this figure: SourceData FS1.

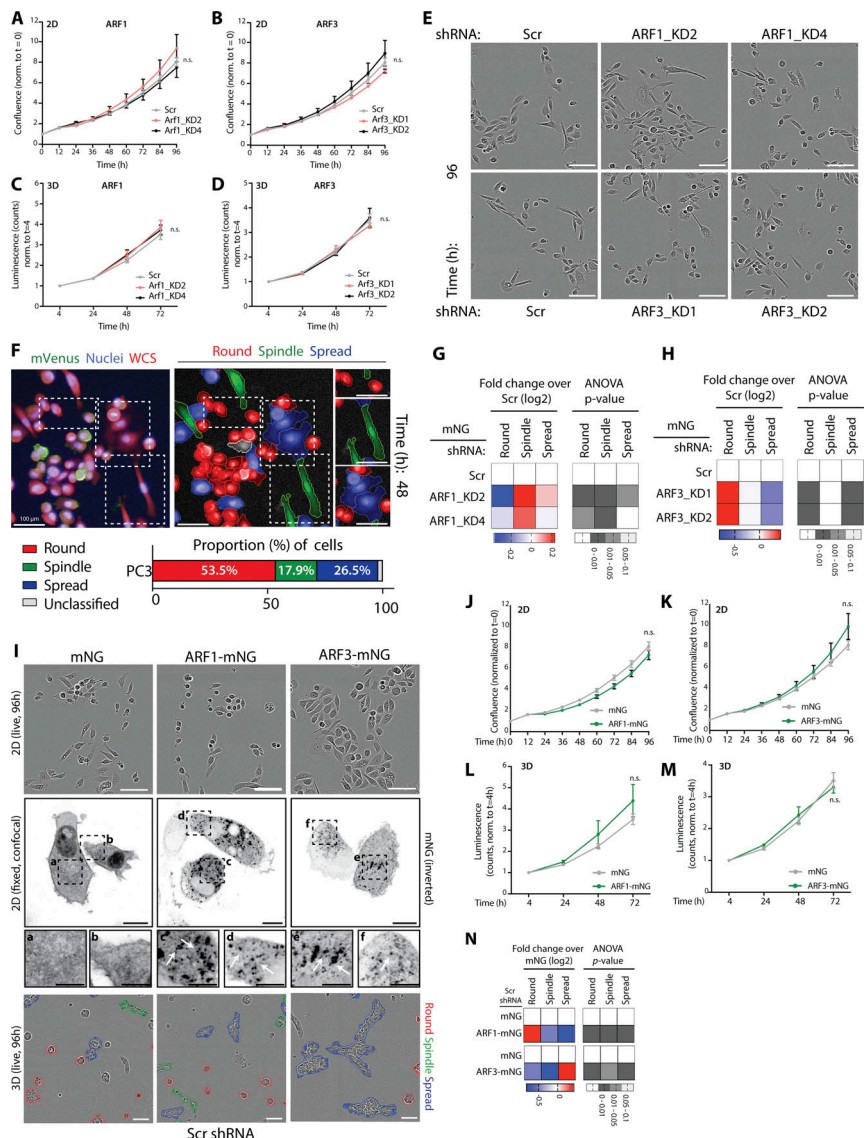

Figure S2. **Effect of depletion or over-expression of Class I ARF GTPases on 2D phenotype and proliferation of prostate cancer cells. (A and B)** PC3 cells expressing mNG and Scr, (A) *ARF*1, or (B) *ARF*3 shRNA were plated at low density and imaged. Data is mean confluence ± SEM, normalized to time 0. *n* = 3 experimental replicates with 4 technical replicates/condition. P values (one-way ANOVA). **(C and D)** PC3 acini expressing mNG and Scr, (C) *ARF*1, or (D) *ARF*3 shRNA were plated for 4–72 h. CellTiter-Glow was added and luminescence measured to assess ATP-based cell viability. Data is mean luminescence ± SEM, normalized to 4-h time point. *n* = 4 experimental replicates with 3 technical replicates/condition. P values (one-way ANOVA). **(E)** Phase images of PC3 cells described in A and B. Scale bars, 100 µm. Representative of *n* = 3 experimental replicates with 4 technical replicates/condition. **(F)** Schema, machine learning applied to classify and quantify 2D PC3 cells into three phenotypic categories. Upper panel, mem:Venus Scr shRNA (green), whole cell stain (WCS, red), and Hoechst (nuclei, blue). Lower panels, Round (blue), Spindle (green), and Spread (red). Scale bar, 100 µm. Mean proportion of PC3 cells, expressing Scr shRNA, with each phenotype shown for *n* = 3 experimental replicates each with 18 technical replicates. Total of 290,830 cells quantified. **(G and H)** PC3 cells in 2D expressing Scr and (G) *ARF*1 or (H) *ARF*3 shRNA were classified into Round, Spindle, and Spread. Heatmaps, Log₂ fold change over Scr. P values, one-way ANOVA, grayscale values as indicated. *n* = 4 and 2 experimental replicates with 3–4 technical replicates/condition for *ARF*1 and *ARF*3, respectively. (G) 15,392 (Scr), 26,486 (*ARF1*_KD2), 20,480 (*ARF1*_KD4), (H) 3,610 (Scr), 2,692 (*ARF3*_KD1), 2,882 (*ARF3*_KD2) cells were quantified in total. **(I)** Phase images of PC3 cells expressing mNG, ARF1-mNG or ARF3-mNG and Scr shRNA (upper panels). Scale bars, 100 µm. *n* = 3 experimental replicates with 4 technical replicates/condition. Confocal images (middle panels) show localization of mNG constructs (black, inverted images) in 2D cells. Scale bars, 20 µm. Magnified images of boxed regions shown (a–f). White arrows, ARF mNG in puncta. Scale bars, 10 µm. Images are representative of observations made in 3 experimental replicates. Also shown (lower panels) are phase images of PC3 acini, Round (red), Spindle (green), and Spread (blue). Scale bar, 100 µm. *n* = 6 and 4 experimental replicates for ARF1-mNG and ARF3-mNG, respectively, each with 2–4 technical replicates/condition. Quantitation shown in Fig. 4, E and F. **(J and K)** Confluence quantified in cells expressing mNG, (J) ARF1-mNG or (K) ARF3-mNG and Scr shRNA using phase images. Data is mean ± SEM, normalized to time 0. *n* = 3 experimental replicates with 4 technical replicates/condition. P values (one-way ANOVA). **(L and M)** PC3 acini expressing mNG, (L) ARF1-mNG or (M) ARF3-mNG and Scr shRNA were plated for 4–72 h. CellTiter-Glow was added and luminescence measured to assess ATP-based cell viability. Data is mean ± SEM, normalized to 4-h time point. *n* = 4 experimental replicates with 3 technical replicates/condition. P values (one-way ANOVA). **(N)** PC3 cells expressing mNG, ARF1-mNG, or ARF3-mNG and Scr shRNA were classified into Round, Spindle, and Spread. Heatmaps, Log₂ fold change over mNG. P values, one-way ANOVA, grayscale values as indicated. *n* = 3 experimental replicates with 3–4 technical replicates/condition. 4,309 (mNG), 4,766 (ARF1), and 5,261 (mNG), 6,508 (ARF3) mNG-positive cells quantified in total.

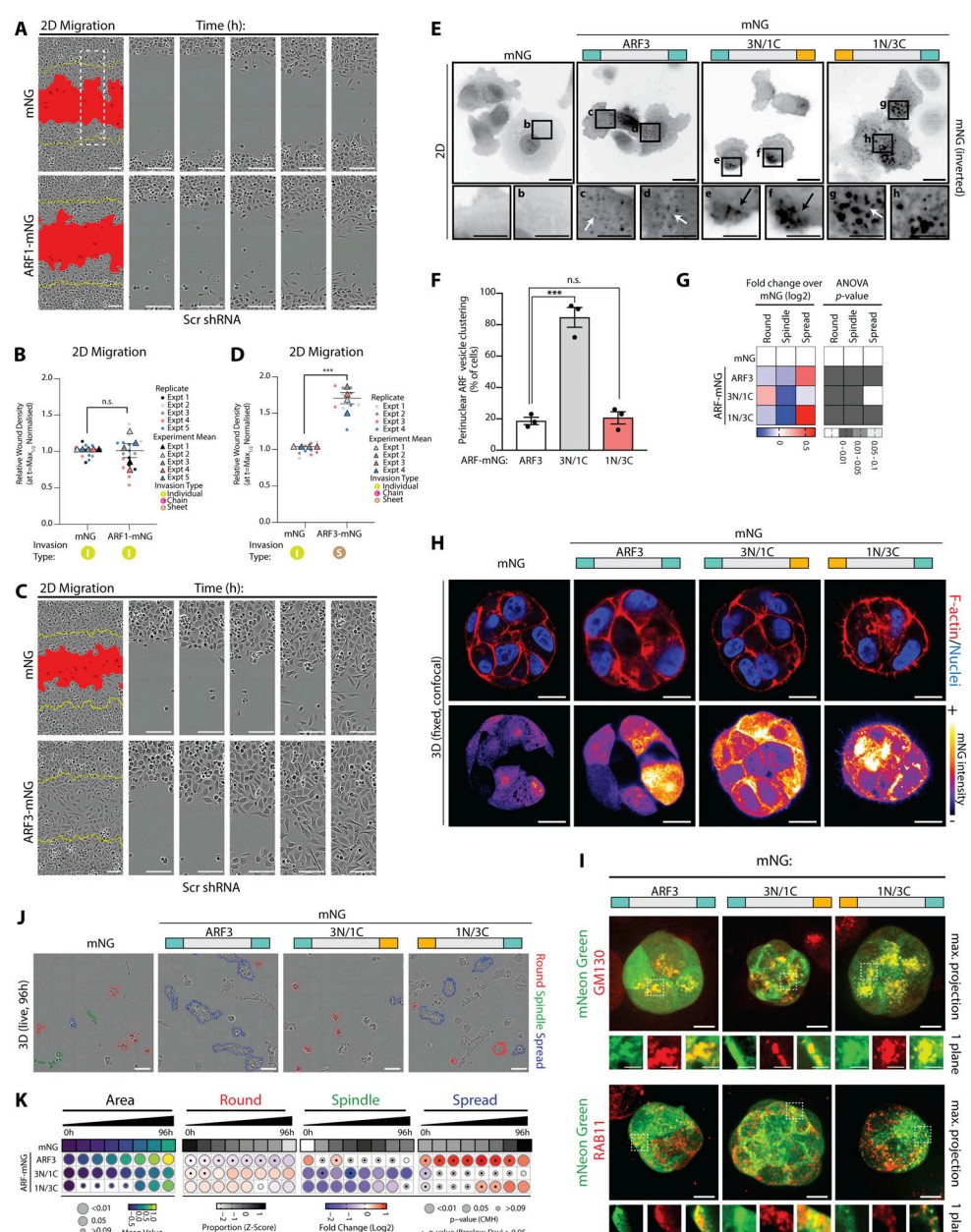

Figure S3. **ARF1 or ARF3 over-expression has different effects on migration and invasion of prostate cancer cells. (A–D)** Phase images of cells expressing mNG, (A) ARF1-mNG or (C) ARF3-mNG and Scr shRNA in 2D migration assay. Yellow lines, initial wound, and red pseudo color, wound at $t$ = Max$_{1/2}$. Scale bars, 100 μm. Magnified images of boxed regions shown. RWD at $t$ = Max$_{1/2}$, normalized to mNG is shown for (B) ARF1-mNG or (D) ARF3-mNG. Data is mean ± SEM (4–5 experimental replicates, triangles, 3–4 technical replicates, circles). P values (Student's two-tailed $t$ test), ***P ≤ 0.001. **(E)** Confocal images of PC3 cells expressing mNG, ARF3-mNG or ARF-mNG chimeras (black, inverted images). Scale bars, 20 μm. Magnified images of boxed regions shown (a–h). Scale bars, 10 μm. White arrows, ARF mNG in discrete puncta, black arrows, areas of concentrated peri-nuclear staining. $n$ = 3 experimental replicates. **(F)** Quantitation of E (upper panels). $n$ = 3 experimental replicates with 43 (ARF3), 97 (3N/1C), 76 (1N/3C) cells quantified in total. Graph is percentage of cells with mNG concentrated in the peri-nuclear region. Data is mean ± SEM. P values (Student's two-tailed $t$ test), ***P ≤ 0.001. **(G)** 2D PC3 cells expressing mNG, ARF3-mNG, or ARF-mNG chimeras classified into Round, Spindle, and Spread. Heatmaps, Log$_2$ fold change over mNG. P values, one-way ANOVA, grayscale values as indicated. $n$ = 2 experimental replicates with 4 technical replicates/condition. 4,746 (mNG), 7,342 (ARF3-mNG), 5,567 (3N/1C), and 4,717 (1N/3C) mNG-positive cells quantified in total. **(H)** Confocal images of PC3 acini stained with F-actin (red) and Hoechst (nuclei, blue; middle panels). Intensity of F-actin staining can be appreciated using FIRE LUT. Scale bars, 20 μm. $n$ = 3 experimental replicates. **(I)** Maximum intensity projections of PC3 acini expressing ARF3-mNG or ARF-mNG chimeras (green) stained with GM130 or RAB11 (red). Scale bars, 20 μm. Magnified images of boxed regions are shown, single plane. Scale bars, 10 μm. $n$ = 2 experimental replicates. **(J)** Phase images of PC3 acini (lower panels); Round (red), Spindle (green), and Spread (blue). Scale bar, 100 μm. $n$ = 3 experimental replicates each with 3 technical replicates/condition. **(K)** Quantitation of J. $n$ = 3 experimental replicates each with 3 technical replicates/condition. 3,765 (mNG), 2,218 (ARF3-mNG), 1,150 (3N/1C), and 3,067 (1N/3C) mNG-positive acini quantified in total. Heatmaps, Area is mean of Z-score normalized values (purple to yellow). P values, Student's $t$ test, Bonferroni adjustment, represented by size of bubble. Heatmaps, Round, Spindle, or Spread is Log$_2$ fold change from control (mNG; blue to red). Proportion of control at each time is also Z-score normalized (white to black). P values, CMH test, Bonferroni adjusted, represented by size of bubble. Dot indicates P value (Breslow–Day test, Bonferroni-adjusted) for consistent effect magnitude.

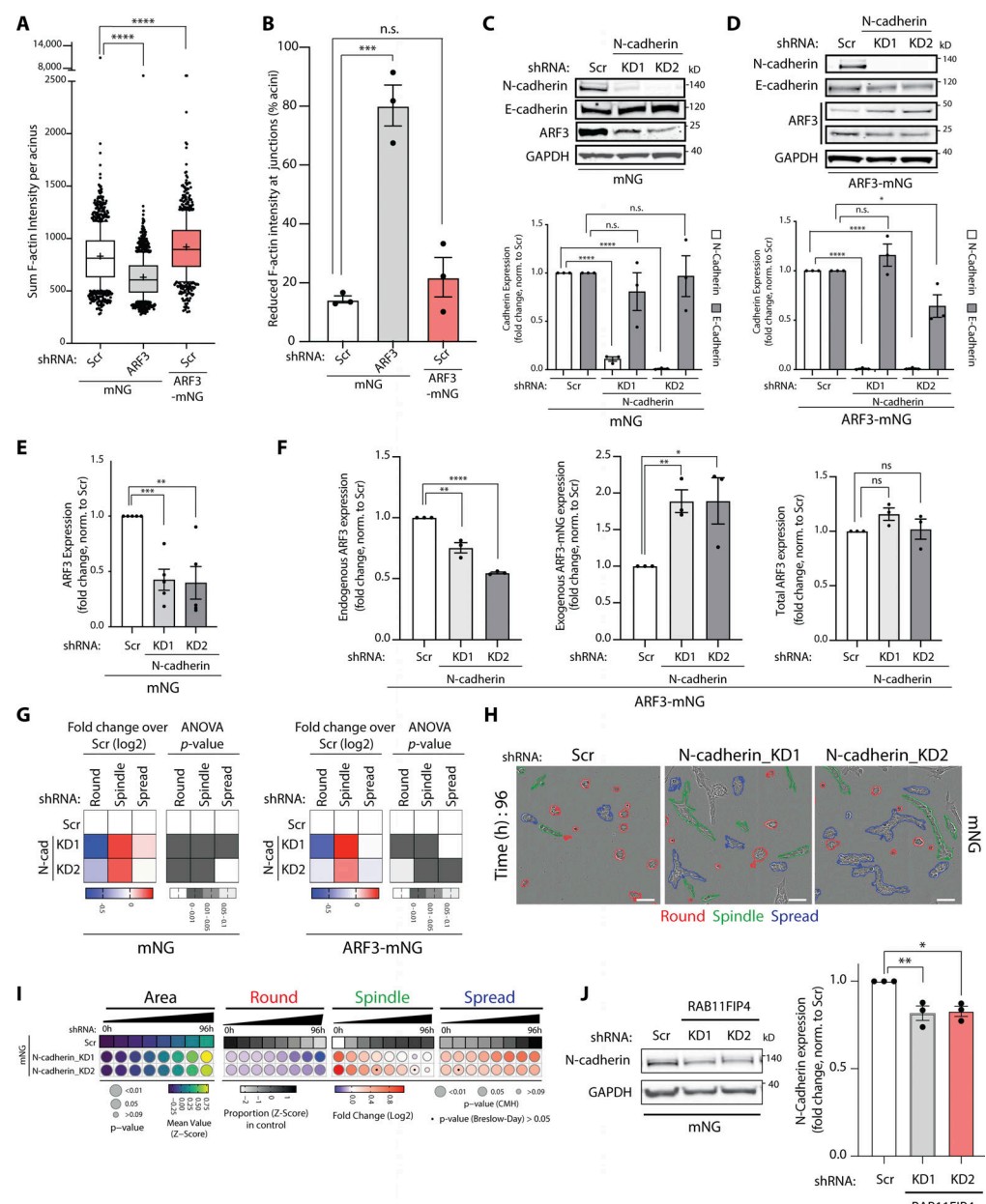

Figure S4. **N-cadherin depletion mimics ARF3 depletion in 2D and 3D assays. (A)** Quantitation of total F-actin intensity/acini described in Fig. 6 A. Box and whiskers plot, 10–90 percentile; +, mean; dots, outliers; midline, median; boundaries, quartiles. n = 2 experimental replicates with 643 (mNG, Scr shRNA), 712 (mNG, *ARF3* KD1 shRNA), and 383 (ARF3-mNG, Scr shRNA) cells quantified in total. P values (Student's two-tailed t test), ****P ≤ 0.0001. **(B)** Quantitation of percentage of PC3 acini described in Fig. 6 A with F-actin intensity visibly reduced in junctions. Data is mean ± SEM. n = 3 experimental replicates with 121 (mNG, Scr shRNA), 108 (mNG, *ARF3* KD1 shRNA), and 121 (ARF3-mNG, Scr shRNA) cells quantified in total. P values (Student's two-tailed t test), ***P ≤ 0.001. **(C–F)** Representative Western blots of PC3 cells expressing (C) mNG or (D) ARF3-mNG and Scr or N-cadherin shRNA for N-cadherin, E-cadherin, and ARF3 antibodies. GAPDH is a loading control for both cadherin blots and a sample control for ARF3. Graphs are fold change, normalized to Scr. Data is mean ± SEM for n = 3 or 5 independent lysate preparations for mNG (C and E) and 3 independent preparations for ARF3-mNG (D and F). P values (Student's two-tailed t test), *P ≤ 0.05, **P ≤ 0.01, ***P ≤ 0.001, and ****P ≤ 0.0001. **(G)** 2D PC3 cells expressing mNG or ARF3-mNG and Scr or N-cadherin shRNA were classified into Round, Spindle, and Spread. Heatmaps, Log₂ fold change over Scr. P values, one-way ANOVA, grayscale values as indicated. n = 3 experimental replicates with 3 technical replicates/condition. 19,184 (Scr), 37,230 (N-cadherin_KD1), 26,284 (N-cadherin_KD2) and 16,210 (Scr), 29,065 (N-cadherin_KD1), 46,352 (N-cadherin_KD2) cells were quantified for mNG or ARF3-mNG, respectively. **(H)** Representative phase images of PC3 acini expressing mNG and Scr or N-cadherin shRNA. Outlines: Round (red), Spindle (green), and Spread (blue). Scale bar, 100 µm. n = 5 experimental replicates each with 3–4 technical replicates/condition. 23,538 (Scr), 34,624 (N-cadherin_KD1), and 36,432 (N-cadherin_KD2) acini quantified in total. **(I)** Quantitation of H. Heatmaps, Area is mean of Z-score normalized values (purple to yellow). P values, Student's t test, Bonferroni adjustment, represented by size of bubble. Heatmaps, Round, Spindle, or Spread is Log₂ fold change from control (Scr; blue to red). Proportion of control at each time is Z-score normalized (white to black). P values, CMH test, Bonferroni adjusted, represented by size of bubble. Dot indicates P value (Breslow–Day test, Bonferroni-adjusted) for consistent effect magnitude. **(J)** Western blot of PC3 cells expressing mNG and Scr or *RAB11FIP4* shRNA for N-cadherin and GAPDH, as a loading control. Graph is fold change, normalized to Scr. Data is mean ± SEM. Panels shown are representative of 3 independent lysate preparations. P values (Student's two-tailed t test), *P ≤ 0.05 and **P ≤ 0.01. Source data are available for this figure: SourceData FS4.

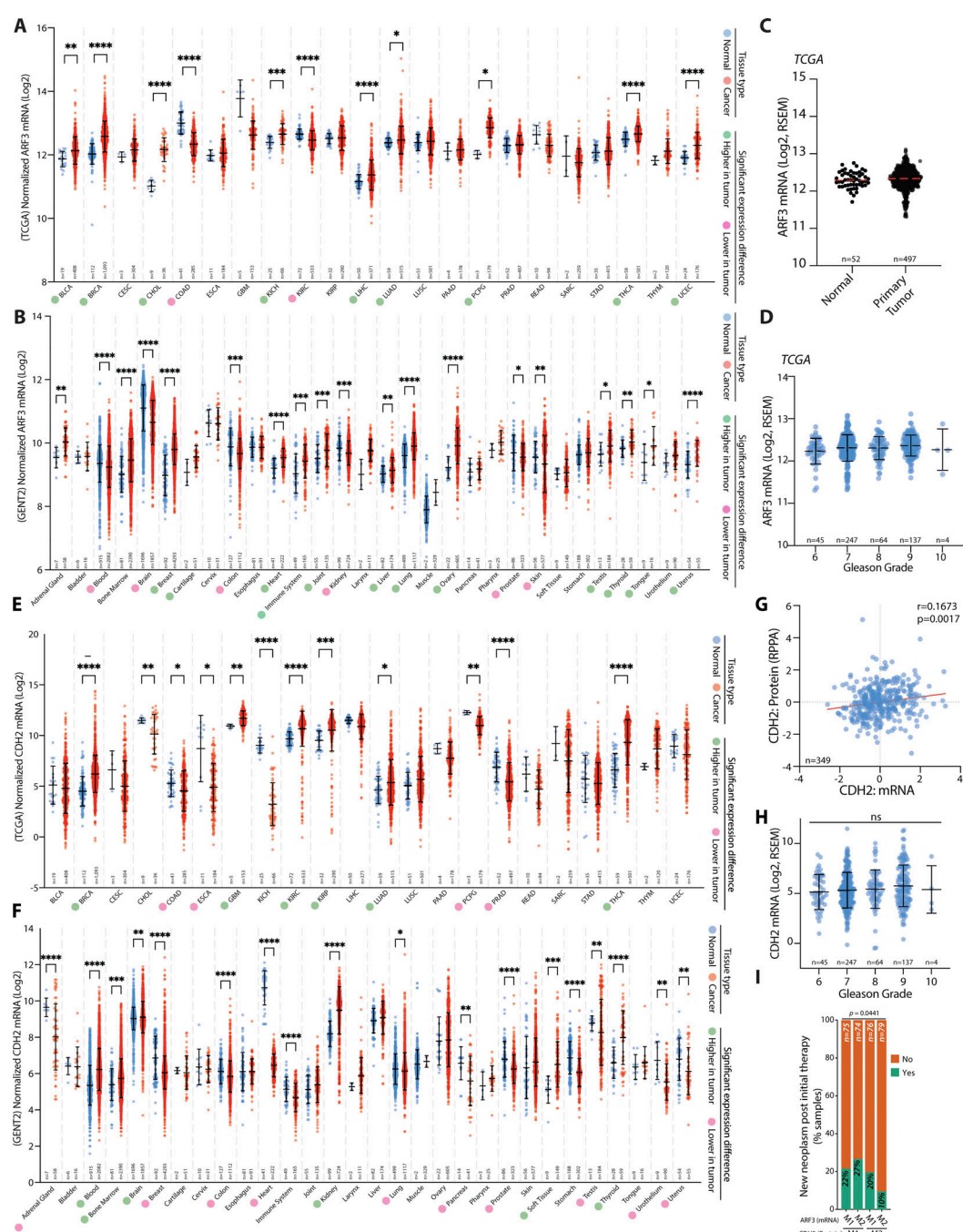

Figure S5. **N-cadherin and ARF3 expression in prostate cancer patients. (A and B)** Analysis of Normal vs. Tumor mRNA expression (Log₂) for *ARF3* from TCGA and GENT2 datasets (see Materials and methods for dataset IDs). Dots, expression per patient. Blue, normal; red, tumor. Circle next to Tumor Type indicates significant expression directionality change in tumor compared to normal tissue: Green, higher in tumor; magenta, lower in tumor. Patient numbers on graph. P values, Brown-Forsythe and Welch ANOVA test with unpaired Welch's correction, with individual variances compared for each comparison. *, 0.0332; **, 0.0021; ***, 0.0002; ****, <0.0001. **(C)** *ARF3* mRNA (Log2, RSEM) from Normal vs. Primary Tumor prostate samples (dataset, TCGA). Patient numbers, on graph. P values, two-tailed unpaired *t* test with Welch's correction. **(D)** *ARF3* mRNA (Log₂, RSEM) from prostate tumor samples with different Gleason Grade scores (dataset, TCGA). Patient numbers, on graph. P values, one-way ANOVA with Tukey corrections for multiple testing. **(E and F)** Analysis of Normal versus Tumor mRNA expression (Log₂) for *CDH2* from TCGA and GENT2 datasets (see Materials and methods for dataset IDs). Dots, expression per patient. Blue, normal; red, tumor. Circle next to Tumor Type indicates significant expression directionality change in tumor compared to normal tissue: Green, higher in tumor; magenta, lower in tumor. Patient numbers on graph. P values, Brown–Forsythe and Welch ANOVA test with unpaired Welch's correction, with individual variances compared for each comparison. *, 0.0332; **, 0.0021; ***, 0.0002; ****, <0.0001. **(G)** Correlation (Pearson) between *CDH2* mRNA and N-cadherin protein in prostate tumors (dataset, TCGA). **(H)** *CDH2*/N-cadherin mRNA levels (Log₂, RSEM) from prostate tumor samples with different Gleason Grade scores (dataset, TCGA). Patient numbers, on graph. P value, one-way ANOVA with Tukey corrections for multiple testing. **(I)** Neoplasm status upon grouping of prostate cancer patients based on median split (M1, low versus M2, high) of CDH2/N-cadherin protein and *ARF3* mRNA expression in prostate tumors. Dataset, TCGA. Patient numbers, on graph. Data presented as % of samples in each quartile grouping in presented categories. P values, chi-squared test.

Video 1.   **Live time-lapse imaging (phase) showing the formation of PC3 acini, expressing mem:Venus pLKO.4 Scr shRNA, from single cells in ECM.** Images were collected every hour for 96 h and displayed at 7 frames/s. Video related to images shown in Fig. 1 E.

Video 2.   **Live time-lapse imaging (Venus) showing the formation of PC3 acini, expressing mem:Venus pLKO.4 Scr shRNA, from single cells in ECM.** Images were collected every hour for 96 h and displayed at 7 frames/s. Video related to images shown in Fig. 1 E.

Video 3.   **Live time-lapse imaging (phase) representative of migration of individual PC3 cells in wounded monolayers described in** Fig. 3 B**.** Frames were collected every hour for 28 h and displayed at 7 frames/s.

Video 4.   **Live time-lapse imaging (phase) representative of sheet-like invasion of PC3 cells in wounded monolayers described in** Fig. 4 C**.** Frames were collected every hour for 56 h and displayed at 7 frames/s.

Video 5.   **Live time-lapse imaging (phase) representative of chain-led invasion of PC3 cells in wounded monolayers described in** Fig. 3 D**.** Frames were collected every hour for 56 h and displayed at 7 frames/s.

**Provided online are five tables. Table S1 shows shRNA library information and result from screen. Table S2 shows non-screen RNAs used in this study. Table S3 lists antibodies used in this study. Table S4 shows combined normal and tumor dataset information for ARF3. Table S5 shows combined normal and tumor dataset information for CDH2.**

