## [Peer Review File · The Journal of Cell Biology]

The small GTPase ARF3 controls invasion modality and metastasis by regulating N-cadherin levels

Emma Sandilands, Eva Freckmann, Erin Cumming, Alvaro Roman-Fernandez, Lynn McGarry, Jayanthi Anand, Laura Galbraith, Susan Mason, Rachana Patel, Colin Nixon, Jared Cartwright, Hing Leung, Karen Blyth, and David Bryant

Corresponding Author(s): David Bryant, University of Glasgow

Review Timeline:

Submission Date:	2022-06-23
Editorial Decision:	2022-07-18
Revision Received:	2022-12-13
Editorial Decision:	2023-01-02
Revision Received:	2023-01-13

Monitoring Editor: Ian Macara

Scientific Editor: Tim Fessenden

Transaction Report:

DOI: <https://doi.org/10.1083/jcb.202206115>

Revision 0

Review #1

1. Evidence, reproducibility and clarity:

Evidence, reproducibility and clarity (Required)

This study represents a tour de force in advancing our understanding of the Arf family and its associated regulators/effectors in controlling the morphology of individual cells and collective behaviours in 3D ECM. The authors use shRNA-mediated knockdown in high throughput imaging and AI approaches to define the influence of the ARFome on the morphology of prostate cancer PC3 cells growing as acini structures in 3D ECM. This allowed classification of ARFs and effectors/regulators on the basis of associated phenotypes, and this in itself is a useful resource for the community. It also drove the authors to focus on ARF3 and its regulator ESD and effector Rab11-FIP4. Analysing cell motility in elegant 2D and 3D assays showed that ARF3 levels control the collective migration of PC3 cells in 3D invasion, and the authors relate these findings to an interaction between Arf3 and N-cadherin. Using a mouse model of intra-prostatic injection of PC3 cells they demonstrate clear links between ARF3 levels and metastasis in vivo, and patient data further supports the link between Arf3, N-cadherin and metastasis. Experiments are well controlled and complex data are beautifully presented. In general data support the conclusions, where this is not as clear is highlighted below.

****Major comments:****

Figure 4: The use of the Arf3/4 chimeras is an interesting approach, used to show that the ARF3 C-terminus is important for its function related to migration/invasion. However the effect this has is not clear- it is not GTP loaded efficiently and may therefore act as a dominant negative. Furthermore the authors do not indicate which intracellular compartment ARF3 associates with, or if this is altered when the ARF3 C-term is replaced by that of ARF4 (ARF3N/1C).

Figure 5: Links to N-cadherin are clearly interesting, but the model proposed in Figure 5K is a little speculative. Clearly it is possible that ARF3/Rab11-FIP4 regulate N-cadherin trafficking such that loss of the pathway leads to degradation and gain promotes stability, but it is also possible that expression levels are controlled at the level of transcription. This could be assessed by a simple surface labelling experiment in wt, overexpressing and knockdown cells, and/or by analysing localisation of N-cad with respect to ARF3 and late endosomes/lysosomes when ARF3 levels are manipulated. Does ESD knockdown similarly impact N-cad?

Figure 6: The metastasis experiments are highly relevant to metastatic prostate cancer. Essentially the overall conclusion that high Arf3 (overexpression) suppresses and low Arf3 (knockdown) supports metastasis are well supported by the data, and the wildtype Arf3 levels sit in between (hence trends are observed but aren't statistically significant). Here it would be interesting to compare ARF3 levels in patient tumours with those in wt, overexpressing and

knockdown PC3 cells in the mouse model (if sections are available) to give confidence that the overexpression is within the physiological range. If it were also possible to analyse N-cadherin levels in tumours or metastases that would provide an even stronger case for the mechanism proposed.

****Minor comments:****

Figure 2: The classification of phenotypes into groups is interesting, but the trends in some groups (eg Group4, 6 and 7) seem very similar. It wasn't clear to me if these are AI generated? Also, knockdown of individual ARFs is often in different groups- is this a reflection of knockdown level?

2. Significance:

Significance (Required)

The manuscript provides a very significant advance in our understanding of the function of the ARFome with respect to cell morphodynamics. The first two figures and supporting data represent a fantastic resource for the field, and the remaining figures provide new insight into the function of ARF3 in collective cell movement and metastasis in mouse models and patients. Whilst ARF6 and its function in cell migration/invasion/metastasis is well studied, ARF3 has received relatively little attention. This study is therefore of broad interest to the trafficking community, and the new links between ARF3 and invasion/metastasis are broadly of interest to the cell biology and cancer communities. The mechanistic link between N-cadherin and ARF3 is fairly well defined and the fact that high/low levels of both correspond to improved/poor outcomes is a major strength of the study.

Expertise: Vesicle trafficking/cell migration/invasion/cancer

3. How much time do you estimate the authors will need to complete the suggested revisions:

Estimated time to Complete Revisions (Required)

(Decision Recommendation)

Between 1 and 3 months

4. Review Commons values the work of reviewers and encourages them to get credit for their work. Select 'Yes' below to register your reviewing activity at Publons; note that the content of your review will not be visible on Publons.

Reviewer Publons

Yes

Review #2

1. Evidence, reproducibility and clarity:

Evidence, reproducibility and clarity (Required)

In this manuscript, Sandilands et al. analyze the role of ARF GTPases ARF1 and ARF3 in human prostate cancer cell setting with specific focus on 3D versus 2D growth and invasion. The study connects to previous work where the authors conducted similar studies with ARF GTPase exchange factor IQSEC1 acting as an invasion promoting factor. Now, the authors interrogate the "ARFome" in prostate cancer cell line PC3 by use of a lentiviral shRNA library. The authors conduct detailed 3D and 2D cell culture analyses to identify specific differences in cell morphology between the different single knockout clones, showing loss of ARF1 and ARF3 as key switches of a spindle like morphology that is associated with enhanced migration and invasion. Interestingly, the authors find ARF3 functioning significantly dependent on the C-terminal region, and, further, that ARF3 is a direct companion of N-cadherin levels, whose downregulation leads to enhanced migratory capability.

The main findings of the manuscript are: (1) ARF1 and ARF3 knockdown elicits key-differences in 3D morphology and migratory capacity, (2) specifically ARF3 is associated with maintenance of N-cadherin levels via PSD and RAB11/FIP4 effectors, (3) whose downregulation leads to enhanced metastatic spread in a orthotopic xenograft mouse model, and (4) lowered N-cadherin protein / ARF3 mRNA levels identify more aggressive human prostate tumors.

Analyses and experiments conducted in the manuscript are highly extensive, they follow robust methods and are well controlled. Results described are highly detailed and are impressively visualized and presented. Key data are highlighted, and the manuscript is clearly structured. However, some data could be described and argued more concisely, which would strongly support the results shown. I recommend publication after some minor but important changes.

****Major comments:****

1. The entire results are based on studies of a single cell line PC3 derived from a highly aggressive metastatic lesion. To infer such an essential principle of tumor invasion and migration from this may be a bit precarious, and perhaps this principle should also be demonstrated in another cell line. The choice of PC3 cells and its implications should at least be discussed.
2. Results showing cell morphology page 6 / Fig.2: end of paragraph: stating "normally suppressing invasion": seems too far at this state of the manuscript, as these experiments are shown in the next section. maybe better "involved in preservation of a rounded phenotype". Results Suppl.Fig.3f: please use the same colors for the morphologies as in Fig. 2 etc. (round - red, spindle - green, spread - blue).
3. Conclusive sentences should not be put at the beginning of an experiment, before one can

know its outcome: Results page 8, Fig.4g, bottom: "This revealed that the C-terminus of ARF3 is required for sheet type invasive activity" maybe put that rather as a conclusion of the whole section.

4. Results showing 2D migration and 3D invasion: In the illustrations of migration and invasion assays shown throughout Figs 3-5 and Suppl. Figs 3 and 4, please clearly state and indicate for each case whether this is 2D migration or 3D invasion, as these two assays are very similar, which is a little confusing throughout the manuscript. Suppl.Fig.3e,k,l: is this 2D or 3D? Results Fig.4b+d and Fig.5g: please amend "3D invasion".

5. Summarizing sketch Results Fig.5k: in the scheme on the right side indication of respective presence / absence of ARF3/N-cadherin is missing. Which state induces which condition? Please amend.

6. ARF3 suppresses PC3 xenograft metastasis shown in Fig.6: N-cadherin stainings of mouse xenografts and metastases are missing.

7. Patient data ARF3 mRNA correlations Fig.7 and Suppl.Fig.6ab / Results page 11: the whole section describing ARF3 mRNA levels in diverse tumor types is too long and a little bit confusing: maybe shorten the text, put Fig.7f+g supplemental, and please indicate the combined GENT2 database in Fig. S6b.

8. Patient data CDH2 mRNA/protein correlations: Fig.7 and Suppl.Fig.6cd / Results page 12: one should also shorten and sharpen this section. Please also exchange Suppl. Fig. panels 6 dc to have the same order as Suppl. Fig.6ab. Regarding the detailed analyses of CDH2 mRNA levels, make a too long story short, essential are N-cadherin protein levels, and these results shown in Fig. 7 hik and Fig. 7rst should be enlarged and highlighted. All other data (Fig. 7jl) and the right bars of panels Fig. 7mno, as well as Fig. 7pq are rather supplemental.

9. The finding of elevated N-cadherin levels correlated with reduced invasion/migration and according better tumor outcome is surprising, particularly with regard to the quite established "EMT" dogma of N-cadherin driven single cell migration. Could you go into more detail about this property of N-cadherin driven mode of reduced tumor spread in the discussion?

****Minor comments:****

1. Manuscript title: maybe rephrase and reverse order of events: first invasion, then metastasis?

2. Results Suppl.Fig.1ij: which cells are shown? Please amend RWPE-1 and PC3.

3. Results page 5: although already described in Nacke et al 2021, please explain the term "acinus".

4. Results page 6: maybe introduce an additional subchapter? Some subchapter titles could be more explicative.

5. Results page 6, 2nd paragraph: please describe what was done: "revealed that knockdown of ..."

6. Results page 7: please explain more detailed class I ARFs, which ARFs are included in this class?

7. Results page 7, Fig.3f-j and 2nd paragraph: maybe better switch: first 3D, then 2D? Results Fig.3j: maybe amend indication of knockdown "KD" as indicated in Figure 3b-e.

8. The conclusion on page 8 top "this suggests that a function of ClassI ARFs may be to regulate molecules that control collective behaviours" is quite broad, please be more specific.

9. Suppl.Fig.4n: should be named "I"?

10. Results page 8, introducing sentence: please formulate more clearly and following the previous results.

2. Significance:

Significance (Required)

****Contents/Level of interest/merit:****

This study by Sandilands et al. analyzes cellular models of ARF-GTPase linked changes of cell morphology and migration to detect altered prostate cancer cell metastatic behaviour. Understanding the contribution of specific ARF GTPases in cancer cell shape and movement might help to identify markers of disease progression and metastasis.

****Strengths/Conclusions:****

The authors perform whole ARF-compendium knockdown and conduct detailed data analysis and visualization. The authors perform morphological analyses and conduct migration and invasion studies. Mechanistically, they confirm expression changes of N-cadherin, the key adhesive protein that is regulated. This study underlines the importance of analyzing subtle GTPase pathway differences by detailed morphological observations and methods.

****Comment/Weakness:****

The authors show extensive and detailed data that have been thoroughly analyzed, and results are presented and described fluently. There is a clear sequence of results description that is presented detailed. Form and contents of the paper is sound. The experiments are highly connected to previous experiments and data and this is also the major drawback of this manuscript: there is a lack of clear description of what is shown because it is already presupposed. Therefore, some sections should be worded and presented in more detail to present results more explicitly. The manuscript can be accepted with minor but essential revisions.

3. How much time do you estimate the authors will need to complete the suggested revisions:

Estimated time to Complete Revisions (Required)

(Decision Recommendation)

More than 6 months

4. Review Commons values the work of reviewers and encourages them to get credit for their work. Select 'Yes'

below to register your reviewing activity at Publons; note that the content of your review will not be visible on Publons.

Reviewer Publons

Yes

Review #3

1. Evidence, reproducibility and clarity:

Evidence, reproducibility and clarity (Required)

****Summary:****

Provide a short summary of the findings and key conclusions (including methodology and model system(s) where appropriate).

The manuscript is dedicated to a study of functional roles of a panel of cell migration regulatory factors, and notably the highly homologous family of ARF GTPases. The chosen model is a prostate cancer cell line, used in a number of assays in culture, as well as in a study of primary tumor formation and metastasis in mice. The authors apply a rigorous quantitative approach to their assays of 2D and 3D migration in culture, and use artificial intelligence for the analysis of results, thus upgrading their work from mere phenotypic observations, and gaining statistically significant results. The main finding of the study is the discovery of a unique role of ARF3, a regulatory protein that is shown to control a switch between individual and collective cell migration depending on its abundance. In fact, a depletion of ARF3 leads to an increased individual cell migration and invasion, and to increased metastasis formation in mice, whereas an overexpression of ARF3 favors a sheet-like collective cell migration, which is also more efficient than control in culture, but does not induce metastasis in vivo. This phenomenon appears to depend on the levels of cellular N-cadherin, that is shown to be positively regulated by ARF3 on the protein level, by a mechanism that remains unclear. Finally, the authors analyze the expression of ARF3 and N-cad in a variety of tumors of different origins and grades, and attempt to show the prognostic value of these factors for progression-free survival.

****Major comments:****

- Are the key conclusions convincing?

The majority of conclusions are convincing, with the exception of the observations I will address in the next paragraph.

- Should the authors qualify some of their claims as preliminary or speculative, or remove them altogether?

In Fig. 2B, the phenotypes 1 (Spread, pink line) and 5 (Spindle, yellow line) are described in the text as showing a "modest but robust increase". No such increase is evident by eye, and if it is statistically robust, the information should be presented in the Figure, or the statement should be modified/removed from the article.

In Fig. 2E, the lower middle panel shows a dramatically increased quantity of acini, whereas the authors specifically say that proliferation is not impacted by the KD of ARF3, and indeed, the KD2 looks very much like the control in this respect. It is misleading, and a more typical panel should be presented for ARF3_KD1.

In Fig. 3, the authors study the effects of simple versus double KD of ARF1 and ARF3, and conclude that a double KD leads to a phenotype "midway" between the two simple KDs. However, with regard to the 3D invasion assays (Figs. 3DEJ), it looks like a double KD is less efficient than either of the simple ones, as if ARF1 and 3 were partially mutually dependent in this regulation. It is not clear what "midway phenotype" the authors are talking about.

- Would additional experiments be essential to support the claims of the paper? Request additional experiments only where necessary for the paper as it is, and do not ask authors to open new lines of experimentation.

The authors make a strong case for physical interaction and mutual stabilization of ARF3 and N-cad, though the negative regulation of N-cad following ARF3 depletion is not obvious from Fig. 5B (positive regulation is very clear). Moreover, it is difficult to understand why a total disappearance of ARF3 has such a discreet effect on N-cad, whereas a very modest overexpression of ARF3 leads to such a dramatic increase of N-cad. Perhaps, some experiments with proteasome inhibition (using MG132, for example) could substantiate the authors' claim about the mutual stabilization of the two proteins.

- Are the suggested experiments realistic in terms of time and resources? It would help if you could add an estimated cost and time investment for substantial experiments.

Yes, the experiments are realistic and should not take more than a month.

- Are the data and the methods presented in such a way that they can be reproduced?

Yes.

- Are the experiments adequately replicated and statistical analysis adequate?

I am not sufficiently qualified in artificial intelligence algorithms to judge this part of the study. In general, as mentioned above, all differences characterized as "significant" or "robust" should have a statistical basis for this statement, which was not always the case in the manuscript.

****Minor comments:****

- Specific experimental issues that are easily addressable.

Please see above.

- Are prior studies referenced appropriately?

To the best of my knowledge, yes.

- Are the text and figures clear and accurate?

Yes, with the exceptions described before.

- Do you have suggestions that would help the authors improve the presentation of their data and conclusions?

Please see above.

****Referees cross-commenting****

I fully agree with the detailed and careful analysis made by the reviewers 1 and 2. I do not have any additional comments.

2. Significance:

Significance (Required)

- Describe the nature and significance of the advance (e.g. conceptual, technical, clinical) for the field.

The study shows, for the first time and in a very clear way, that the small GTPase ARF3 has a unique function in determining the pattern of human cancer cell migration, that this function depends on the C-terminal domain of ARF3 and on N-cadherin (by mechanisms that remain to be elucidated), and that this phenomenon is important for metastases formation in vivo.

- State what audience might be interested in and influenced by the reported findings.

The study will be interesting for the field of cell migration, but also for specialists in cancer and metastasis formation.

- Define your field of expertise with a few keywords to help the authors contextualize your point of view. Indicate if there are any parts of the paper that you do not have sufficient expertise to evaluate.

Cell migration 2D 3D, acini assay, RAC-WAVE-ARP2/3 pathway. I am not sufficiently qualified to evaluate the robustness of the artificial intelligence algorithms, nor to judge the relevance of the analysis presented in Fig. 7.

3. How much time do you estimate the authors will need to complete the suggested revisions:

Estimated time to Complete Revisions (Required)

(Decision Recommendation)

Less than 1 month

4. Review Commons values the work of reviewers and encourages them to get credit for their work. Select 'Yes' below to register your reviewing activity at Publons; note that the content of your review will not be visible on Publons.

Reviewer Publons

Yes

Revision Plan

Manuscript number: RC-2022-01446

Corresponding author(s): David Bryant

[The “revision plan” should delineate the revisions that authors intend to carry out in response to the points raised by the referees. It also provides the authors with the opportunity to explain their view of the paper and of the referee reports.]

The document is important for the editors of affiliate journals when they make a first decision on the transferred manuscript. It will also be useful to readers of the reprint and help them to obtain a balanced view of the paper.

*If you wish to submit a full revision, please use our "Full Revision" template. **It is important to use the appropriate template to clearly inform the editors of your intentions.**]*

1. General Statements [optional]

This section is optional. Insert here any general statements you wish to make about the goal of the study or about the reviews.

The goal of our study was to map the function of ARF GTPases in collective cancer cell behaviours. We addressed this by developing a machine-learning approach to classify collective 3-dimensional spheroid behaviours from long-term time-lapse imaging upon shRNA-mediated depletion of all ARF GTPases and their known interactors. This uncovered an unexpected role for the ARF3 GTPase and interactors in regulating collective invasion modality by acting as a rheostat to control N-cadherin levels. This ARF3 module thereby controlled metastasis in vivo and could be used in conjunction with N-cadherin levels to identify patients with poor clinical outcomes.

We feel that all Reviewers provided a fair and highly addressable set of comments. Their assessment is best appreciated in their own words, that **“this study represents a tour de force”** of **“elegant 2D and 3D assays”** to **“advance ... our understanding of the function of the ARFome”** as a **“useful resource for the community.”**

We thank the Reviewers for appreciating that **“this study is therefore of broad interest to the trafficking community, and the new links between ARF3 and invasion/metastasis are broadly of interest to the cell biology and cancer communities.”**

2. Description of the planned revisions

Insert here a point-by-point reply that explains what revisions, additional experimentations and analyses are planned to address the points raised by the referees.

Reviewer comments in bold and our response in italics.

Reviewer #1 (Evidence, reproducibility and clarity (Required)):

This study represents a tour de force in advancing our understanding of the Arf family and its associated regulators/effectors in controlling the morphology of individual cells and collective behaviours in 3D ECM. The authors use shRNA-mediated knockdown in high throughput imaging and AI approaches to define the influence of the ARFome on the morphology of prostate cancer PC3 cells growing as acini structures in 3D ECM. This allowed classification of ARFs and effectors/regulators on the basis of associated phenotypes, and this in itself is a useful resource for the community. It also drove the authors to focus on ARF3 and its regulator ESD and effector Rab11-FIP4. Analysing cell motility in elegant 2D and 3D assays showed that ARF3 levels control the collective migration of PC3 cells in 3D invasion, and the authors relate these findings to an interaction between Arf3 and N-cadherin. Using a mouse model of intra-prostatic injection of PC3 cells they demonstrate clear links between ARF3 levels and metastasis in vivo, and patient data further supports the link between Arf3, N-cadherin and metastasis. Experiments are well controlled and complex data are beautifully presented. In general data support the conclusions, where this is not as clear is highlighted below.

Major comments:

Figure 4: The use of the Arf3/4 chimeras is an interesting approach, used to show that the ARF3 C-terminus is important for its function related to migration/invasion. However the effect this has is not clear- it is not GTP loaded efficiently and may therefore act as a dominant negative. Furthermore the authors do not indicate which intracellular compartment ARF3 associates with, or if this is altered when the ARF3 C-term is replaced by that of ARF4 (ARF3N/1C).

The ARF 3N/1C chimera may be acting as a dominant negative fits with our data (decreased invasion compared to control in Fig 4k). We will update the manuscript to explicitly state and discuss this point. We can perform immunofluorescence experiments to determine which intracellular compartment ARF3 and each ARF1/3 chimera associates with, in 3D acini, to address the query as to whether and where these mutants differentially localise.

Figure 5: Links to N-cadherin are clearly interesting, but the model proposed in Figure 5K is a little speculative. Clearly it is possible that ARF3/Rab11-FIP4 regulate N-cadherin trafficking such that loss of the pathway leads to degradation and gain promotes

stability, but it is also possible that expression levels are controlled at the level of transcription. This could be assessed by a simple surface labelling experiment in wt, overexpressing and knockdown cells, and/or by analysing localisation of N-cad with respect to ARF3 and late endosomes/lysosomes when ARF3 levels are manipulated. Does ESD knockdown similarly impact N-cad?

We can directly address this point with experimental approaches readily achievable in our lab:

- *We can assess whether a transcriptional effect of ARF3 manipulation (control versus ARF3 knockdown versus ARF3 overexpression) on N-cadherin level by qPCR.*
- *Under the same ARF3 manipulations, we will use cell-surface labelling (biotinylation) of N-cadherin to analyse turnover of N-cadherin to identify how stability of N-cadherin may be affected by ARF3.*
- *We can examine N-cadherin levels upon PSD knockdown.*

Figure 6: If it were also possible to analyse N-cadherin levels in tumours or metastases that would provide an even stronger case for the mechanism proposed.

Our data predict that we should expect high N-cadherin protein levels in tumours and metastasis with high ARF3 expression, and the inverse in low ARF3 expression. Note that it is N-cadherin protein, not mRNA, that is predictive of survival in prostate cancer patients. N-cadherin protein and N-cadherin/CDH2 mRNA are poorly correlated ($r=0.1673$; Fig 7j). Indeed, CDH2 mRNA across multiple prostate cancer patients show no consistent trend when comparing normal to primary tumour to metastasis. As such, we did not show this data in the previous submission but can include if required.

We can directly test whether N-cadherin protein is high in primary tumours and metastases by:

- *Perform immunostaining and quantitation of N-cadherin protein levels in xenografts of cells manipulated for ARF3 levels (control vs ARF3 shRNA vs ARF3 overexpression).*

Minor comments:

Figure 2: The classification of phenotypes into groups is interesting, but the trends in some groups (eg Group4, 6 and 7) seem very similar. It wasn't clear to me if these are AI generated? Also, knockdown of individual ARFs is often in different groups- is this a reflection of knockdown level?

These points can be addressed by text changes to state the answers to these questions.

For detection of phenotype classes, shRNAs were grouped into 7 groups based on a dendrogram from clustering the fold change over time for each shRNA in Round, Spindle, and Spread phenotypes compared to the control (Scr) value. The dendrogram was generated using hierarchical clustering of

Revision Plan

heatmap data by complete linkage of Euclidian distances between samples. We can make this point more clearly in the description to the 'ARFome shRNA screen' section of Methods.

*For the shRNA screen construction, we used 1-2 shRNA sequences per target **if** they had previously been validated by qPCR, or 5 sequences per target if unvalidated. Supplementary Table 1 lists all sequences and validation. We can make this point more clearly in the 'ARFome shRNA screen' section of Methods. As the reviewer notes some of the phenotype groups we describe exhibit similar trends, with the main difference being the degree to which the changes in each phenotype occur. This may be why some shRNAs against the same target appear in different groups (shown in Figure 1c), as they reflect different levels of knockdown efficiency. A comment on this can be added to Results section. To account for such discrepancy from the screen, any targets further investigated were independently knocked down using the same shRNA sequences expressed in a pLKO.1 puromycin vector (described in Supplementary Table S2) and knockdown efficiency confirmed by western blot.*

Reviewer #1 (Significance (Required)):

The manuscript provides a very significant advance in our understanding of the function of the ARFome with respect to cell morphodynamics. The first two figures and supporting data represent a fantastic resource for the field, and the remaining figures provide new insight into the function of ARF3 in collective cell movement and metastasis in mouse models and patients. Whilst ARF6 and its function in cell migration/invasion/metastasis is well studied, ARF3 has received relatively little attention. This study is therefore of broad interest to the trafficking community, and the new links between ARF3 and invasion/metastasis are broadly of interest to the cell biology and cancer communities. The mechanistic link between N-cadherin and ARF3 is fairly well defined and the fact that high/low levels of both correspond to improved/poor outcomes is a major strength of the study.

Expertise: Vesicle trafficking/cell migration/invasion/cancer

Reviewer #2 (Evidence, reproducibility and clarity (Required)):

In this manuscript, Sandilands et al. analyze the role of ARF GTPases ARF1 and ARF3 in human prostate cancer cell setting with specific focus on 3D versus 2D growth and invasion. The study connects to previous work where the authors conducted similar studies with ARF GTPase exchange factor IQSEC1 acting as an invasion promoting factor. Now, the authors interrogate the "ARFome" in prostate cancer cell line PC3 by use of a lentiviral shRNA library. The authors conduct detailed 3D and 2D cell culture analyses to identify specific differences in cell morphology between the different single

knockout clones, showing loss of ARF1 and ARF3 as key switches of a spindle like morphology that is associated with enhanced migration and invasion. Interestingly, the authors find ARF3 functioning significantly dependent on the C-terminal region, and, further, that ARF3 is a direct companion of N-cadherin levels, whose downregulation leads to enhanced migratory capability.

The main findings of the manuscript are: (1) ARF1 and ARF3 knockdown elicits key-differences in 3D morphology and migratory capacity, (2) specifically ARF3 is associated with maintenance of N-cadherin levels via PSD and RAB11FIP4 effectors, (3) whose downregulation leads to enhanced metastatic spread in a orthotopic xenograft mouse model, and (4) lowered N-cadherin protein / ARF3 mRNA levels identify more aggressive human prostate tumors.

Analyses and experiments conducted in the manuscript are highly extensive, they follow robust methods and are well controlled. Results described are highly detailed and are impressively visualized and presented. Key data are highlighted, and the manuscript is clearly structured. However, some data could be described and argued more concisely, which would strongly support the results shown. I recommend publication after some minor but important changes.

Major comments:

2) Results showing cell morphology page 6 / Fig.2: end of paragraph: stating "normally suppressing invasion": seems too far at this state of the manuscript, as these experiments are shown in the next section. maybe better "involved in preservation of a rounded phenotype". Results Suppl.Fig.3f: please use the same colors for the morphologies as in Fig. 2 etc. (round - red, spindle - green, spread - blue).

We can change the text on p6 to "involved in preservation of a rounded phenotype".

We can also change the colours for 2D morphologies to match colours used for 3D morphologies – Round - red, Spindle – green and Spread - blue in Supplementary Figure 3f.

3) Conclusive sentences should not be put at the beginning of an experiment, before one can know its outcome: Results page 8, Fig.4g, bottom: "This revealed that the C-terminus of ARF3 is required for sheet type invasive activity" maybe put that rather as a conclusion of the whole section.

We can remove this sentence from p8.

4) Results showing 2D migration and 3D invasion: In the illustrations of migration and invasion assays shown throughout Figs 3-5 and Suppl. Figs 3 and 4, please clearly state and indicate for each case whether this is 2D migration or 3D invasion, as these two assays are very similar, which is a little confusing throughout the manuscript. Suppl.Fig.3e,k,l: is this 2D or 3D? Results Fig.4b+d and Fig.5g: please amend "3D invasion".

These assays are indeed very similar, and we apologise for any confusion. We can add labels to 2D migration assays in Figures 3b, 3c and Supplementary Figure 4g-j and to 3D Invasion assays in Figures 3d-e, 3j, 4a-d, 4j-k, 5g-h and Supplementary Figure 3k-l.

5) Summarizing sketch Results Fig.5k: in the scheme on the right side indication of respective presence / absence of ARF3/N-cadherin is missing. Which state induces which condition? Please amend.

ARF3 and N-cadherin labels can be added to Figure 5k schema.

6) ARF3 suppresses PC3 xenograft metastasis shown in Fig.6: N-cadherin stainings of mouse xenografts and metastases are missing.

We can use immunolabelling to assess N-cadherin levels in tumours and metastasis sections from our xenograft mouse model with PC3 with altered ARF3 levels. Reviewer 1 shares this query, so please see our response to this reviewer.

7) Patient data ARF3 mRNA correlations Fig.7 and Suppl.Fig.6ab / Results page 11: the whole section describing ARF3 mRNA levels in diverse tumor types is too long and a little bit confusing: maybe shorten the text, put Fig.7f+g supplemental, and please indicate the combined GENT2 database in Fig. S6b.

We can shorten this section of the manuscript to make conclusions clearer. We intend to move Fig7f-g to Supplementary Figure 6 and will indicate GENT2 on Supplementary Figure 6b as requested.

8) Patient data CDH2 mRNA/protein correlations: Fig.7 and Suppl.Fig.6cd / Results page 12: one should also shorten and sharpen this section. Please also exchange Suppl. Fig. panels 6 dc to have the same order as Suppl. Fig.6ab. Regarding the detailed analyses of CDH2 mRNA levels, make a too long story short, essential are N-cadherin protein levels, and these results shown in Fig. 7 hik and Fig. 7rst should be enlarged and highlighted. All other data (Fig. 7jl) and the right bars of panels Fig. 7mno, as well as Fig. 7pq are rather supplemental.

Revision Plan

As described above we can shorten this section of the manuscript and either remove or move Figure 7f-g, 7j, 7l, right hand panels of 7m-o, 7p and 7q to Supplementary Figure 6 as suggested by the reviewer. We can also exchange Supplementary Figure 6c and 6d.

9) The finding of elevated N-cadherin levels correlated with reduced invasion/migration and according better tumor outcome is surprising, particularly with regard to the quite established "EMT" dogma of N-cadherin driven single cell migration. Could you go into more detail about this property of N-cadherin driven mode of reduced tumor spread in the discussion?

Much of the differential role of E-cadherin and N-cadherin comes from studies on cadherin switches, wherein cells express either E-cadherin or N-cadherin. Much less is known about what happens when both are simultaneously co-expressed (which is the situation we have studied). Indeed, the dogma that E-cadherin anti-invasive and N-cadherin is for invasion has been challenged in recent years from the work of groups such as Andy Ewald and Peter Friedl, who have showed that E-cadherin is required for collective invasion and metastasis. The reviewer is right that we need to expand Discussion section of manuscript, which we will do.

Minor comments:

1) Manuscript title: maybe rephrase and reverse order of events: first invasion, then metastasis?

We can change the manuscript title to 'The small GTPase ARF3 controls invasion modality and metastasis by regulating N-cadherin levels' as suggested.

2) Results Suppl.Fig.1ij: which cells are shown? Please amend RWPE-1 and PC3.

These are PC3 and can be labelled as such.

3) Results page 5: although already described in Nacke et al 2021, please explain the term "acinus".

We can add a description of what we are referring to as acini/acinus and expand upon our methodology for setting up 3D cultures to p5.

4) Results page 6: maybe introduce an additional subchapter? Some subchapter titles could be more explicative.

We can add an additional subchapter heading to p6.

5) Results page 6, 2nd paragraph: please describe what was done: "revealed that knockdown of ..."

We can add the following to p6:

Revision Plan

'PC3 cells stably expressing each shRNA were then cultured in ECM and imaged every hour for 96 hours as described for the ARFome shRNA screen. Size, shape and movement features were generated for each acini and our machine learning classifications applied to categorise and quantify Round, Spindle and Spread phenotypes.'

6) Results page 7: please explain more detailed class I ARFs, which ARFs are included in this class?

On p7 we can explain that Class I ARFs include ARF1, ARF2 and ARF3. In humans, ARF2 has been lost during evolution. We will also state that the highly homologous ARF1 and ARF3 have been shown to have some functional redundancy in previous studies, but that here we identify new and distinct functions.

7) Results page 7, Fig.3f-j and 2nd paragraph: maybe better switch: first 3D, then 2D? Results Fig.3j: maybe amend indication of knockdown "KD" as indicated in Figure 3b-e.

We can switch the order of 2D and 3D on p7 and add 'KD' to labels for Figure 3f-j.

8) The conclusion on page 8 top "this suggests that a function of Class I ARFs may be to regulate molecules that control collective behaviours" is quite broad, please be more specific.

We will update this to state that although ARF1 and ARF3 were previously considered to be somewhat redundant, these show distinct profiles of association with the key cell-cell adhesion molecules E-cadherin and N-cadherin, respectively.

9) Suppl.Fig.4n: should be named "l"?

We think Supplementary Figure 4n is correctly labelled – l and m are used to show 2D phenotype elsewhere in the Figure but this numbering may change as we carry out revisions.

10) Results page 8, introducing sentence: please formulate more clearly and following the previous results.

We can change this sentence to better reflect previous results. Given our observations that depletion of ARF1 or ARF3 altered shape and movement in both 2D and 3D (Fig. 2e-g, 3b-e) we examined whether over-expression of Class 1 ARFs would also affect these processes.

Reviewer #2 (Significance (Required)):

Contents/Level of interest/merit:

Revision Plan

This study by Sandilands et al. analyzes cellular models of ARF-GTPase linked changes of cell morphology and migration to detect altered prostate cancer cell metastatic behaviour. Understanding the contribution of specific ARF GTPases in cancer cell shape and movement might help to identify markers of disease progression and metastasis.

Strengths/Conclusions:

The authors perform whole ARF-compendium knockdown and conduct detailed data analysis and visualization. The authors perform morphological analyses and conduct migration and invasion studies. Mechanistically, they confirm expression changes of N-cadherin, the key adhesive protein that is regulated. This study underlines the importance of analyzing subtle GTPase pathway differences by detailed morphological observations and methods.

Comment/Weakness:

The authors show extensive and detailed data that have been thoroughly analyzed, and results are presented and described fluently. There is a clear sequence of results description that is presented detailed. Form and contents of the paper is sound. The experiments are highly connected to previous experiments and data and this is also the major drawback of this manuscript: there is a lack of clear description of what is shown because it is already presupposed. Therefore, some sections should be worded and presented in more detail to present results more explicitly. The manuscript can be accepted with minor but essential revisions.

Reviewer #3 (Evidence, reproducibility and clarity (Required)):

Summary:

Provide a short summary of the findings and key conclusions (including methodology and model system(s) where appropriate).

The manuscript is dedicated to a study of functional roles of a panel of cell migration regulatory factors, and notably the highly homologous family of ARF GTPases. The chosen model is a prostate cancer cell line, used in a number of assays in culture, as well as in a study of primary tumor formation and metastasis in mice. The authors apply a rigorous quantitative approach to their assays of 2D and 3D migration in culture, and use artificial intelligence for the analysis of results, thus upgrading their work from mere phenotypic observations, and gaining statistically significant results. The main finding of the study is the discovery of a unique role of ARF3, a regulatory protein that is shown to control a switch between individual and collective cell migration depending on its

Revision Plan

abundance. In fact, a depletion of ARF3 leads to an increased individual cell migration and invasion, and to increased metastasis formation in mice, whereas an overexpression of ARF3 favors a sheet-like collective cell migration, which is also more efficient than control in culture, but does not induce metastasis in vivo. This phenomenon appears to depend on the levels of cellular N-cadherin, that is shown to be positively regulated by ARF3 on the protein level, by a mechanism that remains unclear. Finally, the authors analyze the expression of ARF3 and N-cad in a variety of tumors of different origins and grades, and attempt to show the prognostic value of these factors for progression-free survival.

Major comments:

- Are the key conclusions convincing?

The majority of conclusions are convincing, with the exception of the observations I will address in the next paragraph.

- Should the authors qualify some of their claims as preliminary or speculative, or remove them altogether?

In Fig. 2B, the phenotypes 1 (Spread, pink line) and 5 (Spindle, yellow line) are described in the text as showing a "modest but robust increase". No such increase is evident by eye, and if it is statistically robust, the information should be presented in the Figure, or the statement should be modified/removed from the article.

We believe that this point may be related to some confusion. In the text we state 'both ARF1 and ARF3 had one shRNA in each of Phenotype Groups 5 and 1, which displayed a modest but robust increase in Spindle and Spread behaviours, respectively.' In Figure 2B these changes are visible as Spread is represented in graphs by blue line and Spindle by green line. We think the pink (or red) line the reviewer is describing is actually the Round phenotype which is indeed unchanged or decreased.

Statistical significance is shown as s.e.m. on each graph in Figure 2b as a shaded region around each line and is described in Figure Legend. However, in some cases it may be hard to see this region as the effect across replicates is highly concordant and accordingly the s.e.m. is very small.

In Fig. 2E, the lower middle panel shows a dramatically increased quantity of acini, whereas the authors specifically say that proliferation is not impacted by the KD of ARF3, and indeed, the KD2 looks very much like the control in this respect. It is misleading, and a more typical panel should be presented for ARF3_KD1.

We can change the image in Figure 2e to show a more typical panel for ARF3_KD1.

Revision Plan

In Fig. 3, the authors study the effects of simple versus double KD of ARF1 and ARF3, and conclude that a double KD leads to a phenotype "midway" between the two simple KDs. However, with regard to the 3D invasion assays (Figs. 3DEJ), it looks like a double KD is less efficient than either of the simple ones, as if ARF1 and 3 were partially mutually dependent in this regulation. It is not clear what "midway phenotype" the authors are talking about.

We agree that this statement is confusing. We will update this section. The midway concept for ARF1+3 double knockdown was inferred from the 3D spheroid phenotype, not the invasion assays that the reviewer refers to. ARF1 knockdown induces a spindle phenotype but not major changes to spread phenotype, while ARF3 knockdown induces both spindle and spread 3D phenotypes (Fig 2e-g). ARF1+3 double knockdown results in spindle phenotype, but only a modest induction of spread behaviours. We therefore considered this as a midway between ARF1/3 alone.

- Would additional experiments be essential to support the claims of the paper? Request additional experiments only where necessary for the paper as it is, and do not ask authors to open new lines of experimentation.

The authors make a strong case for physical interaction and mutual stabilization of ARF3 and N-cad, though the negative regulation of N-cad following ARF3 depletion is not obvious from Fig. 5B (positive regulation is very clear). Moreover, it is difficult to understand why a total disappearance of ARF3 has such a discreet effect on N-cad, whereas a very modest overexpression of ARF3 leads to such a dramatic increase of N-cad. Perhaps, some experiments with proteasome inhibition (using MG132, for example) could substantiate the authors' claim about the mutual stabilization of the two proteins.

Our data suggest that while increasing ARF3 levels clearly increases N-cadherin levels, ARF3 is not obligate for N-cadherin expression. It may be that ARF3 regulates the turnover of a small but crucial pool of N-cadherin, which is counter-balanced by biosynthetic N-cadherin production. This may explain significant, but modest effects of ARF3 depletion on N-cadherin levels, but robust effect upon ARF3 overexpression. As the effect of ARF3 on N-cadherin turnover is also queried by Reviewer 1, we will address this by cell surface biotinylation of N-cadherin followed by analysis of N-cadherin turnover upon ARF3 knockdown or overexpression, with and without MG132. Please also see response to Reviewer 1.

We also note that while depletion of ARF3 induces strong phenotypic effects, these shRNAs result in about 75% reduction in expression. We were unable to recover clones using CRISPR for ARF3 knockout. It may also be that the remaining small amount of ARF3 is sufficient to maintain the N-cadherin that remains in cells. We can add discussion of these points to the manuscript.

Revision Plan

Are the suggested experiments realistic in terms of time and resources? It would help if you could add an estimated cost and time investment for substantial experiments.

Yes, the experiments are realistic and should not take more than a month.

- Are the data and the methods presented in such a way that they can be reproduced?

Yes.

- Are the experiments adequately replicated and statistical analysis adequate?

I am not sufficiently qualified in artificial intelligence algorithms to judge this part of the study. In general, as mentioned above, all differences characterized as "significant" or "robust" should have a statistical basis for this statement, which was not always the case in the manuscript.

Minor comments:

- Specific experimental issues that are easily addressable.

Please see above.

- Are prior studies referenced appropriately?

To the best of my knowledge, yes.

- Are the text and figures clear and accurate?

Yes, with the exceptions described before.

- Do you have suggestions that would help the authors improve the presentation of their data and conclusions?

Please see above.

CROSS-CONSULTATION COMMENTS

Revision Plan

I fully agree with the detailed and careful analysis made by the reviewers 1 and 2. I do not have any additional comments.

Reviewer #3 (Significance (Required)):

- Describe the nature and significance of the advance (e.g. conceptual, technical, clinical) for the field.

The study shows, for the first time and in a very clear way, that the small GTPase ARF3 has a unique function in determining the pattern of human cancer cell migration, that this function depends on the C-terminal domain of ARF3 and on N-cadherin (by mechanisms that remain to be elucidated), and that this phenomenon is important for metastases formation in vivo.

- State what audience might be interested in and influenced by the reported findings.

The study will be interesting for the field of cell migration, but also for specialists in cancer and metastasis formation.

- Define your field of expertise with a few keywords to help the authors contextualize your point of view. Indicate if there are any parts of the paper that you do not have sufficient expertise to evaluate.

Cell migration 2D 3D, acini assay, RAC-WAVE-ARP2/3 pathway. I am not sufficiently qualified to evaluate the robustness of the artificial intelligence algorithms, nor to judge the relevance of the analysis presented in Fig. 7.

3. Description of the revisions that have already been incorporated in the transferred manuscript

Please insert a point-by-point reply describing the revisions that were already carried out and included in the transferred manuscript. If no revisions have been carried out yet, please leave this section empty.

4. Description of analyses that authors prefer not to carry out

Please include a point-by-point response explaining why some of the requested data or additional analyses might not be necessary or cannot be provided within the scope of a revision. This can be due to time or resource limitations or in case of disagreement about the necessity of such additional data given the scope of the study. Please leave empty if not applicable.

Reviewer 1 - Figure 6: The metastasis experiments are highly relevant to metastatic prostate cancer. Essentially the overall conclusion that high Arf3 (overexpression) suppresses and low Arf3 (knockdown) supports metastasis are well supported by the data, and the wildtype Arf3 levels sit in between (hence trends are observed but aren't statistically significant). Here it would be interesting to compare ARF3 levels in patient tumours with those in wt, overexpressing and knockdown PC3 cells in the mouse model (if sections are available) to give confidence that the overexpression is within the physiological range.

The suggestion of comparing the xenograft mouse model expression of ARF3 to that of patient sections from tumours/metastases to infer these being in the same physiological range is intriguing. However, this is not technically feasible. This is because any comparison between these very different sample types (patient and xenograft mouse model) requires that all samples collected with the same experimental conditions. Such conditions could include time from surgery to embedding of samples, fixation methodology, embedding methodology, antigen retrieval methodology, or a number of other variables. All of these could affect the resulting staining for ARF3 in ways that preclude such analyses being reliable.

In silico analysis of ARF3 mRNA levels in PC3 cells performed at the same time as patient samples (Taylor, Cancer Cell, 2010; GSE21032) indicate that PC3 are on the higher end of, but still in, the physiological range of ARF3 expression in patient samples. However, this analysis should be considered in context; this involves comparison of mRNA from a tissue sample, including all sample types present, to an isolated cancer cell line. Therefore, a direct comparison between these two very different modalities will not answer this question.

Revision Plan

Reviewer 2 - 1) The entire results are based on studies of a single cell line PC3 derived from a highly aggressive metastatic lesion. To infer such an essential principle of tumor invasion and migration from this may be a bit precarious, and perhaps this principle should also be demonstrated in another cell line. The choice of PC3 cells and its implications should at least be discussed.

We also agree in theory that it would be useful to show key points of our mechanism in other cell lines. If absolutely required, we could profile additional cell lines for the effect of ARF3 depletion and overexpression. Note that this is a major undertaking; every additional cell line requires lentivirally transduced stable cell line generation (~1 month) for i-ii) scramble shRNA and ARF3 shRNA & iii-iv) mNeonGreen and ARF3-mNeonGreen expression (i.e. 4 stable cell lines). All of these then require 3D culture condition optimisation, before profiling from long-term time-lapse imaging (~2-3 months). Even profiling 5 additional lines would in reality be 20 stable lines over a 3-month period. This is an achievable, but arduous, set of experiments. We prefer to only do this if absolutely necessary.

PC3 were chosen because a) they have been extensively used in the literature as a model for metastatic prostate tumourigenesis, b) we have previously shown that they can be used to identify ARF GTPase modules that regulate 3D spheroid phenotype, in vivo metastasis, and elucidation of ARF GTPase modules that predict survival in prostate cancer patients, and c) of prostate cancer cell lines, they have high levels of all ARF GTPases. We can add discussion of these points to the paper.

July 18, 2022

Re: JCB manuscript #202206115T

Dr. David Bryant
University of Glasgow
CRUK Beatson Institute
Garscube Estate
Switchback Road
Glasgow G61 1BD
United Kingdom

Dear Dave,

Thank you for your patience with the additional review process for your manuscript "The small GTPase ARF3 controls metastasis and invasion modality by regulating N-cadherin levels". We apologize for the extra time this has taken but are pleased to report that the external reviewer was very supportive of publication and considers the study a technological tour de force. We invite you to submit a revision if you can address the reviewer's key concerns, outlined below.

Their only concern is the lack of mechanism by which Arf3 (and only Arf3) regulates N-cadherin behavior, and several suggested approaches are mentioned in the review. We feel that, since you are already planning on addressing the comments from the first round of Reviewer Commons evaluations, we would not expect you to tackle all of the proposed experiments but do feel that providing data on the surface levels of N-cadherin, and testing whether either EFA6A or Rab11FIP4 interact with Arf3 would strengthen the mechanistic aspects of the manuscript.

Please provide a point-by-point response to the review with a revised version of the manuscript. Given the relatively minor additions that are requested the revision can likely be handled at an editorial level and not returned to the reviewer for further evaluation.

GENERAL GUIDELINES:

Text limits: Character count for an Transfer is < 40,000, not including spaces. Count includes title page, abstract, introduction, results, discussion, and acknowledgments. Count does not include materials and methods, figure legends, references, tables, or supplemental legends.

Figures: Transfers may have up to 10 main text figures. Figures must be prepared according to the policies outlined in our Instructions to Authors, under Data Presentation, <https://jcb.rupress.org/site/misc/ifora.xhtml>. All figures in accepted manuscripts will be screened prior to publication.

Supplemental information: There are strict limits on the allowable amount of supplemental data. Transfers may have up to 5 supplemental figures. Up to 10 supplemental videos or flash animations are allowed. A summary of all supplemental material should appear at the end of the Materials and methods section.

Please note that JCB now requires authors to submit Source Data used to generate figures containing gels and Western blots with all revised manuscripts. This Source Data consists of fully uncropped and unprocessed images for each gel/blot displayed in the main and supplemental figures. Since your paper includes cropped gel and/or blot images, please be sure to provide one Source Data file for each figure that contains gels and/or blots along with your revised manuscript files. File names for Source Data figures should be alphanumeric without any spaces or special characters (i.e., SourceDataF#, where F# refers to the associated main figure number or SourceDataFS# for those associated with Supplementary figures). The lanes of the gels/blots should be labeled as they are in the associated figure, the place where cropping was applied should be marked (with a box), and molecular weight/size standards should be labeled wherever possible.

The typical timeframe for revisions is three to four months. While most universities and institutes have reopened labs and allowed researchers to begin working at nearly pre-pandemic levels, we at JCB realize that the lingering effects of the COVID-19 pandemic may still be impacting some aspects of your work, including the acquisition of equipment and reagents. Therefore, if you anticipate any difficulties in meeting this aforementioned revision time limit, please contact us and we can work with you to find an appropriate time frame for resubmission. Please note that papers are generally considered through only one revision cycle, so any revised manuscript will likely be either accepted or rejected.

Thank you for this interesting contribution to Journal of Cell Biology. You can contact us at the journal office with any questions, cellbio@rockefeller.edu or call (212) 327-8588.

Sincerely,

Ian Macara, Ph.D.
Editor
The Journal of Cell Biology

Tim Fessenden
Scientific Editor
Journal of Cell Biology

Reviewer #1 (Comments to the Authors (Required)):

This manuscript describes an interesting and novel role for the small GTPase Arf3 in metastasis of prostate cancer. As the authors note, high sequence homology among the 5 human Arfs and potential redundancies among their GEFs and GAPs has historically made it difficult to attribute specific functions to individual Arfs. Here the authors use an impressive array of technical approaches to define a very specific role for Arf3 in controlling 3D, but not 2D migration of PC3 prostate cancer cells in vitro. These in vitro studies are paralleled by in vivo studies showing that metastasis, but not growth, of PC3-derived tumors is modulated by Arf3. The specificity for Arf3 is surprising because Arf3 and Arf1 differ in only 7 amino acids, 4 at N-terminus and 3 at the C-terminus. Chimeras suggest that the C-terminal residues are more important, however a corresponding chimera containing the Arf3 N-terminus and Arf1 C-terminus doesn't bind GTP making it difficult to draw precise conclusions.

As noted above, this study is an impressive technical achievement - it combines high-throughput knockdown of not only the 5 human ARFs but also all known GEFs, GAPs and most effectors (i.e. the ARFome). The authors analyzed cell behavior for each knockdown in both 2D and 3D cultures over 4 days and used machine learning to classify corresponding phenotypes. Importantly, a distinction between Arf1 and Arf3 phenotypes was only observable in 3D.

This manuscript has many strengths, including the breadth of the analysis (encompassing the entire ArfOME), the range of analytical methods used and the careful quantitation of most of the outcomes. There are, however, a few weaknesses that will need to be addressed:

1. The data suggest that Arf3 acts to suppress prostate cancer metastasis through a regulated interaction N-cadherin. The data showing interaction of Arf3 with N-cadherin (Fig. 5f) and colocalization with it on intracellular compartments (Fig. 5e) are convincing (although the colocalization needs to be quantified). What is lacking is any measurement of N-cadherin localization in the presence and absence of Arf3. Is the ratio of surface/intracellular N-cadherin altered? This could be readily measured by flow cytometry, or by careful image quantitation.
2. How do the authors think Arf3 acts to control N-cadherin surface level? Does it control secretion and/or recycling or does it act allosterically to stabilize the protein? Does inhibition of lysosomal enzyme activity restore N-cadherin levels in Arf3-depleted cells (Fig. 5b,d)? This would be consistent with the only modest correlation between N-cadherin mRNA levels across tumor types or Gleason grade (Fig. 7k,l).
3. What are the puncta shown in Fig. 5e? Are they endosomes? Is there more N-cadherin in endosomes in Arf3-depleted cells?

Do these puncta colocalize with Rab11FIP4? This all gets at the potential mechanisms through which Arf3 might control N-cadherin behavior.

4. Two Arf effectors had similar phenotypes when depleted - PSD/EFA6A (an ArfGEF) and Rab11FIP4 (dual Rab/Arf effector). EFA6A is odd, since EFA6-family ArfGEFs are reported to be specific for Arf6. The authors could test this easily using a pulldown assay. Is Arf3 activity decreased upon depletion of EFA6A relative to Arf6 activity?. Conversely, it is certainly plausible that Rab11FIP4 is an Arf3 effector - both are present on Rab11-positive recycling endosomes. Is Rab11FIP4 distribution altered in Arf3-deficient cells?

Response to reviewers

We thank the reviewers for their extremely positive assessment of our work. In this response, Reviewer comments are presented in bold, our responses in non-bold. We number reviewer comments and our responses for ease of reference. In the main manuscript, altered text is indicated in blue for ease of review. In this response, we underline reference to new data in figures.

Reviewer #1 (Evidence, reproducibility and clarity (Required)):

This study represents a tour de force in advancing our understanding of the Arf family and its associated regulators/effectors in controlling the morphology of individual cells and collective behaviours in 3D ECM. The authors use shRNA-mediated knockdown in high throughput imaging and AI approaches to define the influence of the ARFome on the morphology of prostate cancer PC3 cells growing as acini structures in 3D ECM. This allowed classification of ARFs and effectors/regulators on the basis of associated phenotypes, and this in itself is a useful resource for the community. It also drove the authors to focus on ARF3 and its regulator ESD and effector Rab11-FIP4. Analysing cell motility in elegant 2D and 3D assays showed that ARF3 levels control the collective migration of PC3 cells in 3D invasion, and the authors relate these findings to an interaction between Arf3 and N-cadherin. Using a mouse model of intra-prostatic injection of PC3 cells they demonstrate clear links between ARF3 levels and metastasis in vivo, and patient data further supports the link between Arf3, N-cadherin and metastasis. Experiments are well controlled and complex data are beautifully presented. In general data support the conclusions, where this is not as clear is highlighted below.

Major comments:

1. Figure 4: The use of the Arf3/4 chimeras is an interesting approach, used to show that the ARF3 C-terminus is important for its function related to migration/invasion. However, the effect this has is not clear- it is not GTP loaded efficiently and may therefore act as a dominant negative.

We agree with this point. That the ARF 3N/1C chimera may be acting as a dominant negative fits with our data: decreased invasion compared to control (Fig 4k). We have updated the manuscript results section to explicitly state and discuss this point.

Furthermore the authors do not indicate which intracellular compartment ARF3 associates with, or if this is altered when the ARF3 C-term is replaced by that of ARF4 (ARF3N/1C).

We add confocal imaging of ARF3-mNG and each ARF1/3 chimera, in 3D acini (new data Fig S3I). This indicates that each of ARF3-mNG and the chimeras co-localised with markers of the Golgi apparatus (GM130) and recycling endosomes (RAB11). Although the 3N/1C chimera was poorly GTP loaded (as above) a significant pool also localised to cell junctions that did not co-localise with these markers. We include extensive co-localisation analysis of the key ARF3 cargo, N-cadherin, +/- ARF3, which further corroborates the major site of action of ARF3, at least in this model context, as the Rab11 recycling endosome (new data Fig. 5H-J; Fig. 6J-L).

2. Figure 5: Links to N-cadherin are clearly interesting, but the model proposed in Figure 5K is a little speculative. Clearly it is possible that ARF3/Rab11-FIP4 regulate N-cadherin trafficking such that loss of the pathway leads to degradation and gain promotes stability, but it is also possible

that expression levels are controlled at the level of transcription. This could be assessed by a simple surface labelling experiment in wt, overexpressing and knockdown cells, and/or by analysing localisation of N-cad with respect to ARF3 and late endosomes/lysosomes when ARF3 levels are manipulated. Does ESD knockdown similarly impact N-cad?

We have directly addressed this point with multiple experimental approaches. Collectively, these new experiments indicate that ARF3 controls turnover of post-endocytic N-cadherin, seemingly at the Rab11 endosome. A summary of this new data is:

- Despite N-cadherin protein levels being decreased upon ARF3 knockdown, and increased upon ARF3 overexpression (Fig. 6B,D), new qPCR data (new Fig. 6E) indicates that, unexpectedly, *N-cadherin* mRNA levels are decreased in both ARF3 knockdown and overexpression conditions. We do not know why this occurs this way. However, this strengthens a role for ARF3 control of N-cadherin protein levels, as ARF3 overexpression increases N-cadherin protein despite there being less N-cadherin mRNA than in control cells.
- Flow cytometry (new data Fig. 6M,N) indicates no change in N-cadherin surface expression upon loss of ARF3. In contrast, a significant increase occurs upon overexpression ARF3.
- Cell-surface labelling (biotinylation) of N-cadherin to analyse its turnover from the surface (new data Fig. 6O) indicates that enhanced turnover upon ARF3 knockdown or, conversely, attenuated turnover upon ARF3 overexpression.
- We have also analysed the subcellular localisation of N-cadherin using immunofluorescence. We observed fewer, smaller N-cadherin-positive puncta upon loss of ARF3 and increased number and size of N-cadherin-positive puncta upon ARF3 overexpression (new data Fig. 6H-I). When segmenting the cell (juxtannuclear, cytoplasmic and peripheral regions), we observed no positional bias in these changes.
- Depletion of the key effector of ARF3 that we identified, RAB11FIP4, consistently decreased N-cadherin levels (Fig. S4J). We examined N-cadherin levels by western blot upon PSD knockdown but didn't include the data here since we found the results highly variable across multiple PSD shRNA sequences.

3. Figure 6: The metastasis experiments are highly relevant to metastatic prostate cancer. Essentially the overall conclusion that high Arf3 (overexpression) suppresses and low Arf3 (knockdown) supports metastasis are well supported by the data, and the wildtype Arf3 levels sit in between (hence trends are observed but aren't statistically significant). Here it would be interesting to compare ARF3 levels in patient tumours with those in wt, overexpressing and knockdown PC3 cells in the mouse model (if sections are available) to give confidence that the overexpression is within the physiological range. If it were also possible to analyse N-cadherin levels in tumours or metastases that would provide an even stronger case for the mechanism proposed.

We agree that comparing the xenograft mouse model expression of ARF3 to that of patient sections from tumours/metastases to infer these being in the same physiological range is intriguing. However, this is not technically feasible. This is because any comparison between these very different sample types (patient and xenograft mouse model) requires that all samples collected with the same experimental conditions. Such conditions could include time from surgery to embedding of samples, fixation methodology, embedding methodology, antigen retrieval methodology, or a number of other variables. All of these could affect the resulting staining for ARF3 in ways that preclude such analyses being reliable.

In silico analysis of ARF3 mRNA levels in PC3 cells performed at the same time as patient samples (Taylor, Cancer Cell, 2010; GSE21032) indicate that PC3 are on the higher end of, but still in, the physiological range of ARF3 expression in patient samples (Reviewer Only Figure 1). However, this analysis should be considered in context: this involves comparison of mRNA from a tissue sample, including all sample types present, to an isolated cancer cell line. Therefore, a direct comparison between these two very different modalities is not appropriate.

Our data predict that we should expect high N-cadherin protein levels in tumours and metastasis with high ARF3 expression, and the inverse in low ARF3 expression. Note that it is N-cadherin protein, not mRNA, that is predictive of survival in prostate cancer patients (Fig. 9M,N). N-cadherin protein and N-cadherin/CDH2 mRNA are poorly correlated ($r=0.1673$; Fig S5G).

We examined N-cadherin immunohistochemical labelling in PC3 murine intraprostatic xenografts upon ARF3 knockdown or overexpression. We performed serial sectioning on primary tumour blocks and selected samples for staining where the primary tumour was easily detected and readily available.

Unexpectedly, N-cadherin was heterogeneous in tumours, with variable regions of both N-cadherin-positive and -negative labelling, as well as regions of N-cadherin with different intensity (Fig. 8G, new data). ARF3 manipulation affected the homogeneity of N-cadherin distribution across tumours. ARF3-overexpressing tumours displayed large areas of somewhat homogeneous N-cadherin expression with 40% of tumours exhibiting greater than 40% N-cadherin positivity. In comparison, 25% of ARF3-depleted tumours displayed this homogeneity, instead displaying larger patches of weak or no N-cadherin labelling, with some regions of often smaller N-cadherin positivity (Fig. 8G, H). Quantitation of the region of each tumour positive for N-cadherin and the weighted histoscore (Fig. 8H,I, new data) trended towards the corresponding effects observed *in vitro* for ARF3 depletion vs overexpression, however these did not reach statistical significance.

This experiment set uncovers an important point that we would have missed without the reviewer's excellent suggestion. Rather than having a 'high-' or 'low-N-cadherin' tumour, distinct regions within a tumour can be high or low for N-cadherin. ARF3 appears to be part of the mechanism driving this heterogeneity: homogeneous experimental ARF3 OX or KD pushes the tumour towards homogeneity of N-cadherin levels, whatever those levels are, rather than simply making levels high or low.

One caveat to these experiments is that our mouse experiment was powered to analyse primary tumour (rates between 53-60%) and macrometastasis incidence. Due to the high variability of N-cadherin expression levels within tumours, we now know that we have insufficient sample numbers available to power the requested labelling experiment appropriately. It may be that a difference in absolute levels of N-cadherin could be altered in ARF3-manipulated tumours rather than the observed homogeneity in labelling change. This would require additional murine experiments. We do not feel this is warranted given the clarifying, extensive molecular details we have added to the manuscript.

Minor comments:

4. Figure 2: The classification of phenotypes into groups is interesting, but the trends in some groups (eg Group4, 6 and 7) seem very similar. It wasn't clear to me if these are AI generated?

Also, knockdown of individual ARFs is often in different groups- is this a reflection of knockdown level?

For detection of phenotype classes, shRNAs were assigned into 7 groups based on a dendrogram from clustering the fold change over time for each shRNA in Round, Spindle, and Spread phenotypes compared to the control (Scr) value (see Fig. S1K). The dendrogram was generated using hierarchical clustering of heatmap data by complete linkage of Euclidian distances between samples. This is described in 'ARFome shRNA screen' section of Methods.

For the shRNA screen we used 1 or 2 independent shRNA sequences/target if they had previously been validated by qPCR, or 5 sequences/target if unvalidated. This point now clarified in the 'ARFome shRNA screen' section of Methods. As the reviewer notes some of the phenotype groups we describe exhibit similar trends with the main difference being the degree to which the changes in each phenotype occur. ShRNAs against the same target appear in different groups as they reflect different levels of knockdown efficiency. We now explicitly state this in the results section. To account for this, any targets further assessed were independently knocked down using shRNA sequences expressed in a constitutive lentiviral vector with antibiotic selection (pLKO.1 puromycin vector; described in Supplementary Table S2) and knockdown efficiency confirmed by western blot.

Reviewer #1 (Significance (Required)):

The manuscript provides a very significant advance in our understanding of the function of the ARFome with respect to cell morphodynamics. The first two figures and supporting data represent a fantastic resource for the field, and the remaining figures provide new insight into the function of ARF3 in collective cell movement and metastasis in mouse models and patients. Whilst ARF6 and its function in cell migration/invasion/metastasis is well studied, ARF3 has received relatively little attention. This study is therefore of broad interest to the trafficking community, and the new links between ARF3 and invasion/metastasis are broadly of interest to the cell biology and cancer communities. The mechanistic link between N-cadherin and ARF3 is fairly well defined and the fact that high/low levels of both correspond to improved/poor outcomes is a major strength of the study.

Expertise: Vesicle trafficking/cell migration/invasion/cancer

Reviewer #2 (Evidence, reproducibility and clarity (Required)):

In this manuscript, Sandilands et al. analyze the role of ARF GTPases ARF1 and ARF3 in human prostate cancer cell setting with specific focus on 3D versus 2D growth and invasion. The study connects to previous work where the authors conducted similar studies with ARF GTPase exchange factor IQSEC1 acting as an invasion promoting factor. Now, the authors interrogate the "ARFome" in prostate cancer cell line PC3 by use of a lentiviral shRNA library. The authors conduct detailed 3D and 2D cell culture analyses to identify specific differences in cell morphology between the different single knockout clones, showing loss of ARF1 and ARF3 as key switches of a spindle like morphology that is associated with enhanced migration and invasion. Interestingly, the authors find ARF3 functioning significantly dependent on the C-terminal region, and, further, that ARF3 is a direct companion of N-cadherin levels, whose downregulation leads to enhanced migratory capability.

The main findings of the manuscript are: (1) ARF1 and ARF3 knockdown elicits key-differences in 3D morphology and migratory capacity, (2) specifically ARF3 is associated with maintenance of N-cadherin levels via PSD and RAB11FIP4 effectors, (3) whose downregulation leads to enhanced metastatic spread in a orthotopic xenograft mouse model, and (4) lowered N-cadherin protein / ARF3 mRNA levels identify more aggressive human prostate tumors.

Analyses and experiments conducted in the manuscript are highly extensive, they follow robust methods and are well controlled. Results described are highly detailed and are impressively visualized and presented. Key data are highlighted, and the manuscript is clearly structured. However, some data could be described and argued more concisely, which would strongly support the results shown. I recommend publication after some minor but important changes.

Major comments:

1. The entire results are based on studies of a single cell line PC3 derived from a highly aggressive metastatic lesion. To infer such an essential principle of tumor invasion and migration from this may be a bit precarious, and perhaps this principle should also be demonstrated in another cell line. The choice of PC3 cells and its implications should at least be discussed.

We update the results section to state explicitly why we use PC3, and our justification of the use of these as an excellent model to study prostate cancer cell behaviour:

“When PC3 cells are plated on a thin coat of extracellular matrix (ECM) as a suspension of single cells in low percentage ECM-containing medium, they form heterogenous multicellular structures polarised around a central lumen, which we term acini. We used these PC3 acini to examine ARFome contribution to 3D morphogenesis as (i) they have high levels of all ARF GTPases, (ii) they, upon intraprostatic xenograft, provide a model for metastatic tumorigenesis, and (iii) we have shown that they can be used to identify ARF GTPase modules that regulate 3D invasion, in vivo metastasis, and predict patient survival.”

In addition to expressing all of the ARFome components (Fig. S1J), they have elevated expression of almost all ARF GTPases compared to other prostate cancer cell lines (Fig. S1A-G). We have used these to identify an ARF GEF (IQSEC1) that regulates invasion in vitro and metastasis in vivo, findings which also could be used to identify prostate cancer patients with poor clinical outlook (PMID: 33712589). Moreover, these cells are very plastic and we have demonstrated that we can use their inherent plasticity to identify molecular pathways that switch cells between different 3D behaviours. Therefore, we feel that these cells are highly useful as a model to understand complex biology, but also identify molecular players that can be clinical and translationally important.

2) Results showing cell morphology page 6 / Fig.2: end of paragraph: stating "normally suppressing invasion": seems too far at this state of the manuscript, as these experiments are shown in the next section. maybe better "involved in preservation of a rounded phenotype". Results Suppl.Fig.3f: please use the same colors for the morphologies as in Fig. 2 etc. (round - red, spindle - green, spread - blue).

We have now changed the text and altered the pseudocolouring used for 2D morphologies so that they match the colours used for 3D morphologies (Round - red, Spindle - green and Spread - blue) in Supplementary Fig. 2F.

3) Conclusive sentences should not be put at the beginning of an experiment, before one can know its outcome: Results page 8, Fig.4g, bottom: "This revealed that the C-terminus of ARF3 is required for sheet type invasive activity" maybe put that rather as a conclusion of the whole section.

We have now removed this sentence.

4) Results showing 2D migration and 3D invasion: In the illustrations of migration and invasion assays shown throughout Figs 3-5 and Suppl. Figs 3 and 4, please clearly state and indicate for each case whether this is 2D migration or 3D invasion, as these two assays are very similar, which is a little confusing throughout the manuscript. Suppl.Fig.3e,k,l: is this 2D or 3D? Results Fig.4b+d and Fig.5g: please amend "3D invasion".

We apologise for any confusion. We have now added labels to all such assays.

5) Summarizing sketch Results Fig.5k: in the scheme on the right side indication of respective presence / absence of ARF3/N-cadherin is missing. Which state induces which condition? Please amend.

ARF3 and N-cadherin labels have now been added to schema, which is now Fig. 7E.

6) ARF3 suppresses PC3 xenograft metastasis shown in Fig.6: N-cadherin stainings of mouse xenografts and metastases are missing.

We have now added this as new data Fig.8. This reveals that rather than controlling set levels of high or low N-cadherin (from OX or KD, respectively) ARF3 overexpression effects the homogeneity of N-cadherin expression across tumour cells. Please also see response to Reviewer 1 Comment 3.

7) Patient data ARF3 mRNA correlations Fig.7 and Suppl.Fig.6ab / Results page 11: the whole section describing ARF3 mRNA levels in diverse tumor types is too long and a little bit confusing: maybe shorten the text, put Fig.7f+g supplemental, and please indicate the combined GENT2 database in Fig. S6b.

We have completely rewritten this section to make it concise and (hopefully) clearer. This data is now in Fig. 9, which includes some data being moved to Fig. S5. We have indicated GENT2 on the supplementary figure (Fig. 5B, F) as requested. This also addresses your query number 8 below.

8) Patient data CDH2 mRNA/protein correlations: Fig.7 and Suppl.Fig.6cd / Results page 12: one should also shorten and sharpen this section. Please also exchange Suppl. Fig. panels 6 dc to have the same order as Suppl. Fig.6ab. Regarding the detailed analyses of CDH2 mRNA levels, make a too long story short, essential are N-cadherin protein levels, and these results shown in Fig. 7 hik and Fig. 7rst should be enlarged and highlighted. All other data (Fig. 7jl) and the right bars of panels Fig. 7mno, as well as Fig. 7pq are rather supplemental.

Please see response to point 7 above.

9) The finding of elevated N-cadherin levels correlated with reduced invasion/migration and according better tumor outcome is surprising, particularly with regard to the quite established "EMT" dogma of N-cadherin driven single cell migration. Could you go into more detail about this property of N-cadherin driven mode of reduced tumor spread in the discussion?

Much of the differential role of E-cadherin and N-cadherin comes from studies on cadherin switches, wherein cells express either E-cadherin or N-cadherin. Much less is known about what happens when both are simultaneously co-expressed (which is the situation we have studied). Indeed, the dogma that E-cadherin anti-invasive and N-cadherin is for invasion has been challenged in recent years from the work of groups such as Andy Ewald and Peter Friedl, who have showed that E-cadherin is required for collective invasion and metastasis. The concept of E-cadherin= no invasion vs N-cad = invasion is now understood not to be universal and an oversimplification. We therefore expand the discussion section of the manuscript regarding the point.

Minor comments:

1) Manuscript title: maybe rephrase and reverse order of events: first invasion, then metastasis?

Great suggestion. We have changed the manuscript title to 'The small GTPase ARF3 controls invasion modality and metastasis by regulating N-cadherin levels' as suggested.

2) Results Suppl.Fig.1ij: which cells are shown? Please amend RWPE-1 and PC3.

We update the figure to label that these are PC3 (this is now Fig. S1H-I).

3) Results page 5: although already described in Nacke et al 2021, please explain the term "acinus".

We now clear define acini/acinus in the results section. Please see our response to your point 1.

4) Results page 6: maybe introduce an additional subchapter? Some subchapter titles could be more explicative.

We have added an additional subchapter heading here as suggested.

5) Results page 6, 2nd paragraph: please describe what was done: "revealed that knockdown of ..."

We have now added the following to p6 – 'PC3 cells stably expressing shRNA (two per gene) to ARF1 or ARF3 were cultured in ECM and imaged every hour for 96 hours, as described for the ARFome shRNA screen. Size, shape and movement features were measured for each acinus and machine learning classifications applied to categorise and quantify Round, Spindle and Spread phenotypes.'

6) Results page 7: please explain more detailed class I ARFs, which ARFs are included in this class?

We discuss which ARFs are included in each class and explain Class 1 ARFs in more detail in the Introduction section.

7) Results page 7, Fig.3f-j and 2nd paragraph: maybe better switch: first 3D, then 2D? Results Fig.3j: maybe amend indication of knockdown "KD" as indicated in Figure 3b-e.

We have switched the order of 2D and 3D on p7 and added 'KD' to labels for Figure 3F-J.

8) The conclusion on page 8 top "this suggests that a function of ClassI ARFs may be to regulate molecules that control collective behaviours" is quite broad, please be more specific.

Upon reflection this sentence was superfluous and we have removed.

9) Suppl.Fig.4n: should be named "I"?

This numbering has changed as we carried out revisions.

10) Results page 8, introducing sentence: please formulate more clearly and following the previous results.

We have changed this sentence to better reflect previous results. 'Given our observations that depletion of ARF1 or ARF3 altered shape and movement in both 2D and 3D (Fig. 2E-G, 3B-E) we examined whether over-expression of Class I ARFs would also affect these processes.'

Reviewer #2 (Significance (Required)):

Contents/Level of interest/merit:

This study by Sandilands et al. analyzes cellular models of ARF-GTPase linked changes of cell morphology and migration to detect altered prostate cancer cell metastatic behaviour. Understanding the contribution of specific ARF GTPases in cancer cell shape and movement might help to identify markers of disease progression and metastasis.

Strengths/Conclusions:

The authors perform whole ARF-compendium knockdown and conduct detailed data analysis and visualization. The authors perform morphological analyses and conduct migration and invasion studies. Mechanistically, they confirm expression changes of N-cadherin, the key adhesive protein that is regulated. This study underlines the importance of analyzing subtle GTPase pathway differences by detailed morphological observations and methods.

Comment/Weakness:

The authors show extensive and detailed data that have been thoroughly analyzed, and results are presented and described fluently. There is a clear sequence of results description that is presented detailed. Form and contents of the paper is sound. The experiments are highly connected to previous experiments and data and this is also the major drawback of this manuscript: there is a lack of clear description of what is shown because it is already presupposed. Therefore, some sections should be worded and presented in more detail to present results more explicitly. The manuscript can be accepted with minor but essential revisions.

Reviewer #3 (Evidence, reproducibility and clarity (Required)):

Summary:

Provide a short summary of the findings and key conclusions (including methodology and model system(s) where appropriate).

The manuscript is dedicated to a study of functional roles of a panel of cell migration regulatory factors, and notably the highly homologous family of ARF GTPases. The chosen model is a prostate cancer cell line, used in a number of assays in culture, as well as in a study of primary tumor formation and metastasis in mice. The authors apply a rigorous quantitative approach to their

assays of 2D and 3D migration in culture, and use artificial intelligence for the analysis of results, thus upgrading their work from mere phenotypic observations, and gaining statistically significant results. The main finding of the study is the discovery of a unique role of ARF3, a regulatory protein that is shown to control a switch between individual and collective cell migration depending on its abundance. In fact, a depletion of ARF3 leads to an increased individual cell migration and invasion, and to increased metastasis formation in mice, whereas an overexpression of ARF3 favors a sheet-like collective cell migration, which is also more efficient than control in culture, but does not induce metastasis in vivo. This phenomenon appears to depend on the levels of cellular N-cadherin, that is shown to be positively regulated by ARF3 on the protein level, by a mechanism that remains unclear. Finally, the authors analyze the expression of ARF3 and N-cad in a variety of tumors of different origins and grades, and attempt to show the prognostic value of these factors for progression-free survival.

Major comments:

- Are the key conclusions convincing?

The majority of conclusions are convincing, with the exception of the observations I will address in the next paragraph.

- Should the authors qualify some of their claims as preliminary or speculative, or remove them altogether?

1. In Fig. 2B, the phenotypes 1 (Spread, pink line) and 5 (Spindle, yellow line) are described in the text as showing a "modest but robust increase". No such increase is evident by eye, and if it is statistically robust, the information should be presented in the Figure, or the statement should be modified/removed from the article.

We believe that this point may be related to some confusion. In the text we state 'both ARF1 and ARF3 had one shRNA in each of Phenotype Groups 5 and 1, which displayed a modest but robust increase in Spindle and Spread behaviours, respectively.' In Figure 2B these changes are visible as Spread is represented in graphs by blue line and Spindle by green line. We think the pink (or red) line the reviewer is describing is actually the Round phenotype which is indeed unchanged or decreased.

Statistical significance is shown as s.e.m. on each graph in Figure 2B as a shaded region around each line and is described in Figure Legend. However, in some cases it may be hard to see this region as the effect across replicates is highly concordant and accordingly the s.e.m. is very small.

2. In Fig. 2E, the lower middle panel shows a dramatically increased quantity of acini, whereas the authors specifically say that proliferation is not impacted by the KD of ARF3, and indeed, the KD2 looks very much like the control in this respect. It is misleading, and a more typical panel should be presented for ARF3_KD1.

We have now changed the image in Figure 2E to show a more typical panel for ARF3_KD1.

3. In Fig. 3, the authors study the effects of simple versus double KD of ARF1 and ARF3, and conclude that a double KD leads to a phenotype "midway" between the two simple KDs. However, with regard to the 3D invasion assays (Figs. 3DEJ), it looks like a double KD is less efficient than either of the simple ones, as if ARF1 and 3 were partially mutually dependent in this regulation. It is not clear what "midway phenotype" the authors are talking about.

We agree that this statement was confusing and removed the 'midway' concept from this section

4. Would additional experiments be essential to support the claims of the paper? Request additional experiments only where necessary for the paper as it is, and do not ask authors to open new lines of experimentation.

The authors make a strong case for physical interaction and mutual stabilization of ARF3 and N-cad, though the negative regulation of N-cad following ARF3 depletion is not obvious from Fig. 5B (positive regulation is very clear). Moreover, it is difficult to understand why a total disappearance of ARF3 has such a discreet effect on N-cad, whereas a very modest overexpression of ARF3 leads to such a dramatic increase of N-cad. Perhaps, some experiments with proteasome inhibition (using MG132, for example) could substantiate the authors' claim about the mutual stabilization of the two proteins.

Please see our response to Reviewer 1, point 2. We now demonstrate that ARF3 regulates the turnover of post-endocytic N-cadherin, by controlling trafficking of N-cadherin into a RAB11FIP4-positive population of RAB11 recycling compartments. The consequence of this is to increase (ARF3 knockdown) or attenuate (ARF3 overexpression) the rate of turnover of post-endocytic N-cadherin. It may be that ARF3 regulates the turnover of a small but crucial pool of N-cadherin from Rab11 endosomes, which is counter-balanced by biosynthetic N-cadherin production. This may explain the significant but modest effects of ARF3 depletion on N-cadherin levels, but robust effect upon ARF3 overexpression.

It was straightforward to overexpress ARF3 and increase its levels. But we note that while depletion of ARF3 induces strong phenotypic effects, these shRNAs result in about 75% reduction in expression. We were unable to recover clones using CRISPR for ARF3 knockout. It may also be that the remaining small amount of ARF3 is sufficient to maintain the N-cadherin that remains in cells.

Are the suggested experiments realistic in terms of time and resources? It would help if you could add an estimated cost and time investment for substantial experiments.

Yes, the experiments are realistic and should not take more than a month.

- Are the data and the methods presented in such a way that they can be reproduced?

Yes.

- Are the experiments adequately replicated and statistical analysis adequate?

I am not sufficiently qualified in artificial intelligence algorithms to judge this part of the study. In general, as mentioned above, all differences characterized as "significant" or "robust" should have a statistical basis for this statement, which was not always the case in the manuscript.

Minor comments:

- Specific experimental issues that are easily addressable.

Please see above.

- Are prior studies referenced appropriately?

To the best of my knowledge, yes.

- Are the text and figures clear and accurate?

Yes, with the exceptions described before.

- Do you have suggestions that would help the authors improve the presentation of their data and conclusions?

Please see above.

CROSS-CONSULTATION COMMENTS

I fully agree with the detailed and careful analysis made by the reviewers 1 and 2. I do not have any additional comments.

Reviewer #3 (Significance (Required)):

- Describe the nature and significance of the advance (e.g. conceptual, technical, clinical) for the field.

The study shows, for the first time and in a very clear way, that the small GTPase ARF3 has a unique function in determining the pattern of human cancer cell migration, that this function depends on the C-terminal domain of ARF3 and on N-cadherin (by mechanisms that remain to be elucidated), and that this phenomenon is important for metastases formation *in vivo*.

- State what audience might be interested in and influenced by the reported findings.

The study will be interesting for the field of cell migration, but also for specialists in cancer and metastasis formation.

- Define your field of expertise with a few keywords to help the authors contextualize your point of view. Indicate if there are any parts of the paper that you do not have sufficient expertise to evaluate.

Cell migration 2D 3D, acini assay, RAC-WAVE-ARP2/3 pathway. I am not sufficiently qualified to evaluate the robustness of the artificial intelligence algorithms, nor to judge the relevance of the analysis presented in Fig. 7.

Reviewer #4 (JCB) (Comments to the Authors (Required)):

This manuscript describes an interesting and novel role for the small GTPase Arf3 in metastasis of prostate cancer. As the authors note, high sequence homology among the 5 human Arfs and potential redundancies among their GEFs and GAPs has historically made it difficult to attribute specific functions to individual Arfs. Here the authors use an impressive array of technical approaches to define a very specific role for Arf3 in controlling 3D, but not 2D migration of PC3 prostate cancer cells *in vitro*. These *in vitro* studies are paralleled by *in vivo* studies showing that metastasis, but not growth, of PC3-derived tumors is modulated by Arf3. The specificity for Arf3 is surprising because Arf3 and Arf1 differ in only 7 amino acids, 4 at N-terminus and 3 at the C-terminus. Chimeras suggest that the C-terminal residues are more important, however a corresponding chimera containing the Arf3 N-terminus and Arf1 C-terminus doesn't bind GTP

making it difficult to draw precise conclusions.

As noted above, this study is an impressive technical achievement - it combines high-throughput knockdown of not only the 5 human ARFs but also all known GEFs, GAPs and most effectors (i.e. the ARFome). The authors analyzed cell behavior for each knockdown in both 2D and 3D cultures over 4 days and used machine learning to classify corresponding phenotypes. Importantly, a distinction between Arf1 and Arf3 phenotypes was only observable in 3D.

This manuscript has many strengths, including the breadth of the analysis (encompassing the entire ArfOME), the range of analytical methods used and the careful quantitation of most of the outcomes. There are, however, a few weaknesses that will need to be addressed:

1. The data suggest that Arf3 acts to suppress prostate cancer metastasis through a regulated interaction N-cadherin. The data showing interaction of Arf3 with N-cadherin (Fig. 5f) and colocalization with it on intracellular compartments (Fig. 5e) are convincing (although the colocalization needs to be quantified). What is lacking is any measurement of N-cadherin localization in the presence and absence of Arf3. Is the ratio of surface/intracellular N-cadherin altered? This could be readily measured by flow cytometry, or by careful image quantitation.

Please see our response to Reviewer 1, point 2, for our response to this point. This details our extensive additional new experiments demonstrating that ARF3 controls the turnover of post-endocytic N-cadherin, likely at RAB11FIP4-positive RAB11 recycling endosomes.

2. How do the authors think Arf3 acts to control N-cadherin surface level? Does it control secretion and/or recycling or does it act allosterically to stabilize the protein? Does inhibition of lysosomal enzyme activity restore N-cadherin levels in Arf3-depleted cells (Fig. 5b,d)? This would be consistent with the only modest correlation between N-cadherin mRNA levels across tumor types or Gleason grade (Fig. 7k,l).

Please see our response to Reviewer 1, point 2, for our response to this point and your point 1 above.

3. What are the puncta shown in Fig. 5e? Are they endosomes? Is there more N-cadherin in endosomes in Arf3-depleted cells? Do these puncta colocalize with Rab11FIP4? This all gets at the potential mechanisms through which Arf3 might control N-cadherin behavior.

We have carried out extensive IF experiments. Summary of the new data are:

- ARF3 and RAB11FIP4 interact (new data Fig. 5F,G) and colocalise in puncta (new data Fig. 5H). The number, size and distribution of RAB11FIP4 puncta are not affected by ARF3 knockdown (new data Fig. 5I,J), in line with RAB11FIP4 localisation being primarily regulated by RAB11.
- N-cadherin in puncta co-localises with ARF3 (Fig. 6G). ARF3 KD decreases, while ARF3 overexpression increases, the number and size of N-cadherin puncta (new data Fig. 6H,I).
- N-cadherin puncta label for markers of recycling endosome compartments (RAB4, RAB11, RAB11FIP4) and late endosomes/lysosomes (LAMP2) (new data Fig. 6J). Approximately 20% of N-cadherin puncta overlap with each of these markers (new data Fig. 6K,L). ARF3 knockdown decreases the number of these puncta (new data Fig. 6H), which are modestly more positive for RAB11FIP4 (new data Fig. 6K).

Functionally, increasing ARF3 levels *a)* increases total N-cadherin levels, *b)* increases the amount of N-cadherin on the surface, *c)* increases the number of Rab11FIP4-positive N-cadherin puncta, and *d)* attenuates post-endocytic N-cadherin. This suggests that a major role of ARF3 is to control transit of N-cadherin into RAB11FIP4-positive recycling endosomes to promote recycling rather than degradation of post-endocytic N-cadherin.

4. Two Arf effectors had similar phenotypes when depleted - PSD/EFA6A (an ArfGEF) and Rab11FIP4 (dual Rab/Arf effector). EFA6A is odd, since EFA6-family ArfGEFs are reported to be specific for Arf6. The authors could test this easily using a pulldown assay. Is Arf3 activity decreased upon depletion of EFA6A relative to Arf6 activity?. Conversely, it is certainly plausible that Rab11FIP4 is an Arf3 effector - both are present on Rab11-positive recycling endosomes. Is Rab11FIP4 distribution altered in Arf3-deficient cells?

The reviewer is correct that PSD/EFA6A has traditionally be considered an 'Exchange Factor for ARF6', especially given its namesake, and previous investigations that indicated that PSD/EFA6A was a poor GEF for class I ARFs in solution. However, Jacqueline Cherfils' group demonstrated in the presence of membrane, PSD/EFA6 is a potent GEF for the other class I ARF, ARF1 (PMID: 25114232). In our system, we now show upon PSD knockdown that *a)* **total** levels of ARF3, but not ARF6, are increased, and that *b)* despite this, **GTP loading** of ARF3, but not ARF6, is attenuated (new data Fig. 5A,B). This supports identification of PSD as a GEF for ARF3 in this system.

We add new experiments confirming that ARF3 associates with RAB11FIP4 (new data Fig. 5F,G), and that these colocalise in puncta (new data Fig. 5H). The number, size and distribution of RAB11FIP4 are not affected by ARF3 knockdown (new data Fig. 5I,J), in line with RAB11FIP4 localisation being primarily regulated by RAB11. N-cadherin puncta co-localise with ARF3, RAB11FIP4 and RAB11. Knockdown of ARF3 decreases, while ARF3 overexpression increases, the number and size of N-cadherin puncta; these puncta modestly increase their overlap with RAB11FIP4 upon ARF3 depletion (new Fig. 6G-L).

Collectively, these data support a PSD (as GEF), ARF3 (as GTPase) and RAB11FIP4 (as effector) module, operating at Rab11 recycling endosomes, controlling post-endocytic N-cadherin turnover.

January 2, 2023

RE: JCB Manuscript #202206115R

Dr. David Bryant
University of Glasgow
CRUK Beatson Institute
Garscube Estate
Switchback Road
Glasgow G61 1BD
United Kingdom

Dear Dr. Bryant:

Thank you for submitting your revised manuscript entitled "The small GTPase ARF3 controls invasion modality and metastasis by regulating N-cadherin levels". We would be happy to publish your paper in JCB pending final revisions necessary to meet our formatting guidelines (see details below).

A. MANUSCRIPT ORGANIZATION AND FORMATTING:

Full guidelines are available on our Instructions for Authors page, <http://jcb.rupress.org/submission-guidelines#revised>. Submission of a paper that does not conform to JCB guidelines will delay the acceptance of your manuscript.

1) Text limits: Character count for Articles is < 40,000, not including spaces. Count includes abstract, introduction, results, discussion, and acknowledgments. Count does not include title page, figure legends, materials and methods, references, tables, or supplemental legends.

2) Figures limits: Articles may have up to 10 main figures and 5 supplemental figures/tables.

3) Figure formatting: Scale bars must be present on all microscopy images, including inset magnifications. Molecular weight or nucleic acid size markers must be included on all gel electrophoresis.

** Please include scale bars on inset images throughout.

4) Statistical analysis: Error bars on graphic representations of numerical data must be clearly described in the figure legend. The number of independent data points (n) represented in a graph must be indicated in the legend. Statistical methods should be explained in full in the materials and methods. For figures presenting pooled data the statistical measure should be defined in the figure legends. Please also be sure to indicate the statistical tests used in each of your experiments (either in the figure legend itself or in a separate methods section) as well as the parameters of the test (for example, if you ran a t-test, please indicate if it was one- or two-sided, etc.). Also, if you used parametric tests, please indicate if the data distribution was tested for normality (and if so, how). If not, you must state something to the effect that "Data distribution was assumed to be normal but this was not formally tested."

5) Abstract and title: The abstract should be no longer than 160 words and should communicate the significance of the paper for a general audience. The title should be less than 100 characters including spaces. Make the title concise but accessible to a general readership.

** Given a broad readership that includes developmental biologists, the term "morphogenesis" used in the abstract, while certainly accurate, might pose some confusion for readers. We suggest considering a change to render this term consistent with language elsewhere in the abstract, as appropriate.

6) Materials and methods: Should be comprehensive and not simply reference a previous publication for details on how an experiment was performed. Please provide full descriptions in the text for readers who may not have access to referenced manuscripts.

7) Please be sure to provide the sequences for all of your primers/oligos and RNAi constructs in the materials and methods. You must also indicate in the methods the source, species, and catalog numbers (where appropriate) for all of your antibodies. Please also indicate the acquisition and quantification methods for immunoblotting/western blots.

8) Microscope image acquisition: The following information must be provided about the acquisition and processing of images:
a. Make and model of microscope

- b. Type, magnification, and numerical aperture of the objective lenses
- c. Temperature
- d. Imaging medium
- e. Fluorochromes
- f. Camera make and model
- g. Acquisition software
- h. Any software used for image processing subsequent to data acquisition. Please include details and types of operations involved (e.g., type of deconvolution, 3D reconstitutions, surface or volume rendering, gamma adjustments, etc.).

10) Supplemental materials: There are strict limits on the allowable amount of supplemental data. Articles may have up to 5 supplemental figures. Please also note that tables, like figures, should be provided as individual, editable files. A summary of all supplemental material should appear at the end of the Materials and methods section.

13) ORCID IDs: ORCID IDs are unique identifiers allowing researchers to create a record of their various scholarly contributions in a single place. At resubmission of your final files, please consider providing an ORCID ID for as many contributing authors as possible.

Please note that JCB now requires authors to submit Source Data used to generate figures containing gels and Western blots with all revised manuscripts. This Source Data consists of fully uncropped and unprocessed images for each gel/blot displayed in the main and supplemental figures. Since your paper includes cropped gel and/or blot images, please be sure to provide one Source Data file for each figure that contains gels and/or blots along with your revised manuscript files. File names for Source Data figures should be alphanumeric without any spaces or special characters (i.e., SourceDataF#, where F# refers to the associated main figure number or SourceDataFS# for those associated with Supplementary figures). The lanes of the gels/blots should be labeled as they are in the associated figure, the place where cropping was applied should be marked (with a box), and molecular weight/size standards should be labeled wherever possible. Source Data files will be made available to reviewers during evaluation of revised manuscripts and, if your paper is eventually published in JCB, the files will be directly linked to specific figures in the published article.

WHEN APPROPRIATE: The source code for all custom computational methods published in JCB must be made freely available as supplemental material hosted at www.jcb.org. Please contact the JCB Editorial Office to find out how to submit your custom macros, code for custom algorithms, etc. Generally, these are provided as raw code in a .txt file or as other file types in a .zip file. Please also include a one-sentence summary of each file in the Online Supplemental Material paragraph of your manuscript.

B. FINAL FILES:

-- Cover images: If you have any striking images related to this story, we would be happy to consider them for inclusion on the journal cover. Submitted images may also be chosen for highlighting on the journal table of contents or JCB homepage carousel.

Images should be uploaded as TIFF or EPS files and must be at least 300 dpi resolution.

****It is JCB policy that if requested, original data images must be made available to the editors. Failure to provide original images upon request will result in unavoidable delays in publication. Please ensure that you have access to all original data images prior to final submission.****

****The license to publish form must be signed before your manuscript can be sent to production. A link to the electronic license to publish form will be sent to the corresponding author only. Please take a moment to check your funder requirements before choosing the appropriate license.****

Thank you for this interesting contribution, we look forward to publishing your paper in Journal of Cell Biology.

Sincerely,

Ian Macara, Ph.D.
Editor
The Journal of Cell Biology

Tim Fessenden
Scientific Editor
Journal of Cell Biology

Reviewer #1 (Comments to the Authors (Required)):

The authors have done an excellent job of addressing earlier reviewer concerns, including the addition of considerable new data. No further changes are necessary.